# One-step Optimal Transport via Regularized Distribution Matching Distillation

## Abstract

Unpaired domain translation remains a challenging task due to the need of finding a balance between faithfulness and realism. Diffusion-based methods for unpaired translation typically excel at realism, but require numerous inference steps and tend to offer suboptimal input-output alignment. Many of the optimal transport (OT) based methods, on the other hand, offer efficient few-step inference and reach superior input-output alignment, but heavily rely on adversarial training and inherit its shortcomings. In this paper, we propose a method called Regularized Distribution Matching Distillation (RDMD), which combines the best of both worlds. It replaces the adversarial training with diffusion-based distribution matching, addressing the typical shortcomings of OT methods and providing a strong initialization for the trained models. RDMD maintains the advantages of the OT methods by providing one-step inference and explicitly controlling the input-output faithfulness via regularization of the transport cost. We prove that in theory RDMD approximates the OT map and demonstrate its empirical performance on several tasks, including unpaired image-to-image translation in pixel and latent space and unpaired text detoxification. Empirical results show that RDMD achieves a comparable or better faithfulness-realism trade-off compared to the diffusion and OT-based baselines.

## 1 Introduction

Learning a mapping between two distributions from non-aligned data, a task known as *unpaired translation*, is essential when paired datasets are prohibitively expensive or impossible to collect. In computer vision, a prominent example is unpaired image-to-image translation (Isola et al., 2017; Zhu et al., 2017), which aims to preserve the cross-domain properties of an input image while changing its source-domain features to match the target. Common examples include transforming cats into dogs (Choi et al., 2020) or human faces into anime (Korotin et al., 2022).

Unpaired translation remains a fundamentally challenging problem due to the absence of input-output alignment. This implies that a desirable translation method should reconcile two competing objectives: *faithfulness*, which ensures the translated output preserves the core content of the source input, and *realism*, which requires the output to be indistinguishable from true samples of the target domain. Achieving an optimal balance in this trade-off is central to unpaired translation.

Current state-of-the-art approaches tend to excel at one objective at the expense of the other, a dichotomy that we summarize in Table 1. On one side, Diffusion models (DMs) (Ho et al., 2020; Song et al., 2020b; Dhariwal & Nichol, 2021; Karras et al., 2022) offer exceptional realism via high-quality generation. DM-based unpaired translation methods typically manipulate their latent space or sampling scheme to maintain alignment. Broadly, one-sided unpaired translation methods (Choi et al., 2021; Meng et al., 2021; Zhao et al., 2022) based on DMs commonly guide target DM sampling process towards samples similar to the input image from the source domain. Two-sided translation models (Su et al., 2022; Wu & De la Torre, 2023) enforce faithfulness by training an additional model on the source domain to ensure that a more content-rich source encoding is used for generating the target object. However, explicitly controlling faithfulness in DMs is *non-trivial* and constitutes their main drawback alongside their high inference costs.

Alternatively, unpaired translation can be formalized as an optimal transport (OT) (Villani et al., 2009; Santambrogio, 2015) problem, which consists of finding the *minimal-cost* mapping between

Figure 1: One-step translation between ImageNet classes with RDMD.

Table 1: Qualitative comparison of different families of unpaired translation methods. We denote strong advantage by ●, moderate advantage by ◐, and no advantage by ○. * stands for a realism that highly depends on the sufficient amount of data.

| Family | DM (1-sided) | DM (2-sided) | OT | RDMD |
|---|---|---|---|---|
| Faithfulness | ○ | ◐ | ● | ● |
| Realism | ● | ● | ●* | ◐ |
| One(few)-step | ○ | ○ | ● | ● |
| Data efficiency | ● | ● | ○ | ● |
| Theory | ◐ | ◐ | ● | ● |

distributions. This formulation provides significant advantages: a strong theoretical foundation, guaranteed faithfulness through explicit cost regularization, and highly efficient one-step inference. Despite these benefits, OT-based methods (Korotin et al., 2022; Gushchin et al., 2024b; Choi et al., 2024) typically rely on adversarial training to match the target distribution. This dependency makes them prone to training instabilities and limits their generative quality, preventing them from matching the realism of DMs.

Table 1 contextualizes our work in comparison with other method families. In particular, it highlights the downsides of the existing methods. We show limited faithfulness of the DM-based methods by providing the trade-off curves in Figure 3 and samples in Figure 5. In case of OT methods, the most significant problem is adversarial training. We highlight that OT-based methods that utilize it struggle with producing realistic samples in low-data regime (Figure 4 and Table 3) and may produce artifacts in general (Figures 12, 13, 14, 15).

In this work, we introduce a method called *Regularized Distribution Matching Distillation (RDMD)*, which overcomes shortcomings of both paradigms and achieves a better faithfulness-realism trade-off than diffusion-based and OT methods. The key idea behind RDMD is to replace the unstable adversarial objective in OT methods with a diffusion-based distribution matching loss (Yin et al., 2023; Nguyen & Tran, 2023).

We summarize our contributions as follows:

1. We provide a theoretical analysis of the method and show that with the novel objective, RDMD approximately solves the OT problem;

2. We emphasize *one-step* inference, *strong initialization*, and *fast convergence* of the method made possible due to using the diffusion paradigm and utilizing the pre-trained target DM;

3. We demonstrate that RDMD maintains quality even with significant data constraints, a common failure case of OT methods (Table 3, Figure 4);

4. We validate the applicability of RDMD across different modalities, including the unpaired image-to-image translation in pixel and latent space and text detoxification;

5. Our experiments show strong empirical results: RDMD surpasses OT methods in terms of realism and diffusion methods at faithfulness, achieving a better trade-off than the baselines on different unpaired image-to-image problems.

## 2 BACKGROUND

### 2.1 DIFFUSION MODELS

Diffusion models (Song & Ermon, 2019; Ho et al., 2020) are a class of models that sequentially perturb data distribution $p^{\text{data}}$ with noise, transforming it into a tractable unstructured distribution. Using this distribution as a prior and reversing the process by progressively removing the noise yields a sampling procedure from $p^{\text{data}}$. A common way to formalize diffusion models consists in defining distribution dynamics $\{p_t(\boldsymbol{x}_t)\}_{t \in [0,T]}$, obtained by adding an independent Gaussian noise $\sigma_t \varepsilon$ with progressively growing variance $\sigma_t^2$ to the original data sample $\boldsymbol{x}_0 \sim p^{\text{data}}$: $\boldsymbol{x}_t = \boldsymbol{x}_0 + \sigma_t \varepsilon$ [1].

Conveniently, the equivalent distribution dynamics can be represented via a deterministic counterpart given by the ordinary differential equation (ODE)

$$\mathrm{d}\boldsymbol{x}_t = -\frac{1}{2} \left(\sigma_t^2\right)' \cdot \nabla_{\boldsymbol{x}_t} \log p_t(\boldsymbol{x}_t) \mathrm{d}t; \quad \boldsymbol{x}_0 \sim p^{\text{data}}, \tag{1}$$

where $\nabla_{\boldsymbol{x}_t} \log p_t(\boldsymbol{x}_t)$ is called the *score function* of $p_t(\boldsymbol{x}_t)$. Equation 1 is also called Probability Flow ODE (PF-ODE). This formulation allows one to obtain a *backward* process of data generation by simply reversing the velocity of the particle. In particular, one can obtain samples from $p^{\text{data}}$ by taking $\boldsymbol{x}_T \sim p_T$ and running the PF-ODE backwards in time, given access to the score function. The sampling procedure is essentially multi-step, which imposes computational challenges but enables control of the resources-quality trade-off.

Diffusion models learn score functions $\nabla_{\boldsymbol{x}_t} \log p_t(\boldsymbol{x}_t)$ of noisy distributions by approximating them via the Denoising Score Matching (Vincent, 2011) objective:

$$\min_\theta \int_0^T \beta_t \, \mathbb{E}_{p_{0,t}(\boldsymbol{x}_0, \boldsymbol{x}_t)} \| D_t^\theta(\boldsymbol{x}_t) - \boldsymbol{x}_0 \|^2 \mathrm{d}t, \tag{2}$$

where $D_t^\theta$ is called the denoising network and $\beta_t$ is some positive weighting function. The minimum in the Equation 2 is attained at $D_t^*(\boldsymbol{x}_t) = \mathbb{E}_{p_{0|t}(\boldsymbol{x}_0|\boldsymbol{x}_t)}[\boldsymbol{x}_0]$ and is related to the corresponding score function via Tweedie's formula (Efron, 2011) $s_t(\boldsymbol{x}_t) := \nabla_{\boldsymbol{x}_t} \log p_t(\boldsymbol{x}_t) = (\boldsymbol{x}_t - D_t^*(\boldsymbol{x}_t))/\sigma_t^2$ (also called the score identity). Therefore, diffusion models optimize the score functions of the perturbed distributions by learning to denoise objects at various noise levels via the denoiser $D_t^\theta$ and setting $\nabla_{\boldsymbol{x}_t} \log p_t(\boldsymbol{x}_t) \approx s_t^\theta(\boldsymbol{x}_t) = (\boldsymbol{x}_t - D_t^\theta(\boldsymbol{x}_t))/\sigma_t^2$.

### 2.2 DISTRIBUTION MATCHING DISTILLATION

Distribution Matching Distillation (Luo et al., 2024; Yin et al., 2023; 2024) aims to train a free-form generator $G_\theta(\boldsymbol{z})$ to match the given distribution $p^{\text{data}}$. Its input $\boldsymbol{z}$ is assumed to come from a tractable input distribution $p^{\text{noise}}$. Formally, matching two distributions can be achieved by optimizing the KL divergence $\mathrm{KL}(p^{G_\theta} \| p^{\text{data}})$ between the distribution $p^{G_\theta}$ of $G_\theta(\boldsymbol{z})$ and the data distribution $p^{\text{data}}$. However, the authors modify the functional to be tractable through the diffusion framework. They relax the original loss by using an ensemble of KL divergences between distributions, which are perturbed by the forward diffusion process:

$$\int_0^T \omega_t \, \mathrm{KL}\left(p_t^{G_\theta} \,\|\, p_t^{\text{data}}\right) \mathrm{d}t. \tag{3}$$

Here, $\omega_t$ is a weighting function, $p_t^{G_\theta}$ and $p_t^{\text{data}}$ are the perturbed versions of the generator distribution and $p^{\text{data}}$ up to the time step $t$. In theory, the minima of Equation 3 objective is attained if and only if (Wang et al., 2024, Thm. 1) $p^{G_\theta} = p^{\text{data}}$. In practice, the ensemble of KL divergences, which can be equivalently written as

$$\int_0^T \omega_t \, \mathbb{E}_{\mathcal{N}(\varepsilon|0,I)p^{\text{noise}}(\boldsymbol{z})} \log \frac{p_t^{G_\theta}(G_\theta(\boldsymbol{z}) + \sigma_t \varepsilon)}{p_t^{\text{data}}(G_\theta(\boldsymbol{z}) + \sigma_t \varepsilon)} \, \mathrm{d}t, \tag{4}$$

---

[1]This noising scheme is called Variance Exploding (VE) (Song et al., 2020b). While there are other noising schemes, such as e.g., Variance Preserving (VP), they are equivalent up to multiplication (Song et al., 2020a), so we stick to VE for simplicity.

produces gradient $\omega_t \left( s_t^{G_\theta} - s_t^{\text{data}} \right) \nabla_\theta G_\theta(z)$. It amounts to calculating the scores of noisy distributions at the point $G_\theta(z) + \sigma_t \varepsilon$ and performing backpropagation. [2]

Given this, the authors approximate $s_t^{\text{data}}$ with the pre-trained diffusion model, which we will denote $s_t^{\text{data}}$ as well with a slight abuse of notation. The whole procedure now can be considered as the distillation of $s_t^{\text{data}}$ into $G_\theta$. At the same time, $s_t^{G_\theta}$ represents the score of the noised distribution of the generator, which is intractable and is therefore approximated by an additional "fake" diffusion model $s_t^\phi$ and the corresponding denoiser $D_t^\phi$. It is trained on the standard denoising score matching objective with the generator's samples at the input. The joint training procedure is essentially the coordinate descent

$$
\begin{cases}
\min_\theta \int_0^T \omega_t \, \mathbb{E}_{\varepsilon, z} \log \dfrac{p_t^\phi(G_\theta(z) + \sigma_t \varepsilon)}{p_t^{\text{data}}(G_\theta(z) + \sigma_t \varepsilon)} \, dt; \\
\min_\phi \int_0^T \beta_t \, \mathbb{E}_{\varepsilon, z} \| D_t^\phi(G_\theta(z) + \sigma_t \varepsilon) - G_\theta(z) \|^2 \, dt,
\end{cases}
\tag{5}
$$

where the stochastic gradient with respect to the fake network parameters $\phi$ is calculated by backpropagation, and the generator's stochastic gradient is calculated directly as $\omega_t (s_t^\phi - s_t^{\text{data}}) \nabla_\theta G_\theta(z)$ with the scores are evaluated at the point $G_\theta(z) + \sigma_t \varepsilon$. Minimization of the fake network's objective ensures $s_t^\phi = s_t^{G_\theta} \Leftrightarrow p_t^\phi = p_t^{G_\theta}$. Under this condition, the generator's objective is equal to the original ensemble of KL divergences from Equation 3, minimizing which solves the initial problem and implies $p^{G_\theta} = p^{\text{data}}$.

### 2.3 Unpaired translation and optimal transport

The problem of unpaired translation consists of learning a mapping $G$ between the *source* distribution $p^\mathcal{S}$ and the *target* distribution $p^\mathcal{T}$ given the corresponding independent data sets of samples. When optimized, the mapping should appropriately adapt $G(x)$ to the target distribution $p^\mathcal{T}$, while preserving the input's cross-domain features. One way to formalize this is by introducing the notion of a "transportation cost" $c(\cdot, \cdot)$ between the generator's input and output and stating that it should not be too large on average. Monge's optimal transport (OT) problem (Villani et al., 2009; Santambrogio, 2015) follows this reasoning and aims at finding the mapping with the least average transport cost among all the mappings that fit the target $p^\mathcal{T}$:

$$
\inf_{G: G(x) \sim p^\mathcal{T}} \mathbb{E}_{p^\mathcal{S}(x)} c(x, G(x)),
\tag{6}
$$

which can be seen as a mathematical formalization of the domain translation task. In a practical setting, one can choose $c(\cdot, \cdot)$ to be any reasonable distance between images or their features that one aims to preserve, such as pixel-wise distance or the difference between embeddings.

## 3 Methodology

### 3.1 Regularized Distribution Matching Distillation

We build the method specifically for solving the Monge OT problem (Equation 6). To this end, we train a generator $G_\theta(x)$ to explicitly satisfy both requirements of the Monge problem: realistic samples $p^{G_\theta} \approx p^\mathcal{T}$ and low transport cost $\mathbb{E}_{p^\mathcal{S}} c(x, G_\theta(x))$. We first note that producing realistic samples can be done via minimizing the integral KL divergence

$$
\mathcal{L}(\theta) = \int_0^T \omega_t \, \text{KL}\Big( p_t^{G_\theta} \,\|\, p_t^\mathcal{T} \Big) dt = \int_0^T \omega_t \, \mathbb{E}_{p^\mathcal{S}(x) \mathcal{N}(\varepsilon | 0, I)} \log \dfrac{p_t^{G_\theta}(G_\theta(x) + \sigma_t \varepsilon)}{p_t^\mathcal{T}(G_\theta(x) + \sigma_t \varepsilon)} \, dt,
\tag{7}
$$

where $p_t^{G_\theta}$ and $p_t^\mathcal{T}$ represent, respectively, the distribution of the generator output $G_\theta(x)$ and the target distribution $p^\mathcal{T}$, both perturbed by the forward process up to the timestep $t$.

---

[2]Note that there is one more summand, which contains the parametric score $\nabla_\theta \log p_t^{G_\theta}$. However, its expected value is zero (Williams, 1992), and the summand can be omitted.

Optimizing the objective in Equation 7, one obtains a generator, which takes $\boldsymbol{x} \sim p^{\mathcal{S}}$ and outputs $G_\theta(\boldsymbol{x}) \sim p^{\mathcal{T}}$, so it performs the desired transfer between the two distributions. However, there are no guarantees that the input and the output will be related. We fix the issue by explicitly penalizing the input-output transport cost of the generator and obtain the objective

$$\min_\theta \mathcal{L}_\lambda(\theta) = \min_\theta \left[ \mathcal{L}(\theta) + \lambda \, \mathbb{E}_{p^{\mathcal{S}}(\boldsymbol{x})} c\left(\boldsymbol{x}, G_\theta(\boldsymbol{x})\right) \right], \tag{8}$$

where $c(\cdot, \cdot)$ is the cost function, which describes the object properties that we aim to preserve after transfer, and $\lambda$ is the regularization coefficient. Choosing an appropriate $\lambda$ will result in finding a balance between fitting the target distribution and preserving the properties of the input.

As in DMD, we assume that the perturbed target distributions are represented by a pre-trained diffusion model $\boldsymbol{s}_t^{\mathcal{T}}$ and approximate the generator distribution score $\boldsymbol{s}_t^{G_\theta}$ by the additional fake diffusion model $\boldsymbol{s}_t^\phi$. Analogous to the DMD procedure (Equation 5), we perform the coordinate descent in which, however, the generator objective is now regularized. We call the procedure *Regularized Distribution Matching Distillation* (RDMD). Formally, we optimize

$$\begin{cases} \min_\theta \int_0^T \omega_t \, \mathbb{E}_{\varepsilon, \boldsymbol{x}} \log \dfrac{p_t^\phi(G_\theta(\boldsymbol{x}) + \sigma_t \varepsilon)}{p_t^{\mathcal{T}}(G_\theta(\boldsymbol{x}) + \sigma_t \varepsilon)} \, \mathrm{d}t + \lambda \, \mathbb{E}_{p^{\mathcal{S}}(\boldsymbol{x})} c\left(\boldsymbol{x}, G_\theta(\boldsymbol{x})\right); \\ \min_\phi \int_0^T \beta_t \, \mathbb{E}_{\varepsilon, \boldsymbol{x}} \| D_t^\phi(G_\theta(\boldsymbol{x}) + \sigma_t \varepsilon) - G_\theta(\boldsymbol{x}) \|^2 \, \mathrm{d}t. \end{cases} \tag{9}$$

Given the optimal fake score $\boldsymbol{s}_t^\phi$, the generator's objective becomes equal to the desired loss in Equation 8, which validates the procedure.

### 3.2 ANALYSIS OF THE METHOD

The optimization problem in Equation 8 can be seen as the soft-constrained optimal transport, which balances between satisfying the output distribution constraint and preserving the original image properties. If one takes $\lambda \approx 0$, the objective essentially becomes equivalent to the Monge problem (Equation 6). It can be seen by replacing the $\lambda$ coefficient before the transport cost with the $1/\lambda$ coefficient before the KL divergence. For small $\lambda$, it is almost equal to $+\infty$ whenever the generator's output and the target distributions differ, making the corresponding problem hard-constrained and equivalent to the original optimal transport problem. Based on this observation, we prove

**Theorem 3.1.** *Let $c(\boldsymbol{x}, \boldsymbol{y})$ be the quadratic cost [3] $\|\boldsymbol{x} - \boldsymbol{y}\|^2$ and $G^\lambda$ be the theoretical optimum of the objective in Equation 8. Then, under mild regularity conditions, it converges in probability (with respect to $p^{\mathcal{S}}$) to the optimal transport map $G^*$, i.e.*

$$G^\lambda \xrightarrow[\lambda \to 0]{p^{\mathcal{S}}} G^*. \tag{10}$$

The detailed proof can be found in Appendix B. Informally, it means that the optimal transport map can be approximated by the RDMD generator, trained on Equation 9, given a small regularization coefficient, enough capacity of the architecture, and convergence of the optimization algorithm.

It is important to consider this result from a different perspective. It is ideologically similar to the $L_2$ regularization for over-parameterized least squares regression. The original least squares, in this case, have a manifold of solutions. At the same time, by adding $L_2$ weight penalty and taking the limit as the regularization coefficient goes to zero, one obtains a solution with the least norm based on the Moore-Penrose pseudo-inverse. In our case, numerous maps may be optimal in the original DMD procedure, since it only requires matching the distribution at the output. However, training RDMD with $\lambda \approx 0$ results in a feasible solution with almost optimal transport cost.

## 4 EXPERIMENTS

This section presents the experimental results on several unpaired translation tasks. We explore the effect of varying the regularization coefficient $\lambda$ on the learned mappings in a 2D toy setting

---

[3] We prove the theorem only for the quadratic case due to difficulties in analyzing minima of the Monge Problem (Equation 6) in general cases (De Philippis & Figalli, 2014). In practice, however, one can use any cost function of interest.

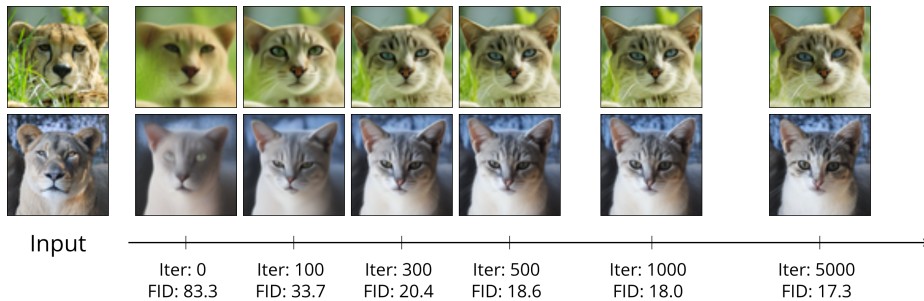

Figure 2: RDMD training dynamics on AFHQv2 *Cat ↔ Wild* translation problem. RDMD achieves strong initialization with meaningful mappings by utilizing the pre-trained target DM. Here, RDMD exhibits rapid convergence to near-optimal performance in 500-1000 training iterations. 500 iterations correspond to approximately 100 minutes of training on $2\times$ NVIDIA A100 GPU.

in Appendix C. In Section 4.1, we extensively compare our method's faithfulness-realism trade-off with the diffusion-based and OT-based baselines on four translation problems in $64 \times 64$ and $128 \times 128$ pixel space. In Section 4.2, we scale our method for latent-space translation between pairs of ImageNet classes. In Section E.7, we verify broader applicability of RDMD on unpaired text detoxification. We choose the transport cost $c(\boldsymbol{x}, \boldsymbol{y}) = \|\boldsymbol{x} - \boldsymbol{y}\|^2$ in image-to-image experiments. The additional training details can be found in Appendix E.

**Initialization** RDMD shares the architecture between all three used networks: the target score, the fake score, and the generator. This setting allows for obtaining strong models' initialization and significantly speeding up convergence. We utilize the pre-trained target score in two ways. First, we initialize the fake model with its copy. Second, we initialize the generator $G_\theta(\boldsymbol{x})$ with the denoiser parameterization $D_{\hat{\sigma}}^{\mathcal{T}}(\boldsymbol{x})$ of the pre-trained target score, but with a fixed $\hat{\sigma} \in [0, T]$ (since the generator is independent of time at input). The denoiser parameterization is trained to denoise images from the target domain. Being an initialization for the generator, the denoiser network $D_{\hat{\sigma}}^{\mathcal{T}}(\boldsymbol{x})$ treats a source object $\boldsymbol{x}$ as the noised target object $\boldsymbol{y} + \hat{\sigma}\varepsilon$ and tries to "denoise" it into the output, realistic for the target domain. It thus tries to generate realistic outputs while preserving high faithfulness, which is crucial for domain translation. This combination of meaningful mappings with strong initialization of weights of all networks allows for the rapid convergence of RDMD. We visualize its training dynamics in Figure 2 and demonstrate it is capable of achieving near-optimal performance in just hundreds of GPU-minutes. We set $\hat{\sigma} = 1.0$ for all experiments except CelebA-128, where $\hat{\sigma} = 3.0$. We explore the choice of $\hat{\sigma}$ in Appendix D.

**Baselines** We compare our method with the three families of baselines. **One-sided DMs** use a single target diffusion model to denoise a perturbed source image (SDEdit, Meng et al. (2021)) or guide sampling by enforcing source closeness (ILVR, Choi et al. (2021)) and classifier-driven domain dissimilarity (EGSDE, Zhao et al. (2022)). **Two-sided DMs** use both source (encoding) and target (decoding) diffusion models, linking them via deterministic ODE sampling (DDIB, Su et al. (2022)) or by replacing target noise with noise predictions from the source process (CycleDiff, Wu & De la Torre (2023)). **OT** methods use discriminator-based training to enforce realism, maintaining faithfulness by utilizing an L2 loss with displacement interpolation (DIOTM, Choi et al. (2024)) or by iteratively refining the underlying Markov process (ASBM, Gushchin et al. (2024b)). We include a complete description of the relevant methods in Appendix A.

## 4.1 I2I IN PIXEL SPACE

Next, we compare the proposed RDMD method with OT-based and diffusion-based baselines on $64 \times 64$ AFHQv2 (Choi et al., 2020) *Cat ↔ Wild* and $128 \times 128$ CelebA (Liu et al., 2015) *Male ↔ Female* translation problems. We do not compare with GAN-based methods since they mostly demonstrate results that are inferior to those of EGSDE (Zhao et al., 2022) in terms of FID and PSNR. We pre-train the target diffusion models with EDM (Karras et al., 2022) parameterization. We use the DDPM++ (Song et al., 2020b) architecture for $64 \times 64$ experiments and ADM (Dhariwal & Nichol, 2021) (with 128 model channels instead of 192) for $128 \times 128$ experiments. The

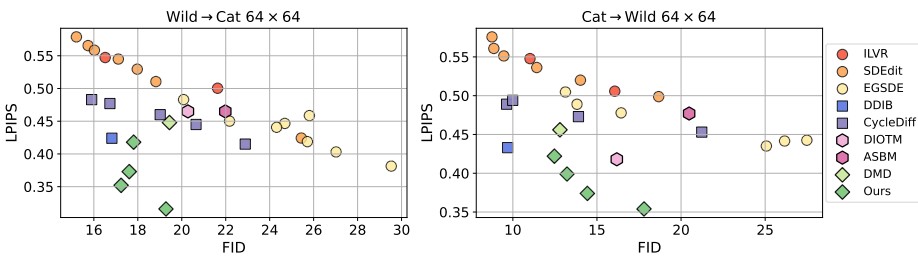

Figure 3: Comparison of RDMD with the baselines on AFHQv2 $64 \times 64$ *Cat* $\leftrightarrow$ *Wild* translation tasks. The figure demonstrates the tradeoff between generation quality (FID↓) and the input-output faithfulness (LPIPS↓).

Table 2: Comparison of RDMD with diffusion and OT-based baselines in pixel space. We mark the best results in **bold** and the best results among few-step methods in ***italic and bold***.



(a) $64 \times 64$ *Cat* $\leftrightarrow$ *Wild*

| Model | Wild→Cat FID | Wild→Cat LPIPS | Cat→Wild FID | Cat→Wild LPIPS | NFE |
|---|---|---|---|---|---|
| ILVR | 21.63 | 0.500 | 16.06 | 0.549 | 35 |
| SDEdit | 17.98 | 0.529 | 14.02 | 0.520 | 35 |
| EGSDE | 20.08 | 0.483 | 13.81 | 0.489 | 35 |
| DDIB | 16.81 | 0.424 | **9.69** | 0.433 | 70 |
| CycleDiff | **16.73** | 0.477 | 13.90 | 0.473 | 140 |
| ASBM | 21.91 | 0.464 | 20.48 | 0.477 | 3 |
| DIOTM | 20.82 | 0.417 | 16.18 | 0.418 | **1** |
| DMD | 19.44 | 0.448 | *12.80* | 0.456 | **1** |
| RDMD | *17.31* | **0.352** | 14.43 | **0.374** | **1** |

(b) $128 \times 128$ *Male* $\leftrightarrow$ *Female*

| Model | Male→Female FID | Male→Female LPIPS | Female→Male FID | Female→Male LPIPS | NFE |
|---|---|---|---|---|---|
| ILVR | 24.42 | 0.503 | 19.21 | 0.514 | 35 |
| SDEdit | 8.85 | 0.530 | 7.72 | 0.533 | 35 |
| EGSDE | 21.90 | 0.464 | 20.03 | 0.472 | 35 |
| DDIB | **5.39** | 0.342 | **3.72** | 0.348 | 70 |
| CycleDiff | 6.82 | 0.327 | 5.11 | 0.335 | 140 |
| ASBM | 15.93 | 0.370 | 26.08 | 0.376 | 3 |
| DIOTM | 9.49 | 0.271 | 10.48 | 0.246 | **1** |
| DMD | 12.58 | 0.333 | 12.66 | 0.330 | **1** |
| RDMD | *9.30* | **0.236** | *6.68* | **0.237** | **1** |



networks have approximately $55M$ and $130M$ parameters, respectively. We slightly adapt the official diffusion baselines' implementations for compatibility with the EDM setting. For each of the diffusion-based baselines, we run a grid of hyperparameters, if applicable. The detailed hyperparameter values can be found in Appendix E.4 and E.5.

**Faithfulness-realism trade-off** In AFHQv2 experiments we focus on comparing the trade-off achieved by our method and the baselines. The quality metric is FID, the faithfulness metric is LPIPS (see Figures 8 and 9 in Appendix F.1 for $L_2$, PSNR and SSIM). In addition, we perform visual comparisons in Figures 12 and 13. We compare our method with the baselines in Figure 3. Specifically, for each method we run a grid of hyperparameters and represent each run with the corresponding point in the plot (see Appendix E for details). We observe that RDMD achieves a better trade-off given moderately strict requirements on faithfulness: all of our models beat the corresponding baselines in the (approximate) LPIPS range $(0.3, 0.4)$ for *Wild* $\rightarrow$ *Cat* and $(0.36, 0.42)$ for *Cat* $\rightarrow$ *Wild*. Here, RDMD also shows strictly better performance than the OT/SB baselines DIOTM and ASBM. If the lower FID is strongly preferable over the transport cost, then it might be better to use one of the diffusion baselines. In this case, DDIB and CycleDiffusion show significantly better faithfulness than one-sided methods.

**Metrics comparison** We further illustrate the observed performance in Table 2 by choosing one RDMD run and comparing it with the baselines' runs with the closest FID (i.e. we compare faithfulness given fixed realism). For all four problems, we beat all the baselines in terms of similarity. In terms of generation quality, DDIB and CycleDiffusion are the only baselines that sometimes achieve noticeably better FID than RDMD at the cost of worse similarity, expensive sampling (2 times more function evaluations than in the diffusion sampling) and requiring pre-trained diffusion models for the source domains. When any of the three limitations becomes a significant concern, RDMD is

Table 3: Comparison of RDMD with OT-based baselines on CelebA $64 \times 64$ with limited data (5k source and target samples). ASBM and DIOTM generate distorted images (Figure 4) and suffer from significant drop in both faithulness and realism.

| | 5k | | Full data | |
|---|---|---|---|---|
| Model | FID | LPIPS | FID | LPIPS |
| ASBM | 43.97 | 0.349 | 23.06 | 0.324 |
| DIOTM | 31.34 | 0.352 | 15.81 | 0.204 |
| RDMD | **20.99** | **0.238** | **10.36** | **0.176** |

Figure 4: Visual comparison of RDMD with OT-based baselines on CelebA $64 \times 64$ with limited (5k source and target samples) and full data .

Table 4: Quantitative comparison of RDMD with two-sided diffusion-based baselines on ImageNet multiclass translation benchmarks. DDIB and CycleDiff perform 100 and 80 encoding-decoding steps, respectively. This number is multiplied by 3 due to the usage of **cfg** during decoding.

| | *Animals* | | *Birds* | | *Fish* | | *Insects* | | |
|---|---|---|---|---|---|---|---|---|---|
| Model | FID | LPIPS | FID | LPIPS | FID | LPIPS | FID | LPIPS | NFE |
| DDIB | **25.99** | 0.457 | **17.68** | 0.505 | 27.04 | 0.478 | 23.32 | 0.454 | 300 + 2 |
| CycleDiff | 30.42 | 0.460 | 18.08 | 0.523 | **24.96** | 0.464 | **20.74** | 0.412 | 240 + 2 |
| RDMD | 39.85 | **0.369** | 24.87 | **0.415** | 34.00 | **0.329** | 29.57 | **0.296** | **1 + 2** |

generally the preferred method. It is also worth mentioning that RDMD (or DMD in case of *Wild $\to$ Cat*) achieves the best FID among the one-step baselines, which we mark in ***italic and bold***. Additionally, in Figures 10 and 11 we visualize faithfulness-realism trade-off achieved by our method and the baselines on *Male $\leftrightarrow$ Female* translation problems.

**Data efficiency** We further highlight the advantages of RDMD over the existing adversarial-based OT methods by demonstrating that they perform poorly in problems with limited data. To this end, we compare RDMD with DIOTM and ASBM on CelebA $64 \times 64$ *Male $\to$ Female* translation task with only $5k$ random samples for source and target data sets (instead of the original $\approx 200k$ samples in total). In Table 3 and Figure 4 we demonstrate that both baselines start to produce distorted and unrealistic images, while RDMD generates blurrier, but still relatively faithful and realistic samples.

### 4.2 LATENT-SPACE MULTICLASS IMAGENET TRANSLATION

We scale our method and apply it to a more challenging scenario of translating between pairs of ImageNet (Deng et al., 2009) classes with a single class-conditional model. To this and, we take $256 \times 256$ class-conditional LDM (Rombach et al., 2022) as the pre-trained target score and use it as initialization for both the generator and the fake score. We train **one model** for translation between all pairs of ImageNet classes. We describe the setup in details in Appendix E.6.

We validate performance of the obtained model by constructing several benchmarks: *Animals, Birds, Fish, and Insects*. In each benchmark we choose 5 related classes and translate 50 test set pictures of each into all other classes, resulting in total of $50 \times 5 = 250$ inputs and $250 \times 4 = 1000$ outputs per benchmark. We measure FID (reference statistics correspond to the 5 benchmark classes from ImageNet training set) and LPIPS and compare with the two-sided diffusion methods DDIB and CycleDiffusion. Here, RDMD significantly outperforms the baselines in terms of faithfulness. At the same time, its higher FID may be explained by the visual comparison in Figure 5. Here, RDMD acts more as an image editing model: it detects only the source object and transforms it into the target, which may result in an unrealistic environment for the target class. We stress, however, that this is a desirable property, which is not demonstrated by DDIB and CycleDiffusion. We additionally verify RDMD's effectiveness beyond similar classes by performing out-of-domain translation in Figures 16, 17, 18, 19, 20 in Appendix F.3.

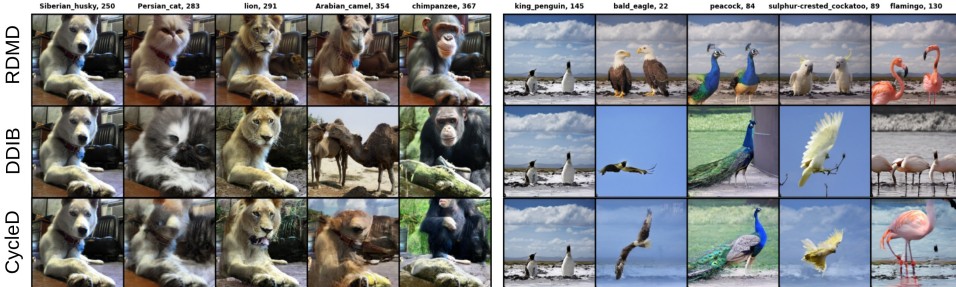

Figure 5: Visual comparison of RDMD with two-sided diffusion baselines on ImageNet translation benchmarks: *Animals*, and *Birds*.

Table 5: Performance of RDMD, two-sided diffusion baselines (CycleDiff and DDIB) and the **paired** (marked by †) translation model Cosmos on text detoxification (ParaDetox).

| Model | ppl ($\downarrow$) | BLEU ($\uparrow$) | Style Acc. ($\uparrow$) | Similarity ($\uparrow$) | Fluency ($\uparrow$) | J-score ($\uparrow$) | NFE |
|---|---|---|---|---|---|---|---|
| Cosmos† | 262.1 | **0.694** | **0.904** | 0.815 | **0.753** | **0.554** | 200 |
| DDIB | 564.9 | 0.537 | 0.661 | 0.758 | 0.436 | 0.244 | 320 |
| CycleDiff | 298.2 | 0.611 | 0.536 | **0.856** | 0.684 | 0.326 | 320 |
| RDMD | **254.8** | *0.665* | *0.864* | 0.837 | *0.736* | *0.537* | **1** |

### 4.3 TEXT DETOXIFICATION

To demonstrate the versatility of RDMD beyond computer vision, we apply our method to the natural language processing task of text detoxification. This task can be framed as an unpaired text-to-text translation problem, where the goal is to paraphrase a toxic text into a neutral one while preserving its original meaning and fluency. For our experiments, we use the ParaDetox dataset (Logacheva et al., 2022). A complete description of the setup is given in Appendix E.7. The results on the text detoxification problem can be seen in Table 5. RDMD significantly outperforms the unpaired baselines and even achieves results comparable to the **paired** Cosmos† (Meshchaninov et al., 2025) model while being unpaired and requiring less than 1% of their inference steps.

## 5 DISCUSSION AND LIMITATIONS

In this paper, we propose RDMD, the novel *one-step* diffusion-based algorithm for the unpaired translation. This algorithm replaces the adversarial loss, prominent in the OT-based approaches, with the diffusion-based distribution matching. The algorithm has efficient one-step inference, explicit control over faithfulness, strong initialization and fast convergence.

From the theoretical standpoint, we prove that at low regularization coefficients, the theoretical optimum of the introduced objective is close to the optimal transport map (Theorem 3.1). In Section 4.1 we compare our method with the OT and diffusion-based baselines in image-to-image experiments. We show that our model achieves strong faithfulness-realism trade-off, exhibits fast convergence, and has low data requirements. In Section 4.2 we showcase the image editing capabilities of our method in the latent space on a challenging multiclass translation problem. In Section 4.3 we demonstrate the capabilities of RDMD beyond computer vision on the text detoxification problem, where it shows superior results in comparison to other unpaired diffusion methods.

In terms of limitations, we admit that our theory works in the asymptotic regime, while one could derive more precise non-limit bounds. Our experimental results are limited in terms of achieving the lowest baselines' FID values (e.g. in Male→Female experiment we achieve 9.3, while one of the multi-step baselines, DDIB, achieves 5.39). We see making few-step modification as a potential way to mitigate this difference. Furthermore, the desired feature of the method would be switching among different regularization coefficients without re-training. Potential impacts include further development and acceleration of unpaired translation models.

## REPRODUCIBILITY STATEMENT

To ensure clarity and reproducibility of our work, we provide excessive description of our method. All experimental details, including batch sizes, optimizer choice, model architectures, and specific hyperparameter configurations are thoroughly documented in Appendix E. Furthermore, our experiments are built upon publicly available datasets (e.g. AFHQv2, CelebA, ImageNet) to ensure our experimental setups are accessible and verifiable.

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

## A  RELATED WORK

**GANs** were the prevalent paradigm in the unpaired tranlsation (unpaired image-to-image, I2I, in particular) for a long time. Among other methods, CycleGAN (Zhu et al., 2017), DualGAN (Yi et al., 2017), and DiscoGAN (Kim et al., 2017) pioneered the utilization of the cycle-consistency paradigm with the adversarial loss. It gave rise to many two-sided methods, including UNIT (Liu et al., 2017) and MUNIT (Huang et al., 2018) that divide the encoding into style-space and content-space, and SCAN (Li et al., 2018) that splits the procedure into coarse and fine stages. **The one-sided** GAN-based methods tackle inpaired tranlsation without learning the inverse for better computational efficiency. DistanceGAN (Benaim & Wolf, 2017) achieves it by learning to preserve the distance between pairs of samples, GCGAN (Fu et al., 2019) imposes geometrical consistency constraints, and CUT (Park et al., 2020) uses the contrastive loss to maximize the patch-wise mutual information between input and output.

**Diffusion-based** unpaired translation models modify the diffusion process using the source image. SDEdit (Meng et al., 2021) initializes the reverse diffusion process for the target distribution with the noisy source picture instead of pure noise to maintain similarity. Many methods guide (Ho & Salimans, 2022; Epstein et al., 2023) the target diffusion process. ILVR (Choi et al., 2021) adds the correction that enforces the current noisy sample to resemble the source. EGSDE (Zhao et al., 2022) combines the idea of ILVR with training a classifier between domains and encouraging dissimilarity between the corresponding embeddings to distinguish between the domains. The other diffusion-based approaches include two-sided methods based on the concatenation of two diffusion models (DDIB (Su et al., 2022) and CycleDiff (Wu & De la Torre, 2023)).

**Optimal transport** (Villani et al., 2009; Peyré et al., 2019) is another useful framework for the unpaired translation. Methods based on it reformulate the OT problem (Eq. 6) and its modifications as Entropic OT (EOT) (Cuturi, 2013) or Schrödinger Bridge (SB) (Föllmer, 1988) to be accessible in practice. In particular, OTM (Fan et al., 2021) and NOT (Korotin et al., 2022) use the Lagrangian multipliers formulation of the distribution matching constraint, which results in adversarial training. DIOTM (Choi et al., 2024) builds on top of this idea by utilizing the displacement interpolation formula for the dynamic OT problem and forcing satisfaction of the Hamilton-Jacobi-Bellman equation. ENOT (Gushchin et al., 2024a) and UNSB (Kim et al., 2023a) utilize similar observations for tackling the Enropic OT problem.

The other methods obtain (partially) simulation-free techniques by iteratively refining the stochastic process between two distributions. De Bortoli et al. (2021); Vargas et al. (2021) define this refinement as learning of the time-reversal with the corresponding initial distribution (source or target). Other methods build on Flow (Lipman et al., 2022; Tong et al., 2023; Albergo & Vanden-Eijnden, 2022) and Bridge (Somnath et al., 2023; Peluchetti, 2023) Matching and their sequential reiteration (Liu et al., 2022; 2023; Shi et al., 2024). DSBM (Shi et al., 2024) reiterates the Bridge Matching, while ASBM (Gushchin et al., 2024b) improves its computational efficiency by considering its discrete-time counterpart.

**Diffusion distillation** techniques are mainly divided into two families. **First** family of methods uses the pre-trained diffusion model as a (multi-step) noise$\rightarrow$ image mapper and learns it. This includes optimizing the regression loss between the outputs (Salimans & Ho, 2022) or learning the integrator of the corresponding ODE (Gu et al., 2023; Song et al., 2023; Kim et al., 2023b), including ODEs with guidance (Meng et al., 2023). **Second** family of methods considers diffusion models as a source of "knowledge" that can push an arbitrary model toward matching the distributional constraint. It is commonly formalized as optimizing the Integrated KL divergence (Luo et al., 2024; Yin et al., 2023; 2024; Nguyen & Tran, 2023) by training an additional "fake" diffusion model on the generator's output distribution. Other methods consider matching scores (Zhou et al., 2024) or moments (Salimans et al., 2024) of the corresponding distributions. Notably, these methods do not have any specific restrictions on the model structure, which allows their wide usage (e.g., in text-to-3D (Poole et al., 2022; Wang et al., 2024)). Importantly, it allows us to push the generator towards the target distribution in the unpaired translation setting, combined with the transport cost regularization.

## B  THEORY

In this section, we aim at proving the main theoretical result of the work: solution of the soft-constrained RDMD objective converges to the solution of the hard-constrained Monge problem. Our proof is largely based on the work of Liero et al. (2018). It introduces the family of entropy-transport problems, consisting in optimizing the transport cost with soft constraints based on the divergence between the map's output distribution and the target. There are, however, differences between the problems, that prevent us from reducing the functional in Eq. 8 to the entropy-transport problems. First, authors consider the case of finite non-negative measures, while we stick to the probability distributions. Second, the family of Csiszár $f$-divergences (Csiszár, 1967), used by Liero et al. (2018), seemingly does not contain the integral ensemble of KL divergences, used in Eq. 8. Finally, we illustrate the proof in a simpler particular setting for the narrative purposes. Nevertheless, the used ideas are very similar.

### B.1  PROOF OUTLINE

We start by giving a simple outline of the proof. Given a pair of source and target distributions $p^{\mathcal{S}}$ and $p^{\mathcal{T}}$, RDMD optimizes the following functional with respect to the generator $G$:

$$\int\limits_0^T \omega_t \, \mathrm{KL}\left(p_t^G \,\|\, p_t^{\mathcal{T}}\right) \mathrm{d}t + \lambda \, \mathbb{E}_{p^{\mathcal{S}}(\boldsymbol{x})} c\left(\boldsymbol{x}, G(\boldsymbol{x})\right), \tag{11}$$

where $p_t^G$ and $p_t^{\mathcal{T}}$ are the generator distribution $p^G$ and the target distribution $p^{\mathcal{T}}$, perturbed by the forward diffusion process up to the time step $t$. Our goal is to prove that the optimal generator of the regularized objective converges to the optimal transport map when $\lambda \to 0$. With a slight abuse of notation, in this section we will use a different objective

$$\mathcal{L}^\alpha(G) = \alpha \int\limits_0^T \omega_t \, \mathrm{KL}\left(p_t^G \,\|\, p_t^{\mathcal{T}}\right) \mathrm{d}t + \mathbb{E}_{p^{\mathcal{S}}(\boldsymbol{x})} c\left(\boldsymbol{x}, G(\boldsymbol{x})\right) \tag{12}$$

and consider the equivalent limit $\alpha \to +\infty$. We also define

$$\mathcal{L}^\infty(G) = \begin{cases} \mathbb{E}_{p^{\mathcal{S}}(\boldsymbol{x})} c\left(\boldsymbol{x}, G(\boldsymbol{x})\right), \text{ if } p^G = p^{\mathcal{T}}; \\ +\infty, \text{ else} \end{cases} \tag{13}$$

to be the objective, corresponding to the unconditional formulation of the Monge problem (Eq. 6). In this section, we will denote minimum of this objective (which is, therefore, the optimal transport map) as $G^\infty$ [4]

We first assume that the infimum of the objective $\mathcal{L}^\alpha$ is reached and define $G^\alpha$ be the optimal generator. We denote by $\{\alpha_n\}_{n=1}^{+\infty}$ an arbitrary sequence with $\alpha_n \to +\infty$. We first make two informal assumptions that need to be proved (and will be in some sence further in the section):

1. The sequence $G^{\alpha_n}$ converges (in some sence) to some function $\hat{G}$;

2. $\mathcal{L}^\alpha$ is continuous with respect to this convergence, i.e. for every convergent sequence $G_n \to G$ holds $\mathcal{L}^\alpha(G_n) \to \mathcal{L}^\alpha(G)$.

Given this, we first observe that for each map $G$ the sequence of objectives $\mathcal{L}^{\alpha_n}(G)$ monotonically converges to the objective $\mathcal{L}^\infty(G)$. It follows from the fact that the first summand of $\mathcal{L}^{\alpha_n}$ converges to $+\infty$ if and only if the KL divergence is non-zero, which is equivalent to saying that $p^G$ and $p^{\mathcal{T}}$ differ (Wang et al., 2024). If instead $p^G = p^{\mathcal{T}}$, the summand zeroes out. This also means that the minimal values of the corresponding objectives form a monotonic sequence:

$$\mathcal{L}^{\alpha_n}(G^{\alpha_n}) \leq \mathcal{L}^{\alpha_{n+1}}(G^{\alpha_{n+1}}) \leq \mathcal{L}^\infty(G^\infty). \tag{14}$$

---

[4]Solution to the Monge problem is not always unique, but we will further impose assumptions that will guarantee the uniqueness.

Finally, the monotonicity implies that for a fixed $m$

$$\lim_{n \to \infty} \mathcal{L}^{\alpha_n}(G^{\alpha_n}) \geq \lim_{n \to \infty} \mathcal{L}^{\alpha_m}(G^{\alpha_n}), \tag{15}$$

since the input $G^{\alpha_n}$ is fixed and $\mathcal{L}^{\alpha_n}$ monotonically increases. Using the assumed continuity of the objective, we obtain

$$\lim_{n \to \infty} \mathcal{L}^{\alpha_n}(G^{\alpha_n}) \geq \mathcal{L}^{\alpha_m}(\hat{G}) \tag{16}$$

for each $m$. Taking the limit $m \to \infty$, we obtain

$$\lim_{n \to \infty} \mathcal{L}^{\alpha_n}(G^{\alpha_n}) \geq \mathcal{L}^{\infty}(\hat{G}). \tag{17}$$

Combining this set of equations, we obtain:

$$\mathcal{L}^{\infty}(G^{\infty}) \geq \lim_{n \to \infty} \mathcal{L}^{\alpha_n}(G^{\alpha_n}) \geq \mathcal{L}^{\infty}(\hat{G}) \geq \mathcal{L}^{\infty}(G^{\infty}), \tag{18}$$

where the first inequality comes from the monotonicity of the minimal values and the last inequality uses that $G^{\infty}$ is the minimum of the objective $\mathcal{L}^{\infty}$. Hence, that limiting map $\hat{G}$ achieves minimal value of the objective $\mathcal{L}^{\infty}$ and is, therefore, the optimal transport map.

At this point, we only need to define and prove some versions of the aforementioned facts:

1. Infimum of $\mathcal{L}^{\alpha}$ is reached;
2. The sequence of minima $G^{\alpha_n}$ converges;
3. $\mathcal{L}^{\alpha}$ is continuous with respect to this convergence.

From now on, we formulate the result in details and stick to the formal proof.

### B.2 Assumptions and theorem statement

First, we list the assumptions.

**Assumption B.1.** The distributions $p^{\mathcal{S}}$ and $p^{\mathcal{T}}$ have densities with respect to the Lebesgue measure. The distributions are defined on open bounded subsets $\mathcal{X} \subset \mathbb{R}^d$ and $\mathcal{Y} \subset \mathbb{R}^d$, where $\mathcal{Y}$ is convex. The densities are bounded away from zero and infinity on $\mathcal{X}$ and $\mathcal{Y}$, respectively.

We admit that boundedness of the support is a very restrictive assumption from the theoretical standpoint, however in our applications (I2I) both source and target distributions are supported on the bounded space of images. We thus can set $\mathcal{X} = \mathcal{Y} = (0, 1)^d$.

**Assumption B.2.** The cost $c(\boldsymbol{x}, \boldsymbol{y})$ is quadratic $\|\boldsymbol{x} - \boldsymbol{y}\|^2$.

Here, we stick to proving the theorem only for $L_2$ cost due to difficulties in investigation of Monge map existence and regularity for general transport costs (De Philippis & Figalli, 2014).

**Assumption B.3.** The weighting function $\omega_t$ is positive and bounded.

**Assumption B.4.** Standard deviation $\sigma_t$ of the noise, defined by the forward process, is continuous in $t$.

**Theorem B.1.** *Let $p^{\mathcal{S}}, p^{\mathcal{T}}, c, \omega_t$, and $\sigma_t$ satisfy the assumptions **1-3**. Then, there exists a minimum $G^{\alpha}$ of the objective $\mathcal{L}^{\alpha}$ from the Eq. 12. If $\alpha_n \to \infty$, the sequence $G^{\alpha_n}$ converges in probability (with respect to the source distribution) to the optimal transport map $G^{\infty}$:*

$$G^{\alpha_n} \xrightarrow[n \to \infty]{p^{\mathcal{S}}} G^{\infty}. \tag{19}$$

### B.3 Theoretical background

We start by listing all the results necessary for the proof. They are mostly related to the topics of measure theory (weak convergence, in particular) and optimal transport. Most of these classic facts can be found in the books (Bogachev & Ruas, 2007; Dudley, 2018). Otherwise, we make the corresponding citations.

**Definition B.2.** A sequence of probability distributions $p^n(\boldsymbol{x})$ converges weakly to the distribution $p(\boldsymbol{x})$ if for all continuous bounded test functions $\varphi \in \mathcal{C}_b(\mathbb{R}^d)$ holds

$$\mathbb{E}_{p^n(\boldsymbol{x})}\varphi(\boldsymbol{x}) \xrightarrow[n\to\infty]{} \mathbb{E}_{p(\boldsymbol{x})}\varphi(\boldsymbol{x}). \tag{20}$$

Notation: $p^n \xrightarrow{w} p$.

**Definition B.3.** A function $f : \mathbb{R}^d \to \mathbb{R}$ is called lower semi-continuous (lsc), if for all $\boldsymbol{x}_n \to \boldsymbol{x}$ holds

$$\liminf_{n\to\infty} f(\boldsymbol{x}_n) \geq f(\boldsymbol{x}). \tag{21}$$

**Theorem B.4** (Portmanteau/Alexandrov). *$p^n \xrightarrow{w} p$ is equivalent to the following statement: for every lsc function $f$, bounded from below, holds*

$$\liminf_{n\to\infty} \mathbb{E}_{p^n(\boldsymbol{x})} f(\boldsymbol{x}) \geq \mathbb{E}_{p(\boldsymbol{x})} f(\boldsymbol{x}). \tag{22}$$

**Definition B.5.** A sequence of probability measures $p^n$ is called relatively compact, if for every subsequence $p^{n_k}$ there exists a weakly convergent subsequece $p^{n_{k_j}}$.

**Definition B.6.** A sequence of probability measures $p^n$ is called tight, if for every $\varepsilon > 0$ there exists a compact set $K_\varepsilon$ such that $p^n(K_\varepsilon) \geq 1 - \varepsilon$ for all $n$.

**Theorem B.7.** *(Prokhorov) A sequence of probability measures $p^n$ is relatively compact if and only if it is tight. In particular, every weakly convergent sequence is tight.*

**Corollary B.8.** *If there exists a function $\varphi(\boldsymbol{x})$ such that its sublevels $\{\boldsymbol{x} : \varphi(x) \leq r\}$ are compact and for all $n$*

$$\mathbb{E}_{p^n(\boldsymbol{x})}\varphi(x) \leq C$$

*holds with some constant $C$, then $p^n$ is tight.*

**Corollary B.9.** *If a sequence $p^n$ is tight and all of its weakly convergent subsequences converge to the same measure $p$, then $p^n \xrightarrow{w} p$.*

**Definition B.10.** The functional $\mathcal{L}(p)$ is called lower semi-continuous (lsc) with respect to the weak convergence if for all weakly convergent sequences $p^n \xrightarrow{w} p$ holds

$$\liminf_{n\to\infty} \mathcal{L}(p^n) \geq \mathcal{L}(p). \tag{23}$$

**Theorem B.11** (Posner (1975)). *The KL divergence $\mathrm{KL}(p \,\|\, q)$ is lsc (in sense of weak convergence) with respect to each argument, i.e. if $p^n \xrightarrow{w} p$ and $q^n \xrightarrow{w} q$, then*

$$\liminf_{n\to\infty} \mathrm{KL}(p^n \,\|\, q) \geq \mathrm{KL}(p \,\|\, q) \tag{24}$$

$$\liminf_{n\to\infty} \mathrm{KL}(p \,\|\, q^n) \geq \mathrm{KL}(p \,\|\, q). \tag{25}$$

**Theorem B.12** ( Donsker & Varadhan (1983)). *The KL divergence can be expressed as*

$$\mathrm{KL}(p\|q) = \sup_g \left( \mathbb{E}_{p(\boldsymbol{x})} g(\boldsymbol{x}) - \log \mathbb{E}_{q(\boldsymbol{x})} e^{g(\boldsymbol{x})} \right). \tag{26}$$

**Definition B.13.** The expression

$$\mathbb{E}_{p(\boldsymbol{x})} e^{i\langle s, \boldsymbol{x}\rangle} \tag{27}$$

is called the characteristic function (Fourier transform) of the distribution $p(\boldsymbol{x})$.

**Theorem B.14** (Lévy). *Weak convergence of probability measures $p^n \xrightarrow{w} p$ is equivalent to the point-wise convergence of characteristic functions, i.e. $\mathbb{E}_{p^n(\boldsymbol{x})} e^{i\langle s, \boldsymbol{x}\rangle} \to \mathbb{E}_{p(\boldsymbol{x})} e^{i\langle s, \boldsymbol{x}\rangle}$ for all $s$.*

**Definition B.15.** A sequence of measurable functions $\varphi^n(\boldsymbol{x})$ is said to converge in measure (in probability) to the function $\varphi$ with respect to the measure $p(\boldsymbol{x})$, if for all $\varepsilon > 0$ holds

$$p\left(\{\boldsymbol{x} : |\varphi^n(\boldsymbol{x}) - \varphi(\boldsymbol{x})| > \varepsilon\}\right) \to 0.$$

**Theorem B.16** (Lebesgue). *Let $\varphi^n, \varphi$ be measurable functions such that $\|\varphi^n(\boldsymbol{x})\|, \|\varphi(\boldsymbol{x})\| \leq C$ and $\varphi^n(\boldsymbol{x}) \to \varphi(\boldsymbol{x})$ pointwise. Then $\mathbb{E}_{p(\boldsymbol{x})} \varphi^n(\boldsymbol{x}) \to \mathbb{E}_{p(\boldsymbol{x})} \varphi(\boldsymbol{x})$.*

**Lemma B.17** (Fatou)**.** *For any sequence of measurable functions $\varphi^n$ the function $\liminf_n \varphi^n$ is measurable and*

$$\int_a^b \liminf_{n\to\infty} \varphi^n(\boldsymbol{x})\mathrm{d}\boldsymbol{x} \le \liminf_{n\to\infty} \int_a^b \varphi^n(\boldsymbol{x})\mathrm{d}\boldsymbol{x}. \tag{28}$$

**Theorem B.18** ( Brenier (1991))**.** *Given the Assumption B.1, there exists a unique optimal transport map that solves the Monge problem 6 for the quadratic cost.*

*Proof.* This result can be found e.g. in (De Philippis & Figalli, 2014, Theorem 3.1). □

**Theorem B.19.** *Given the Assumption B.1, the unique OT Monge map is continuous.*

*Proof.* This is a simplified version of (De Philippis & Figalli, 2014, Theorem 3.3). □

### B.4 LOWER SEMI-CONTINUITY OF THE LOSS

Having defined all the needed terms and results, we start the proof by re-defining the objective in Eq. 12 with respect to the joint distribution $\pi$ input and output of the generator instead of the generator $G$ itself. Analogous to the Kantorovitch formulation of the optimal transport problem (Kantorovitch, 1958), for each measure $\pi$ on $\mathbb{R}^d \times \mathbb{R}^d$ (which is also called a *transport plan* or just plan) we define the corresponding fuctional as

$$\mathcal{L}^\alpha(\pi) = \alpha \int_0^T \omega_t \, \mathrm{KL}\left(\pi_{\boldsymbol{y},t} \,\|\, p_t^{\mathcal{T}}\right) \mathrm{d}t + \mathbb{E}_{\pi(\boldsymbol{x},\boldsymbol{y})} c\left(\boldsymbol{x}, \boldsymbol{y}\right), \tag{29}$$

where $\pi_{\boldsymbol{x}}$ and $\pi_{\boldsymbol{y}}$ are the corresponding projections (marginal distributions) of $\pi$ and $\pi_{\boldsymbol{y},t}$ is the perturbed $\boldsymbol{y}$-marginal distribution of $\pi$. Note that for $\pi$, corresponding to the joint distribution of $(\boldsymbol{x}, G(\boldsymbol{x}))$, $\mathcal{L}^\alpha(\pi)$ coincides with $\mathcal{L}^\alpha(G)$, defined in Eq. 12. Thus, we aim to optimize $\mathcal{L}^\alpha(\pi)$ with respect to such plans $\pi$, that their $\boldsymbol{x}$ marginal is equal to $p^{\mathcal{S}}$ and $\pi(\boldsymbol{y} = G(\boldsymbol{x})) = 1$ for some $G$.

**Definition B.20.** We will call a measure $\pi$ generator-based if its $\boldsymbol{x}$-marginal is equal to $p^{\mathcal{S}}$ and $\pi(\boldsymbol{y} = G(\boldsymbol{x}))$ for some function $G$.

For the sake of clearity, we note that the distributions $\pi_t^{\boldsymbol{y}}$ and $p_t^{\mathcal{T}}$ can be represented as $\pi^{\boldsymbol{y}} * q_t$ and $p^{\mathcal{T}} * q_t$, where $*$ is the convolution operation and $q_t = \mathcal{N}(0, \sigma_t^2 I)$. We thus rewrite the functional as

$$\mathcal{L}^\alpha(\pi) = \alpha \int_0^T \omega_t \, \mathrm{KL}\left(\pi_{\boldsymbol{y}} * q_t \,\|\, p^{\mathcal{T}} * q_t\right) \mathrm{d}t + \mathbb{E}_{\pi(\boldsymbol{x},\boldsymbol{y})} c\left(\boldsymbol{x}, \boldsymbol{y}\right), \tag{30}$$

Previously, we wanted to establish continuity of the objective. This may not be the case in general. Instead, we prove the following

**Lemma B.21.** $\mathcal{L}^\alpha(\pi)$ *is lsc with respect to the weak convergence, i.e. for all weakly convergent sequences $\pi^n \xrightarrow{w} \pi$ holds*

$$\liminf_{n\to\infty} \mathcal{L}^\alpha(\pi^n) \ge \mathcal{L}^\alpha(\pi). \tag{31}$$

This result is a direct consequence of the Theorem B.11 about lower semi-continuity of the KL divergence.

*Proof.* We start by proving that the projection and the convolution operation preserve weak convergence. For the first, we need to prove that for any test function $g \in \mathcal{C}_b(\mathbb{R}^d)$ holds

$$\mathbb{E}_{\pi_{\boldsymbol{y}}^n(\boldsymbol{y})} g(\boldsymbol{y}) \to \mathbb{E}_{\pi_{\boldsymbol{y}}(\boldsymbol{y})} g(\boldsymbol{y}) \tag{32}$$

given $\pi^n \xrightarrow{w} \pi$. For this, we note that the function $\varphi(\boldsymbol{x}, \boldsymbol{y}) = g(\boldsymbol{y})$ is also bounded and continuous and, thus

$$\mathbb{E}_{\pi_{\boldsymbol{y}}^n(\boldsymbol{y})} g(\boldsymbol{y}) = \mathbb{E}_{\pi^n(\boldsymbol{x},\boldsymbol{y})} \varphi(\boldsymbol{x}, \boldsymbol{y}) \to \mathbb{E}_{\pi(\boldsymbol{x},\boldsymbol{y})} \varphi(\boldsymbol{x}, \boldsymbol{y}) = \mathbb{E}_{\pi_{\boldsymbol{y}}(\boldsymbol{y})} g(\boldsymbol{y}). \tag{33}$$

Regarding the convolution, recall that $\pi_{\boldsymbol{y}}^n * q_t$ is the distribution of the sum of independent variables with corresponding distributions. Its characteristic function is equal to

$$\mathbb{E}_{\pi_{\boldsymbol{y}}^n * q_t(\boldsymbol{y}_t)} e^{i\langle s, \boldsymbol{y}_t \rangle} = \mathbb{E}_{\pi_{\boldsymbol{y}}^n(\boldsymbol{y}) q_t(\varepsilon_t)} e^{i\langle s, \boldsymbol{y} + \varepsilon_t \rangle} = \mathbb{E}_{\pi_{\boldsymbol{y}}^n(\boldsymbol{y})} e^{i\langle s, \boldsymbol{y} \rangle} \mathbb{E}_{q_t(\varepsilon_t)} e^{i\langle s, \varepsilon_t \rangle}. \tag{34}$$

Applying the Lévy's continuity theorem to $\pi_{\boldsymbol{y}}^n \xrightarrow{w} \pi_{\boldsymbol{y}}$, we take the limit and obtain

$$\mathbb{E}_{\pi_{\boldsymbol{y}}(\boldsymbol{y})} e^{i\langle s, \boldsymbol{y} \rangle} \mathbb{E}_{q_t(\varepsilon_t)} e^{i\langle s, \varepsilon_t \rangle} = \mathbb{E}_{\pi_{\boldsymbol{y}}(\boldsymbol{y}) q_t(\varepsilon_t)} e^{i\langle s, \boldsymbol{y} + \varepsilon_t \rangle} = \mathbb{E}_{\pi_{\boldsymbol{y}} * q_t(\boldsymbol{y}_t)} e^{i\langle s, \boldsymbol{y}_t \rangle}, \tag{35}$$

which implies

$$\mathbb{E}_{\pi_{\boldsymbol{y}}^n * q_t(\boldsymbol{y}_t)} e^{i\langle s, \boldsymbol{y}_t \rangle} \to \mathbb{E}_{\pi_{\boldsymbol{y}} * q_t(\boldsymbol{y}_t)} e^{i\langle s, \boldsymbol{y}_t \rangle}. \tag{36}$$

We apply the continuity theorem for the convolutions and obtain $\pi_{\boldsymbol{y}}^n * q_t \xrightarrow{w} \pi_{\boldsymbol{y}} * q_t$.

With this observation, we prove that the first term of $\mathcal{L}^\alpha(\pi)$ is lsc. First, we apply Lemma B.17 (Fatou) and move the limit inside the integral

$$\liminf_{n \to \infty} \int_0^T \omega_t \, \mathrm{KL}\left(\pi_{\boldsymbol{y}}^n * q_t \,\|\, p^{\mathcal{T}} * q_t\right) \mathrm{d}t \geq \int_0^T \liminf_{n \to \infty} \omega_t \, \mathrm{KL}\left(\pi_{\boldsymbol{y}}^n * q_t \,\|\, p^{\mathcal{T}} * q_t\right) \mathrm{d}t. \tag{37}$$

Using the lower semi-continuity of the KL divergence (Theorem B.11), we obtain

$$\int_0^T \liminf_{n \to \infty} \omega_t \, \mathrm{KL}\left(\pi_{\boldsymbol{y}}^n * q_t \,\|\, p^{\mathcal{T}} * q_t\right) \mathrm{d}t \geq \int_0^T \omega_t \, \mathrm{KL}\left(\pi_{\boldsymbol{y}} * q_t \,\|\, p^{\mathcal{T}} * q_t\right) \mathrm{d}t. \tag{38}$$

Finally, the Assumption B.2 on the continuity of $c(\cdot, \cdot)$ implies its lower semi-coninuity. Theorem B.4 (Portmanteau) states that

$$\liminf_{n \to \infty} \mathbb{E}_{\pi^n(\boldsymbol{x}, \boldsymbol{y})} c(\boldsymbol{x}, \boldsymbol{y}) \geq \mathbb{E}_{\pi(\boldsymbol{x}, \boldsymbol{y})} c(\boldsymbol{x}, \boldsymbol{y}). \tag{39}$$

Combining inequalities from Eq. 37, Eq. 38 and Eq. 39, we obtain

$$\liminf_{n \to \infty} \mathcal{L}^\alpha(\pi^n) \geq \mathcal{L}^\alpha(\pi). \tag{40}$$

$\square$

## B.5 EXISTENCE OF THE MINIMIZER

Now we aim to prove that the objective $\mathcal{L}^\alpha(\pi)$ has a minimum over generator-based plans. First, we need the following technical lemma about sublevels of the KL part of the functional.

**Lemma B.22.** *Let $\{\pi^n\}_{n=1}^\infty$ be a sequence of generator-based plans that satisfy*

$$\int_0^T \omega_t \, \mathrm{KL}\left(\pi_{\boldsymbol{y}, t}^n \,\|\, p_t^{\mathcal{T}}\right) \mathrm{d}t \leq C \tag{41}$$

*for some constant $C$. Then, the sequence $\{\pi^n\}_{n=1}^\infty$ is tight.*

*Proof.* We take arbitrary $\pi$ from the sequence and apply the Donsker-Varadhan representation (Theorem B.12) of the KL divergence. We take the test function $g(\boldsymbol{x}) = \|x\|^2 / (2\sigma_T^2)$ and obtain

$$\int_0^T \omega_t \, \mathrm{KL}\left(\pi_{\boldsymbol{y}, t} \,\|\, p_t^{\mathcal{T}}\right) \mathrm{d}t \geq \int_0^T \omega_t \left(\mathbb{E}_{\pi_{\boldsymbol{y}, t}(\boldsymbol{y}_t)} \frac{1}{2\sigma_T^2} \|\boldsymbol{y}_t\|^2 - \log \mathbb{E}_{p_t^{\mathcal{T}}(\boldsymbol{y}_t)} e^{\|\boldsymbol{y}_t\|^2 / (2\sigma_T^2)}\right) \mathrm{d}t. \tag{42}$$

The choice of $g(\boldsymbol{x})$ is not very specific, i.e. every function that will produce finite expectations and integrals is suitable. In the right-hand side, we rewrite the expectations with repect to the original variable and noise:

$$\int_0^T \omega_t \left(\mathbb{E}_{\pi_{\boldsymbol{y}}(\boldsymbol{y}) \mathcal{N}(\varepsilon | 0, I)} \frac{1}{2\sigma_T^2} \|\boldsymbol{y} + \sigma_t \varepsilon\|^2 - \log \mathbb{E}_{p^{\mathcal{T}}(\boldsymbol{y}) \mathcal{N}(\varepsilon | 0, I)} e^{\|\boldsymbol{y} + \sigma_t \varepsilon\|^2 / (2\sigma_T^2)}\right) \mathrm{d}t. \tag{43}$$

We rewrite $\|\boldsymbol{y}+\sigma_t\varepsilon\|^2$ as $\|\boldsymbol{y}\|^2+2\sigma_t\langle\boldsymbol{y},\sigma_t\varepsilon\rangle+\sigma_t^2\|\varepsilon\|^2$ and note that expectation of the second term is zero. The first term is then equal to

$$\frac{1}{2\sigma_T^2}\int\limits_0^T \omega_t\,\mathrm{d}t \cdot \mathbb{E}_{\pi_{\boldsymbol{y}}(\boldsymbol{y})}\|\boldsymbol{y}\|^2 + \frac{1}{2\sigma_T^2}\int\limits_0^T \omega_t\,\sigma_t^2\mathrm{d}t \cdot \mathbb{E}_{\mathcal{N}(\varepsilon|0,I)}\|\varepsilon\|^2. \tag{44}$$

Boundedness of $\omega_t$ (Assumption B.3) implies that the first integral is finite and, say, equal to $C_1$. The second integral contains a product of bounded $\omega_t$ and continuous $\sigma_t^2$ (Assumtion B.4), which is also integrable. We then denote the second summand by $C_2$ and rewrite the first summand as

$$C_1\mathbb{E}_{\pi_{\boldsymbol{y}}(\boldsymbol{y})}\|\boldsymbol{y}\|^2 + C_2. \tag{45}$$

As for the second summand, we see that the expectation

$$\mathbb{E}_{p^{\mathcal{T}}(\boldsymbol{y})\mathcal{N}(\varepsilon|0,I)}e^{\|\boldsymbol{y}+\sigma_t\varepsilon\|^2/(2\sigma_T^2)} \tag{46}$$

with respect to $\varepsilon$ will be finite, because $\sigma_t^2/(2\sigma_T^2)$ is always less than $1/2$, which will make the exponent have negative degree. Moreover, simple calculations show that this function will be continuous with respect to $\sigma_t$ and have only quadratic terms with respect to $\boldsymbol{y}$ inside the exponent, i.e. have the form

$$e^{a(\sigma_t)\|\boldsymbol{y}-b(\sigma_t)\|^2+c(\sigma_t)} \tag{47}$$

with continuous $a, b, c$. We now want to prove that the expectation

$$\mathbb{E}_{p^{\mathcal{T}}(\boldsymbol{y})}e^{\alpha(\sigma_t)\|\boldsymbol{y}-\beta(\sigma_t)\|^2+\gamma(\sigma_t)} \tag{48}$$

will also be continuous in $t$. First, due to the boundedness of $\boldsymbol{y}$, this expectation is finite. Second, for $t_n \to t$:

$$\lim_{n\to\infty}\mathbb{E}_{p^{\mathcal{T}}(\boldsymbol{y})}e^{a(\sigma_{t_n})\|\boldsymbol{y}-b(\sigma_{t_n})\|^2+c(\sigma_{t_n})} = \tag{49}$$

$$= \mathbb{E}_{p^{\mathcal{T}}(\boldsymbol{y})}\lim_{n\to\infty}e^{a(\sigma_{t_n})\|\boldsymbol{y}-b(\sigma_{t_n})\|^2+c(\sigma_{t_n})} = \tag{50}$$

$$= \mathbb{E}_{p^{\mathcal{T}}(\boldsymbol{y})}e^{a(\sigma_t)\|\boldsymbol{y}-b(\sigma_t)\|^2+c(\sigma_t)} \tag{51}$$

due to the Theorem B.16 (Lebesgue's dominated convergence). It is applicable, since $\boldsymbol{y}$ is bounded and all the functions are continuous, thus bounded in $[0, T]$.

We thus obtain that the second integral contains bounded $\omega_t$ multiplied by the logarithm of continuous function, which is always $\geq 1$ (positive exponent). This means that the whole integral is finite. Denoting it by $C_3$, we obtain

$$C_1\mathbb{E}_{\pi_{\boldsymbol{y}}(\boldsymbol{y})}\|\boldsymbol{y}\|^2 + C_2 - C_3 \leq \int\limits_0^T \omega_t\,\mathrm{KL}\left(\pi_{\boldsymbol{y},t}\,\|\,p_t^{\mathcal{T}}\right)\mathrm{d}t. \tag{52}$$

Combined with the condition of the lemma, we obtain

$$C_1\mathbb{E}_{\pi_{\boldsymbol{y}}(\boldsymbol{y})}\|\boldsymbol{y}\|^2 + C_2 - C_3 \leq \int\limits_0^T \omega_t\,\mathrm{KL}\left(\pi_{\boldsymbol{y},t}\,\|\,p_t^{\mathcal{T}}\right)\mathrm{d}t \leq C, \tag{53}$$

which implies

$$\mathbb{E}_{\pi_{\boldsymbol{y}}(\boldsymbol{y})}\|\boldsymbol{y}\|^2 \leq \frac{C+C_3-C_2}{C_1} := C_4. \tag{54}$$

We thus obtained a uniform bound on some statistic with respect to all measures from $\{\pi^n\}$. The function $\|\boldsymbol{y}\|^2$ has compact sublevel sets $\{\|\boldsymbol{y}\|^2 \leq r\}$. Lemma B.8 then states that the sequence $\pi_{\boldsymbol{y}}^n$ is tight, i.e. for all $\varepsilon > 0$ there is a compact set $K_\varepsilon$ with $\pi_{\boldsymbol{y}}^n(\boldsymbol{y} \in K_\varepsilon) \geq 1 - \varepsilon$.

Finally, marginal $\boldsymbol{x}$ distribution of each of the $\pi^n$ is $p^{\mathcal{S}}$, which is bounded (Assumption B.1), i.e. there is a compact $K$ that $\pi^n(\boldsymbol{x} \in K) = 1$. Combined with the previous observation, we obtain

$$\pi^n(\boldsymbol{x} \in K, \boldsymbol{y} \in K_\varepsilon) \geq 1 - \varepsilon \tag{55}$$

for all $n$. The cartesian product $K \times K_\varepsilon$ is also compact. Theorem B.7 (Prokhorov) then implies that the sequence $\pi^n$ is tight. $\qquad\square$

Now we are ready to prove the following

**Lemma B.23.** *Infimum of the loss $\mathcal{L}^\alpha(\pi)$ over all generator-based transport plans $\pi$ (with $\pi_{\boldsymbol{x}} = p^{\mathcal{S}}$ and $\pi(\boldsymbol{y} = G(\boldsymbol{x}))$ for some $G$) is attained on some plan $\hat{\pi}$.*

*Proof.* We start by observing that there is at least one feasible $\pi$ with the aforementioned properties. For this purpose one can take the optimal transport map $G^\infty$ between $p^{\mathcal{S}}$ and $p^{\mathcal{T}}$, which is unique by Theorem B.18 under Assumptions B.1, B.2.

Let $\pi^n$ be a sequence of feasible generator-based measures that $\mathcal{L}^\alpha(\pi^n)$ converges to the corresponding infimum $\mathcal{L}^\alpha_{\inf}$ (it exists by the definition of the infimum). Without loss of generality, we can assume that $\mathcal{L}^\alpha(\pi^n) \leq \mathcal{L}^\alpha_{\inf} + 1$ for all $n$ (if not, one can drop large enough sequence prefix). This implies that for all $n$ holds

$$\alpha \int\limits_0^T \omega_t \, \mathrm{KL}\left(\pi_{\boldsymbol{y},t} \,\|\, p_t^{\mathcal{T}}\right) \mathrm{d}t \leq \mathcal{L}^\alpha_{\inf} + 1. \tag{56}$$

Lemma B.22 implies that the sequence $\pi^n$ is tight. Prokhorov theorem then states that $\pi^n$ has a weakly convergent subsequence $\pi^{n_k} \xrightarrow{w} \hat{\pi}$. Lower semi-continuity of the loss $\mathcal{L}^\alpha$ implies that

$$\liminf_{k \to \infty} \mathcal{L}^\alpha(\pi^{n_k}) \geq \mathcal{L}^\alpha(\hat{\pi}) \geq \mathcal{L}^\alpha_{\inf}. \tag{57}$$

At the same time, $\mathcal{L}^\alpha(\pi^{n_k})$ is assumed to converge to $\mathcal{L}^\alpha_{\inf}$, which means that $\hat{\pi}$ is indeed the minimum. $\square$

### B.6 FINISH OF THE PROOF

*Theorem B.1 proof.* Finally, we combine the previous technical observations with the proof sketch from the Section B.1. Let $\alpha_n \to \infty$ be a sequence of coefficients, $G^{\alpha_n}$ be the optimal generators with respect to $\mathcal{L}^{\alpha_n}$ and $\pi^{\alpha_n}$ the joint distributions of $(\boldsymbol{x}, G^{\alpha_n}(\boldsymbol{x}))$. Additionally, we define $\pi^\infty$ to be the optimal transport plan, corresponding to $(\boldsymbol{x}, G^\infty(\boldsymbol{x}))$, where $G^\infty(\boldsymbol{x})$ is the optimal transport map. First, due to the monotonicity of $\mathcal{L}^\alpha$ with respect to $\alpha$, we have

$$\mathcal{L}^{\alpha_n}(\pi^{\alpha_n}) \leq \mathcal{L}^{\alpha_{n+1}}(\pi^{\alpha_{n+1}}) \leq \mathcal{L}^\infty(\pi^\infty). \tag{58}$$

This implies that for all $n$ holds

$$\alpha_n \int\limits_0^T \omega_t \, \mathrm{KL}\left(\pi_{\boldsymbol{y},t}^{\alpha_n} \,\|\, p_t^{\mathcal{T}}\right) \mathrm{d}t \leq \mathcal{L}^\infty(\pi^\infty) \Rightarrow \tag{59}$$

$$\Rightarrow \int\limits_0^T \omega_t \, \mathrm{KL}\left(\pi_{\boldsymbol{y},t}^{\alpha_n} \,\|\, p_t^{\mathcal{T}}\right) \mathrm{d}t \leq \frac{\mathcal{L}^\infty(\pi^\infty)}{\alpha_n} \leq \frac{\mathcal{L}^\infty(\pi^\infty)}{\min\limits_n \alpha_n}, \tag{60}$$

which is finite, since $\alpha_n \to +\infty$. One more time, we apply Lemma B.22 and conclude that the sequence $\pi^{\alpha_n}$ is tight.

Let $\pi^{\alpha_{n_k}}$ be its weakly convergent subsequence: $\pi^{\alpha_{n_k}} \xrightarrow{w} \hat{\pi}$. Analogously to the Section B.1, we observe that

$$\liminf_{k \to \infty} \mathcal{L}^{\alpha_{n_k}}(\pi^{\alpha_{n_k}}) \geq \liminf_{k \to \infty} \mathcal{L}^{\alpha_{n_m}}(\pi^{\alpha_{n_k}}) \geq \mathcal{L}^{\alpha_{n_m}}(\hat{\pi}) \tag{61}$$

for any fixed $m$. The first inequality is due to the monotonicity of $\mathcal{L}^\alpha$ with respect to $\alpha$ and second is the implication of lower semi-continuity of the loss $\mathcal{L}^\alpha$ with respect to weak convergence. Taking the limit $m \to \infty$, we obtain

$$\liminf_{k \to \infty} \mathcal{L}^{\alpha_{n_k}}(\pi^{\alpha_{n_k}}) \geq \mathcal{L}^\infty(\hat{\pi}). \tag{62}$$

Combining all these observations, we obtain the following sequence of inequalities

$$\mathcal{L}^\infty(\pi^\infty) \geq \liminf_{k \to \infty} \mathcal{L}^{\alpha_{n_k}}(\pi^{\alpha_{n_k}}) \geq \mathcal{L}^\infty(\hat{\pi}) \geq \mathcal{L}^\infty(\pi^\infty), \tag{63}$$

which implies that the limiting measure $\hat{\pi}$ reaches the minimum of the objective over generator-based plans. By the uniqueness of the optimal transport map $G^{\infty}$ under the Assumptions B.1, B.2, B.3, we conclude that all the convergent subsequences $\pi^{\alpha_{n_k}}$ converge to the optimal measure $\pi^{\infty}$. Using Corollary B.9 of the Prokhorov theorem, we deduce that $\pi^{\alpha_n} \xrightarrow{w} \pi^{\infty}$.

Finally, we want to replace the weak convergence of $\pi^{\alpha_n}$ to $\pi^{\infty}$ with the convergence in probability of the generators, i.e. show

$$G^{\alpha_n} \xrightarrow{p^{\mathcal{S}}} G^{\infty}. \tag{64}$$

To this end, we represent the corresponding probability as the expectation of the indicator and upper bound it with a continuous function:

$$p^{\mathcal{S}}\left(\|G^{\alpha_n}(\boldsymbol{x}) - G^{\infty}(\boldsymbol{x})\| > \varepsilon\right) = \mathbb{E}_{p^{\mathcal{S}}(\boldsymbol{x})} I\{\|G^{\alpha_n}(\boldsymbol{x}) - G^{\infty}(\boldsymbol{x})\| > \varepsilon\} \tag{65}$$

$$\leq \mathbb{E}_{p^{\mathcal{S}}(\boldsymbol{x})} d\left(G^{\alpha_n}(\boldsymbol{x}), G^{\infty}(\boldsymbol{x})\right), \tag{66}$$

where $d$ is a continuous indicator approximation, defined as

$$d(\boldsymbol{u}, \boldsymbol{v}) = \begin{cases} \frac{\|\boldsymbol{u}-\boldsymbol{v}\|}{\varepsilon}, & \text{if } 0 \leq \|\boldsymbol{u} - \boldsymbol{v}\| < \varepsilon; \\ 1, & \text{if } \|\boldsymbol{u} - \boldsymbol{v}\| \geq \varepsilon. \end{cases} \tag{67}$$

We define the test function

$$\varphi(\boldsymbol{x}, \boldsymbol{y}) = d\left(\boldsymbol{y}, G^{\infty}(\boldsymbol{x})\right) \tag{68}$$

and rewrite the upper bound as

$$\mathbb{E}_{p^{\mathcal{S}}(\boldsymbol{x})} d\left(G^{\alpha_n}(\boldsymbol{x}), G^{\infty}(\boldsymbol{x})\right) = \mathbb{E}_{p^{\mathcal{S}}(\boldsymbol{x})} \varphi(\boldsymbol{x}, G^{\alpha_n}(\boldsymbol{x})) = \mathbb{E}_{\pi^{\alpha_n}(\boldsymbol{x}, \boldsymbol{y})} \varphi(\boldsymbol{x}, \boldsymbol{y}). \tag{69}$$

Due to Assumptions B.1, B.2 and Theorem B.14 the optimal transport map $G^{\infty}$ is continuous, which implies that this test function is bounded and continuous. Given the weak convergence of $\pi^{\alpha_n}$, we have

$$\mathbb{E}_{\pi^{\alpha_n}(\boldsymbol{x}, \boldsymbol{y})} \varphi(\boldsymbol{x}, \boldsymbol{y}) \rightarrow \mathbb{E}_{\pi^{\infty}(\boldsymbol{x}, \boldsymbol{y})} \varphi(\boldsymbol{x}, \boldsymbol{y}) = \mathbb{E}_{p^{\mathcal{S}}(\boldsymbol{x})} \varphi(\boldsymbol{x}, G^{\infty}(\boldsymbol{x})) = \tag{70}$$

$$= \mathbb{E}_{p^{\mathcal{S}}(\boldsymbol{x})} d(G^{\infty}(\boldsymbol{x}), G^{\infty}(\boldsymbol{x})) = 0, \tag{71}$$

which implies the desired

$$p^{\mathcal{S}}\left(\|G^{\alpha_n}(\boldsymbol{x}) - G^{\infty}(\boldsymbol{x})\| > \varepsilon\right) \rightarrow 0. \tag{72}$$

$\square$

## C  Toy Experiment

We validate the qualitative properties of the RDMD method on 2-dimensional *Gaussian → Swiss-roll*. In this setting, we explore the effect of varying the regularization coefficient $\lambda$ on the trained transport map $G_{\theta}$. In particular, we study its impact on the transport cost and fitness to the target distribution $p^{\mathcal{T}}$. In the experiment, both source and target distributions are represented with 5000 independent samples. We use the same small MLP-based architecture from Shi et al. (2024) for all the networks.

The main results are presented in Figure 6. The standard DMD ($\lambda = 0.0$) learns a transport map with several intersections when demonstrated as the set of lines between the inputs and the outputs. This observation means that the learned map is not OT, because it is not cycle-monotone (McCann, 1995). Increasing $\lambda$ yields fewer intersections, which can be used as a proxy evidence of optimality. At the same time, the generator output distribution becomes farther and farther from the desired target. The results show the importance of choosing the appropriate $\lambda$ to obtain a better trade-off between the two properties. Here, the regularization coefficient $\lambda = 0.2$ offers a good trade-off by having small intersections and producing output distribution close to the target.

## D  Ablation of the initialization parameter

In this section, we further explore the design space of our method by investigating the effect of the fixed generator input noise parameter $\sigma$ on the resulting quality. To this end, we take the colored

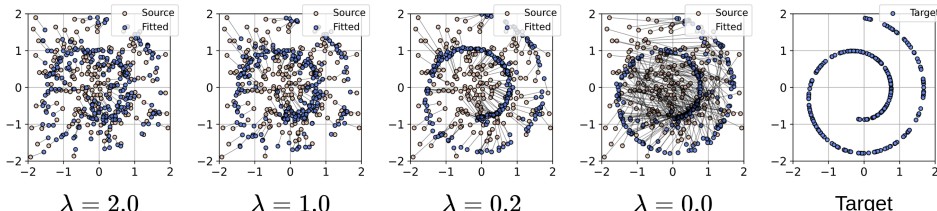

Figure 6: Visualization of RDMD mappings on *Gaussian → Swissroll* with different regularization coefficients $\lambda$.

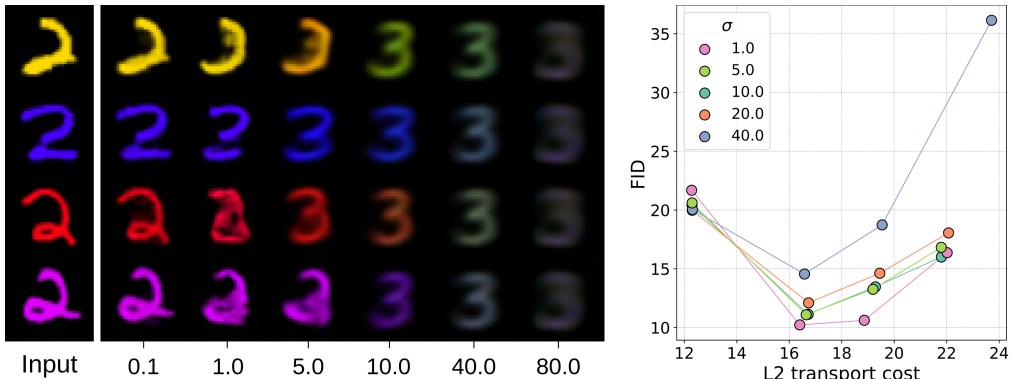

Figure 7: Left: visualization of the generator initialization at various $\sigma \in [0.1, 80.0]$, where $\sigma$ is the noise level parameter residual from the pre-trained diffusion architecture. Right: comparison of different $\sigma$ in terms of the quality-faithfulness trade-off. The metrics are obtained by initializing the generator at the corresponding $\sigma$ level and training it with the RDMD procedure. Here, $\lambda \in \{0, 1.0, 2.0, 4.0\}$. Higher $\lambda$ corresponds to the lower transport cost values.

version of the MNIST (LeCun, 1998) data set and perform translation between the digits "2" and "3" initializing from various $\sigma$. We use a small UNet architecture from Gushchin et al. (2024a).

The parameter $\sigma$ is residual from the pre-trained diffusion architecture and is, therefore, fixed throughout training and evaluation. However, the target denoiser network tries to convert the expected noisy input into the corresponding sample from the output distribution. Consequently, one may expect that at a suitable noise level, the generator may change the input's details to make them look appropriate for the target while preserving the original structural properties.

We demonstrate this effect on various noise levels in Figure 7. Here we observe that the small sigmas lead to the mapping close to the identity, whereas the large sigmas lead to almost constant blurry images, corresponding to the average "3" of the data set. However, there is a segment $[1.0, 10.0]$ of levels that gives a moderate-quality mapping in terms of both faithfulness and realism, which makes it a suitable initial point. Note that the FID-L2 plot is not monotone at high L2 values due to the overall poor quality of the generator, i.e. it outputs bad-quality pictures slightly related to the source. We further investigate optimal $\sigma$ choice by going through a 2D grid of the hyperparameters $(\sigma, \lambda)$ and aim to see if it is possible to choose the uniform best noise level. In Figure 7 we report the faithfulness-quality trade-off concerning various $\sigma$. We observe that there is almost monotone dependence on $\sigma$ on the segment $[1.0, 40.0]$: here the $\sigma = 1.0$ gives almost uniformly best results in terms of both metrics. Similar results are obtained by the values $5.0, 10.0$ which have fair quality visual results at initialization. Therefore, we conclude that it is best to choose the least parameter $\sigma$ among the parameters with appropriate visuals at the initial point.

# E    EXPERIMENTAL DETAILS

## E.1    GENERAL DETAILS

**Metrics measurement.**    In image-to-image experiments, we measure FID, $\sqrt{L_2}$ distance, PSNR, SSIM and LPIPS. We do not preprocess images before calculating the corresponding metrics (i.e. we perform measurements on images in $[0, 1]$ range with the original resolution) except LPIPS, which takes input in $[-1, 1]$. We use the official LPIPS (Zhang et al., 2018) implementation with VGG (Simonyan & Zisserman, 2014) backbone. We calculate FID with the script, provided by Karras et al. (2022). In all pixel-space experiments we use the VE schedule with $\sigma_t = t$ and $T = 80.0$ as in Karras et al. (2022).

In all image-to-image experiments we measure FID between model outputs on the source test data set and the target train data set. This corresponds to the FID measurement pipeline by Park et al. (2020).

As for the transport cost $\sqrt{L_2}$, we first measure the average squared distance between inputs and outputs of the generator (without normalizing with respect to the image dimension). After averaging, we take the square root.

## E.2    2D EXPERIMENTS

**Architecture.**    We take the architecture from toy experiments of De Bortoli et al. (2021) for the diffusion model and the generator. It consists of an input-encoding MLP block, a time-encoding MLP block, and a decoding MLP block. The input-encoding MLP block consists of 4 hidden layers with dimensions $[16, 32, 32, 32]$ interspersed by LeakyReLU activations. The time-encoding MLP consists of a positional encoding layer (Vaswani et al., 2017) and follows the same MLP block structure as the input encoder. The decoding MLP block consists of 5 hidden layers with dimensions $[128, 256, 128, 64, 2]$ and operates on concatenated time embedding and input embedding each obtained from their respective encoder. The model contains $88k$ parameters.

**Training Diffusion Model.**    The diffusion model is trained for 100k iterations with batch size 1024 with Adam optimizer (Kingma & Ba, 2014) with learning rate $10^{-4}$.

**Training RDMD.**    Fake denoising network is trained with Adam optimizer with learning rate $10^{-4}$. The generator model is trained with a different Adam optimizer with a learning rate of $2 \cdot 10^{-5}$. We train RDMD for 100k iterations with batch size 1024.

**Computational resources.**    We conduct all of the toy experiments on the CPU. Running 100k iterations with the batch size 1024 takes approximately 1 hour.

## E.3    COLORED MNIST

**Architecture.**    We use the architecture from Gushchin et al. (2024a), which utilizes convolutional UNet with conditional instance normalization on time embeddings used after each upscaling block of the decoder. The model produces time embeddings via positional encoding. The model has approximately $9.9M$ parameters.

**Training Diffusion Model.**    The diffusion model is trained for 24500 iterations with batch size 8192. We use the Adam optimizer with a learning rate of $4 \cdot 10^{-3}$. The model is trained in FP32. It obtains FID equal to 2.09.

**Training RDMD.**    Fake denoising network is trained with Adam optimizer with a learning rate of $2 \cdot 10^{-3}$. The generator model is trained with Adam optimizer with learning rate $5 \cdot 10^{-5}$. RDMD is trained for 7300 iterations with batch size 4096.

**Computational resources.**    We conduct all of the experiments on 2x NVIDIA GeForce RTX 4090 GPUs. Training Diffusion model for 24500 iterations with the batch size 8192 takes approximately 6 hours. Training RDMD for 7300 iterations with batch size 4096 takes approximately 3 hours.

### E.4 AFHQV2-64 EXPERIMENTS

**Architecture.** We use the SongUNet architecture from EDM (Karras et al., 2022) repository, which corresponds to DDPM++ network, introduced by Song et al. (2020b), for both *Wild* and *Cat* data sets. The model contains approximately $55M$ parameters.

**Training Diffusion Models.** The diffusion models for *Wild* and *Cat* sets are trained for 80k and 35k iterations, respectively. We set the batch size to 512 and choose the best checkpoint according to FID. We use the Adam optimizer with a learning rate of $2 \cdot 10^{-4}$. We use a dropout rate equal to 0.25 during the training and the augmentation pipeline from Karras et al. (2022) with a probability of 0.15. The models are trained in FP32. Training takes approximately 35/15 hours on $4\times$ NVidia Tesla A100 80GB. The models obtain FID equal to 2.01 (*Wild*) and 3.5 (*Cat*).

**Training RDMD.** In all runs, we initialize the generator from the target diffusion model with the fixed $\sigma = 1.0$. We run 5 models, corresponding to the regularization coefficients $\lambda = \{0.0, 0.02, 0.05, 0.1, 0.2\}$. All models are trained with the Adam optimizer with a generator's learning rate of $5 \cdot 10^{-5}$ and a fake diffusion's learning rate of $10^{-4}$. We perform 3 fake score updates per generator update. We train all models for 30000 generator updates with batch size 256. Training takes approximately 3 days on $4\times$ NVidia Tesla V100 32GB.

**ILVR hyperparameters.** The only hyperparameter of ILVR is the downsampling factor $N$ for the low-pass filter, which determines whether guidance would be conducted on coarser or finer information. $n_{\text{steps}}$ denotes the number of sampling steps. All metrics in Figures 3, 8 and 9 for ILVR are obtained on the following hyperparameter grid: $N = [2, 4, 8, 16, 32]$, $n_{\text{steps}} = 18$. We exclude runs with the same statistical significance and achieving FID higher than 30.0. The images in Figures 12, 13 and the results in Table 2 are obtained with hyperparameters ($N = 16, n_{\text{steps}} = 18$) for both *Wild → Cat* and *Cat → Wild* translation problems.

**SDEdit hyperparameters.** The only hyperparameter of SDEdit is the noise level $\sigma$, which acts as a starting point for sampling. The higher the noise level, the closer the sampling procedure is to unconditional generation. The smaller the noise values, the more features are carried over to the target domain at the expense of generation quality. $n_{\text{steps}}$ denotes the number of sampling steps. All metrics in Figures 3, 8 and 9 for SDEdit are obtained on the following hyperparameter grid: $\sigma = [1, 2, 3, 5, 7, 10, 15, 20, 40]$, $n_{\text{steps}} = 18$. We exclude runs with the same statistical significance and achieving FID higher than 30.0. The images in Figures 12, 13 and the results in Table 2 are obtained with hyperparameters ($\sigma = 7, n_{\text{steps}} = 18$) for both *Wild → Cat* and *Cat → Wild* translation problems.

**EGSDE hyperparameters.** EGSDE sampling hyperparameters include the initial noise level $\sigma$ at which the source image is perturbed, and the downsampling factor $N$ for the low-pass filter. $n_{\text{steps}}$ denotes the number of sampling steps. The method also has parameters which regulate the guidance weight of domain-specific energy term $\lambda_s$ and domain-independent energy term $\lambda_i$. We take them by default being equal to $\lambda_s = 500.0$ and $\lambda_i = 2.0$ as in the original EGSDE paper Zhao et al. (2022). All metrics in Figures 3, 8 and 9 for EGSDE are obtained on the following hyperparameter grid: $\sigma = [2, 3.4241, 7, 10, 20]$, $N = [8, 16]$, $n_{\text{steps}} = 18$. Here, $\sigma = 3.4241$ corresponds to the time step $T = 500$ from the original DDPM formulation. We exclude runs with the same statistical significance and achieving FID higher than 30.0. The images in Figures 12, 13 and the results in Table 2 are obtained with hyperparameters ($\sigma = 7, N = 16, n_{\text{steps}} = 18$) for *Wild → Cat* and ($\sigma = 10, N = 16, n_{\text{steps}} = 18$) for *Cat → Wild*.

**DDIB and CycleDiffusion hyperparameters.** We run encoding and decoding in DDIB with the deterministic EDM sampler (2nd order Heun solver) with 18 steps ($35 + 35 = 70$ function evaluations in total).

All metrics in Figures 3, 8 and 9 for CycleDiffusion model are obtained with encoding step $T_{es} = [20, 30, 40, 50, 60, 70, 80]$ in DDIM schedule with $\eta = 0.7$ and 100 steps, which results in $T_{es} + T_{es}$ neural function evaluations needed for encoding the source image with the source domain network and decoding with the target domain network via DDIM ancestral sampling. The

images in Figures 12, 13 and the results in Table 2 are obtained with hyperparameter $T_{es} = 70$ for both *Cat → Wild* and *Wild → Cat* translation problems.

**ASBM hyperparameters.** We follow the experimental setup suggested by Gushchin et al. (2024b). We set the starting coupling as the Mini-Batch Optimal Transport. We use the 0-th outer iteration and perform $1000000$ generator gradient updates to "pretrain" the processes. The next 5 outer iterations perform $40000$ generator gradient updates each. Training Markovian projections consists of training the transitional density networks via the DD-GAN (Xiao et al., 2022). The number of transition (inner) steps $N$ is equal to 3. Generator to Discriminator optimization steps ratio is 1-to-1. Both the generator and the discriminator are trained with the Adam optimizer. The learning rate for the generator is $1.25 \cdot 10^{-4}$ and for the discriminator is $1.6 \cdot 10^{-4}$ and the batch size is equal to 32. Exponential Moving Average is applied to generator's weight during training with decay equal to 0.999.

**DIOTM hyperparameters.** We follow the experimental settings suggested by Choi et al. (2024) and use the code attached as the supplementary material to the ICLR 2025 submission to run the experiments. The method has two main hyperparameters $\alpha$, which regularizes the cost between the input and output of the transport map, and $\lambda$, which controls the intensity of HJB regularization and is important for the training stability. We set $\alpha = 0.0005$ and $\lambda = 10$. We use the Adam optimizer with learning rate $10^{-4}$ and betas $(0, 0.9)$ and train the method for 60K iterations with batch size equal to 64. The cosine schedule is used to gradually decrease the learning rate to $5 \cdot 10^{-5}$. We obtain the best results on the 30K-th iteration and use the checkpoints from it for our evaluations.

### E.5 CELEBA EXPERIMENTS

**Architecture.** We use the DhariwalUNet architecture from EDM (Karras et al., 2022) repository, which corresponds to the ADM network, introduced by Dhariwal & Nichol (2021), for both *Male* and *Female* data sets. The only difference is that we use $128$ model channels instead of the original $192$. The model contains approximately $130M$ parameters.

**Training Diffusion Model.** The diffusion models for *Male* and *Female* are both trained for 340k iterations. We set the batch size to 256 and choose the best checkpoint according to FID. We use the Adam optimizer with a learning rate of $1 \cdot 10^{-4}$. We use a dropout rate equal to $0.05$ during the training and the augmentation pipeline from Karras et al. (2022) with a probability of $0.1$. At training, we sample $\log \sigma$ from the standard normal distribution, which corresponds to parameters $(P_{\mathrm{mean}} = 0.0, P_{\mathrm{std}} = 1.0)$ from Karras et al. (2022). The models are trained in FP16. Training takes approximately 7 days on $8\times$ NVidia Tesla A100 80GB. The models obtain FID equal to 3.57 (*Male*) and 3.17 (*Female*).

**Training RDMD.** In all runs, we initialize the generator from the target diffusion model with the fixed $\sigma = 3.0$. We run 3 models, corresponding to the regularization coefficients $\lambda = \{0.0, 0.05, 0.075\}$. All models are trained with the Adam optimizer with a generator's learning rate of $5 \cdot 10^{-5}$ and fake diffusion's learning rate of $1 \cdot 10^{-4}$. We perform 3 fake score updates per generator update. We train all models for 40000 iterations with batch size 256. Training takes approximately 3.5 days on $8\times$ NVidia Tesla A100 80GB.

**ILVR hyperparameters.** The only hyperparameter of ILVR is the downsampling factor $N$ for the low-pass filter, which determines whether guidance would be conducted on coarser or finer information. $n_{\mathrm{steps}}$ denotes the number of sampling steps. All metrics in Figures 10 and 11 for ILVR are obtained on the following hyperparameter grid: $N = [2, 4, 8, 16, 32, 64]$, $n_{\mathrm{steps}} = 18$. We exclude runs with the same statistical significance and achieving FID higher than $30.0$. The images in Figures 14, 15 and the results in Table 2 are obtained with hyperparameters $(N = 32, n_{\mathrm{steps}} = 18)$ for both *Male → Female* and *Female → Male* translation problems.

**SDEdit hyperparameters.** The only hyperparameter of SDEdit is the noise level $\sigma$, which acts as a starting point for sampling. The higher the noise level, the closer the sampling procedure is to unconditional generation. The smaller the noise values, the more features are carried over to the target domain at the expense of generation quality. $n_{\mathrm{steps}}$ denotes the number of sampling steps.

All metrics in Figures 10 and 11 for SDEdit are obtained on the following hyperparameter grid: $\sigma = [1, 2, 3, 3.4241, 5, 7, 10, 15, 20, 40]$, $n_{\text{steps}} = 18$. Here, $\sigma = 3.4241$ corresponds to the time step $T = 500$ from the original DDPM formulation. We exclude runs with the same statistical significance and achieving FID higher than 30.0. The images in Figures 14, 15 and the results in Table 2 are obtained with hyperparameters ($\sigma = 20, n_{\text{steps}} = 18$) for both *Male → Female* and *Female → Male* translation problems.

**EGSDE hyperparameters.** EGSDE sampling hyperparameters include the initial noise level $\sigma$ at which the source image is perturbed, and the downsampling factor $N$ for the low-pass filter. $n_{\text{steps}}$ denotes the number of sampling steps. The method also has parameters which regulate the guidance weight of domain-specific energy term $\lambda_s$ and domain-independent energy term $\lambda_i$. We take them by default being equal to $\lambda_s = 500.0$ and $\lambda_i = 2.0$ as in the original EGSDE paper Zhao et al. (2022). All metrics in Figures 10 and 11 for EGSDE are obtained on the following hyperparameter grid: $\sigma = [2, 3.4241, 7, 10, 20]$, $N = [16, 32]$, $n_{\text{steps}} = 18$. Here, $\sigma = 3.4241$ corresponds to the time step $T = 500$ from the original DDPM formulation. We exclude runs with the same statistical significance and achieving FID higher than 30.0. The images in Figures 14, 15 and the results in Table 2 are obtained with hyperparameters ($\sigma = 20, N = 32, n_{\text{steps}} = 18$) for both *Male → Female* and *Female → Male* translation problems.

**DDIB and CycleDiffusion hyperparameters.** We run encoding and decoding in DDIB with the deterministic EDM sampler (2nd order Heun solver) with 18 steps ($35 + 35 = 70$ function evaluations in total).

All metrics in Figures 10 and 11 for CycleDiffusion model are obtained with encoding step $T_{es} = [20, 30, 40, 50, 60, 70, 80]$ in DDIM schedule with $\eta = 1.0$ and 100 steps, which results in $T_{es} + T_{es}$ neural function evaluations needed for encoding the source image with the source domain network and decoding with the target domain network via DDIM ancestral sampling. The images in Figures 14, 15 and the results in Table 2 are obtained with hyperparameter $T_{es} = 80$ for *Male → Female* and $T_{es} = 70$ for *Female → Male*.

**ASBM hyperparameters.** We follow the experimental setup suggested by Gushchin et al. (2024b). We set the starting coupling as the Mini-Batch Optimal Transport. We use the 0-th outer iteration and perform 1000000 generator gradient updates to "pretrain" the processes. The next 5 outer iterations perform 40000 generator gradient updates each. Training Markovian projections consists of training the transitional density networks via the DD-GAN (Xiao et al., 2022). The number of transition (inner) steps $N$ is equal to 3. Generator to Discriminator optimization steps ratio is 1-to-1. Both the generator and the discriminator are trained with the Adam optimizer. The learning rate for the generator is $1.25 \cdot 10^{-4}$ and for the discriminator is $1.6 \cdot 10^{-4}$ and the batch size is equal to 32. Exponential Moving Average is applied to generator's weight during training with decay equal to 0.9999.

**DIOTM hyperparameters.** We follow the experimental settings suggested by Choi et al. (2024) and use the code attached as the supplementary material to the ICLR 2025 submission to run the experiments. The method has two main hyperparameters $\alpha$, which regularizes the cost between the input and output of the transport map, and $\lambda$, which controls the intensity of HJB regularization and is important for the training stability. We set $\alpha = 0.001$ and $\lambda = 10$. We use the Adam optimizer with learning rate $10^{-4}$ and betas $(0, 0.9)$ and train the method for 100K iterations with batch size equal to 64. The cosine schedule is used to gradually decrease the learning rate to $5 \cdot 10^{-5}$. We obtain the best results on the 70K-th iteration and use the checkpoints from it for our evaluations.

### E.6 IMAGENET EXPERIMENTS

**Experimental Setup.** In the ImageNet experiment, we train one model to perform translation between any pair of ImageNet classes. Theoretically, one could directly train the model to translate between any pairs of classes, but many of them are not particularly meaningful (e.g. translating dogs into cars) and may harm model's performance. To this end, we construct a constrained dataset, in which each input class is translated into 20 visually nearest classes. We choose the nearest classes by performing zero-shot classification of input class pictures with CLIP (Radford et al., 2021). Specifically, we take 20 of the most probable classes according to the probability vector obtained by

averaging CLIP's classification outputs for 5 input images (see examples of nearest classes in Appendix E.6). We note that this limitation of the dataset **does not** necessarily harm the model's performance for translation between any pairs of classes. To this end, we validate its high-quality results on out-of-domain pairs of classes in Figures 16, 17, 18, 19, 20 in Appendix F.3.

We take $256 \times 256$ class-conditional LDM (Rombach et al., 2022) as the pre-trained target score and use it as initialization for both the generator and the fake score. We use classifier-free guidance scale of 3.0 for the target score during training.

**Architecture.** We use the pre-trained class-conditional LDM-4 Rombach et al. (2022) model with approximately $400M$ parameters. It operates in the latent space of LDM-VQ-4 model of dimension $64 \times 64 \times 3$. It achieves FID=3.6 with classifier-free guidance scale of 1.5.

Table 6: Examples of source-target pairs used for training in ImageNet Experiments

| Source class | Top-20 neighbouring target classes |
|---|---|
| orange | lemon, grocery store, butternut squash, fig, jackfruit, spaghetti squash, custard apple, mixing bowl, bell pepper, pomegranate, acorn squash, Granny Smith, honeycomb, web site, screwdriver, tennis ball, brambling, shopping basket, Petri dish, ping-pong ball |
| ladybug | leaf beetle, leafhopper, long-horned beetle, dung beetle, weevil, ground beetle, rhinoceros beetle, bee, American coot, tick, garden spider, hermit crab, snail, tiger beetle, harvestman, ant, lacewing, European gallinule, African grey, barn spider |
| volcano | mountain tent, geyser, Great Pyrenees, alp, mountain bike, promontory, orange, cliff, radio telescope, jacamar, catamaran, caldron, indri, water ouzel, fire screen, web site, barrow, torch, breakwater, valley |
| giant panda | guinea pig, indri, sloth bear, gibbon, three-toed sloth, lesser panda, French bulldog, colobus, siamang, American black bear, dogsled, badger, skunk, chow, tusker, Border collie, black-footed ferret, capuchin, brown bear, howler monkey |
| golf ball | croquet ball, ping-pong ball, soccer ball, honeycomb, tennis ball, rugby ball, hand blower, earthstar, thimble, bottlecap, mushroom, measuring cup, projectile, tiger, swing, agaric, buckeye, acorn, stinkhorn, racket |

**Training RDMD.** We initialize the generator from the pre-trained LDM with the fixed $t = 241$, which is the closest discrete timestep to the VE $\sigma = 1.0$. We use the class embedding of the generator and the fake score for conditioning on the target class. We do not add the class embedding for the source class. We set the regularization coefficient $\lambda = 0.02$ and train the model with the Adam optimizer with a generator's learning rate of $5 \cdot 10^{-5}$ and fake diffusion's learning rate of $1 \cdot 10^{-4}$. We perform one fake score update per generator update. We train the model for 6000 iterations with batch size 256. Training takes 1 day on $2\times$ NVidia Tesla A100 80GB.

We use the original LDM schedule

$$\beta_t = \left( \sqrt{\beta_{\min}} + \frac{T-t}{T}(\sqrt{\beta_{\max}} - \sqrt{\beta_{\min}}) \right)^2, \tag{73}$$

labeled as "linear" with $\beta_{\min} = 0.0015$ and $\beta_{\max} = 0.0195$ and $T = 1000$. We train the fake score on $\mathcal{L}_{\text{simple}}$ Ho et al. (2020) in the noise prediction parameterization. During training of the generator, we first sample VE $\sigma$ from the standard LogNormal distribution, then convert it into $\alpha = 1/(1+\sigma^2)$ and find the time step $t$ with the closest $\alpha_t = \prod_{s=1}^{t}(1 - \beta_s)$.

**DDIB hyperparameters** We run encoding in DDIB with the deterministic 100-step DDIM Song et al. (2020a) without classifier-free guidance (with unconditional guidance scale equal to 1.0). We run DDIB with decoding (unconditional) classifier-free guidance scale in $\{1.0, 1.5, 2.0\}$. We use the

hyperparameter choice of Wu & De la Torre (2023) and report the run with the best d-CLIP score, achieved with the guidance scale of $1.5$.

**CycleDiffusion hyperparameters**  We run CycleDiffusion with a grid of hyperparameters. As in DDIB, we choose the decoding unconditional guidance scale between $\{1.5, 2.0, 2.5\}$. We choose the encoding step $T_{es}$ in $[60, 70, 80, 90]$ in DDIM schedule with $\eta = 0.1$ and 100 steps, which results in $3T_{es}$ neural function evaluations due to the use of classifier-free guidance. We use the hyperparameter choice of Wu & De la Torre (2023) and report the run with the best d-CLIP score, achieved with the guidance scale of $2.5$ and $T_{es} = 80$.

### E.7 TEXT DETOXIFICATION EXPERIMENTS

**Experimental Setup.**  Although the dataset provides parallel data, we deliberately frame the task as unpaired to test a more challenging and realistic scenario. Consequently, our generator is trained exclusively on toxic sentences from the source domain and distribution matching signal from the target domain, *without* access to pairs. We employ Cosmos (Meshchaninov et al., 2025) as the foundational latent text diffusion model and train it on the conditional generation problem given the class label indicating whether a sentence is toxic or non-toxic. This model is used as the backbone for two-sided baselines (CycleDiff and DDIB). We also train text-conditional Cosmos† model on the paired dataset. In case of RDMD, we fix the non-toxic label and use this as our target DM and initialize the generator in the same way. As a cost function between the sequences of input and output latents, we use the length-averaged cost $c(\boldsymbol{x}, \boldsymbol{y}) = \|\frac{1}{L}\sum_{i=1}^{L} \boldsymbol{x}_i - \frac{1}{L}\sum_{i=1}^{L} \boldsymbol{y}_i\|_2$, which ignores singular latent perturbations and enforces similar semantic content between inputs and outputs. We set the regularization coefficient $\lambda$ to $0.5$.

**Metrics.**  For text detoxification experiments, we use the following metrics, proposed in (Logacheva et al., 2022):

- **Perplexity (ppl ↓):** Measures the fluency of the generated text. Lower is better.

- **BLEU ↑:** Measures the similarity to a ground-truth non-toxic reference, indicating content preservation.

- **Style Accuracy (STA) ↑:** The probability that the generated text is non-toxic, as determined by a style classifier.

- **Similarity (Sim) ↑:** The semantic similarity between the generated text and the original toxic input, measured by cosine similarity of sentence embeddings.

- **Fluency (Flu.) ↑:** Grammatical correctness and readability, as evaluated by a separate model.

- **J-score ↑:** A holistic metric combining Style Accuracy, Similarity, and Fluency.

## F ADDITIONAL COMPARISONS

### F.1 AFHQV2 EXPERIMENTS

We perform an additional visual comparison between the methods on $64 \times 64$ *Cat ↔ Wild* translation problems. To this end, we choose 8 random pictures from the source test data sets and report the corresponding outputs of RDMD and the baselines in Figure 12 and Figure 13. Here, we take RDMD with $\lambda = 0.1$ for both translation problems. As for the baselines, we choose the hyperparameters (see Appendix E.4) with the closest FID to RDMD as it was done in Table 2.

In Section 4.1 we compare the faithfulness-realism tradeoff achieved by RDMD and the baselines. In Figure 3 we report tradeoff in terms of FID and LPIPS for both translation problems. For the sake of completeness, in Figure 8 and Figure 9 we report trade-off in terms of 4 faithfulness metrics: $\sqrt{L_2}$, LPIPS, PSNR and SSIM. Qualitatively, we still see that our method beats all the baselines given at least moderate requirements on faithfulness.

### F.2 CELEBA EXPERIMENTS

We perform an additional visual comparison between the methods on $128 \times 128$ *Male $\leftrightarrow$ Female* translation problems. To this end, we choose 8 random pictures from the source test data sets and report the corresponding outputs of RDMD and the baselines in Figure 14 and Figure 15. Here, we take RDMD with $\lambda = 0.075$ for both translation problems. As for the baselines, we choose the hyperparameters (see Appendix E.5) with the closest FID to RDMD as it was done in Table 2.

For the sake of completeness, in Figure 10 and Figure 11 we report faithfulness-realism trade-off curves for the CelebA experiments in terms of 4 faithfulness metrics: $\sqrt{L_2}$, LPIPS, PSNR and SSIM. Qualitatively, we still see that our method beats all the baselines given at least moderate requirements on faithfulness.

### F.3 IMAGENET SAMPLES

In this section, we further verify applicability of RDMD in the multiclass translation.

First, we choose several pairs of classes, which were not present in the training dataset, but are somewhat meaningful to perform translation between. Specifically, we choose *Orange→Goldfish, Ladybug→Strawberry, Giant Panda $\rightarrow$ Totem Pole, Volcano $\rightarrow$ Totem Pole* and *Volcano $\rightarrow$ Water Jug* and report translation examples in Figures 16, 17, 18, 19, 20. Among them, Figure 16 and Figure 17 show that RDMD succeeds in translating between objects of different nature that are, however, similar in shape and color. Figure 18 demonstrates ¡¡stylization¿¿ of an object. Finally, Figures 19 and 20 demonstrate RDMD's successful applicability even in case of extremal mismatch between the domains. Specifically, it preserves such characterizing traits of the target domain as the refraction of light that passes through the water jug.

Finally, in Figures 21, 22, 23, 24 we present examples of *all-to-all* translation between all pairs of classes in a benchmark.

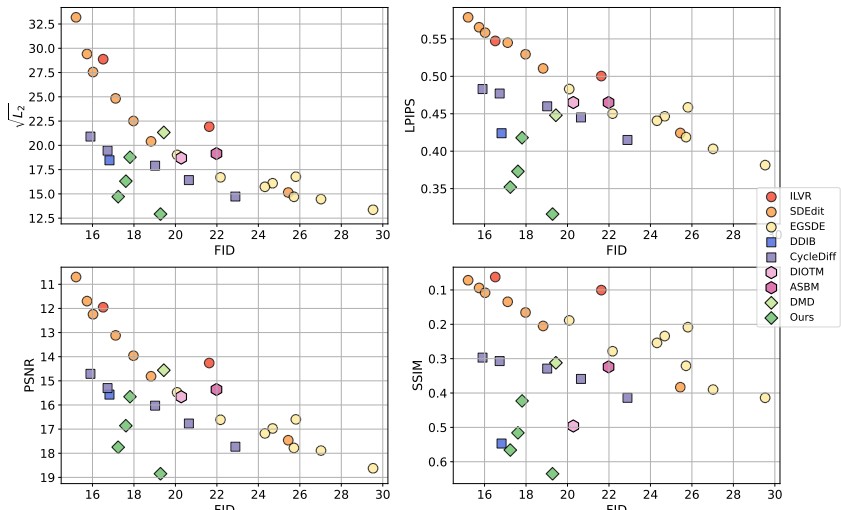

Figure 8: Comparison of RDMD with the baselines on $64 \times 64$ AFHQv2 *Wild* $\rightarrow$ *Cat* translation problem. The figure demonstrates the tradeoff between generation quality (FID↓) and the difference between the input and output (L2↓, LPIPS↓, PSNR↑, SSIM↑). RDMD gives an overall better trade-off given fairly strict requirements on the transport cost. In the cases of PSNR and SSIM, the $y$-axis is swapped for the sake of identical readability with the first plot (left is better, low is better).

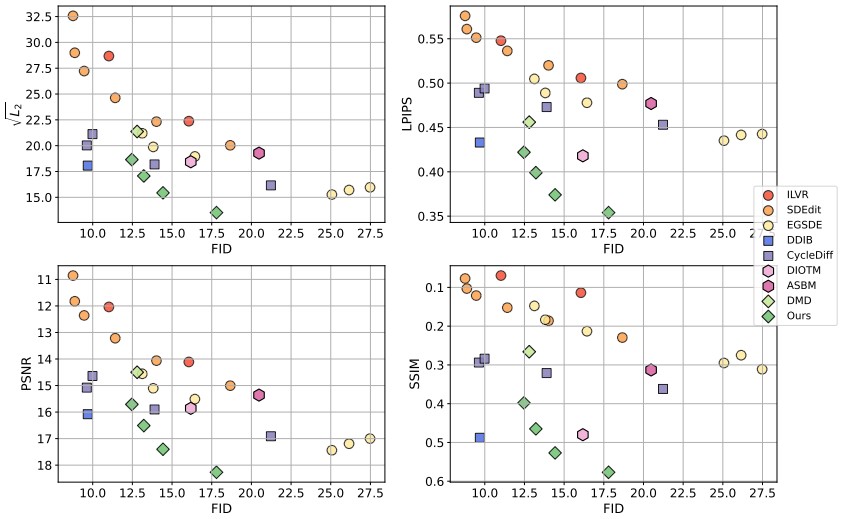

Figure 9: Comparison of RDMD with the baselines on $64 \times 64$ AFHQv2 *Cat* $\rightarrow$ *Wild* translation problem. The figure demonstrates the tradeoff between generation quality (FID↓) and the difference between the input and output (L2↓, LPIPS↓, PSNR↑, SSIM↑). RDMD gives an overall better trade-off given fairly strict requirements on the transport cost. In the cases of PSNR and SSIM, the $y$-axis is swapped for the sake of identical readability with the first plot (left is better, low is better).

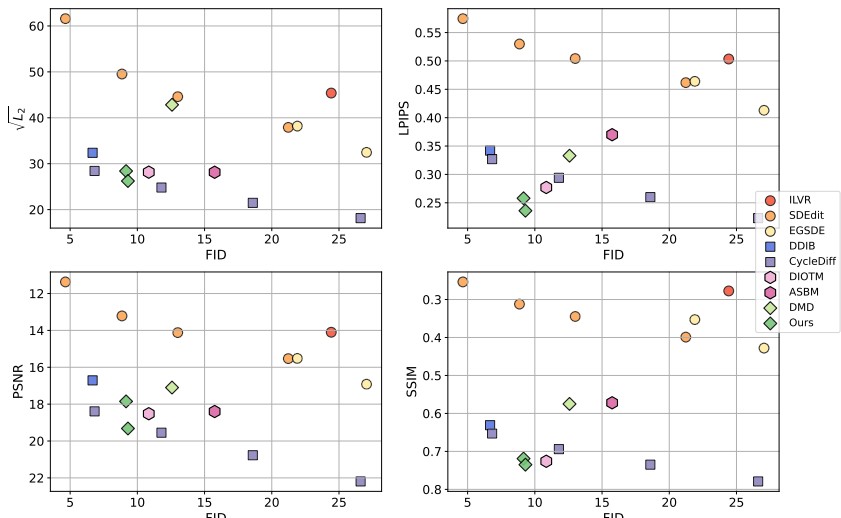

Figure 10: Comparison of RDMD with the baselines on $128 \times 128$ CelebA *Male → Female* translation problem. The figure demonstrates the tradeoff between generation quality (FID↓) and the difference between the input and output (L2↓, LPIPS↓, PSNR↑, SSIM↑). RDMD achieves an overall better quality given fairly strict requirements on the transport cost. In the cases of PSNR and SSIM, the $y$-axis is swapped for the sake of identical readability with the first plot (left is better, low is better).

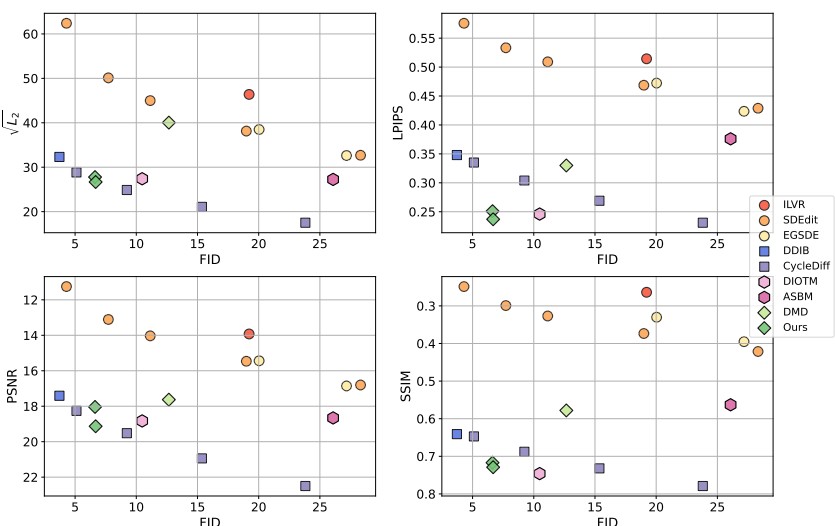

Figure 11: Comparison of RDMD with the baselines on $128 \times 128$ CelebA *Female → Male* translation problem. The figure demonstrates the tradeoff between generation quality (FID↓) and the difference between the input and output (L2↓, LPIPS↓, PSNR↑, SSIM↑). RDMD achieves an overall better quality given fairly strict requirements on the transport cost. In the cases of PSNR and SSIM, the $y$-axis is swapped for the sake of identical readability with the first plot (left is better, low is better).

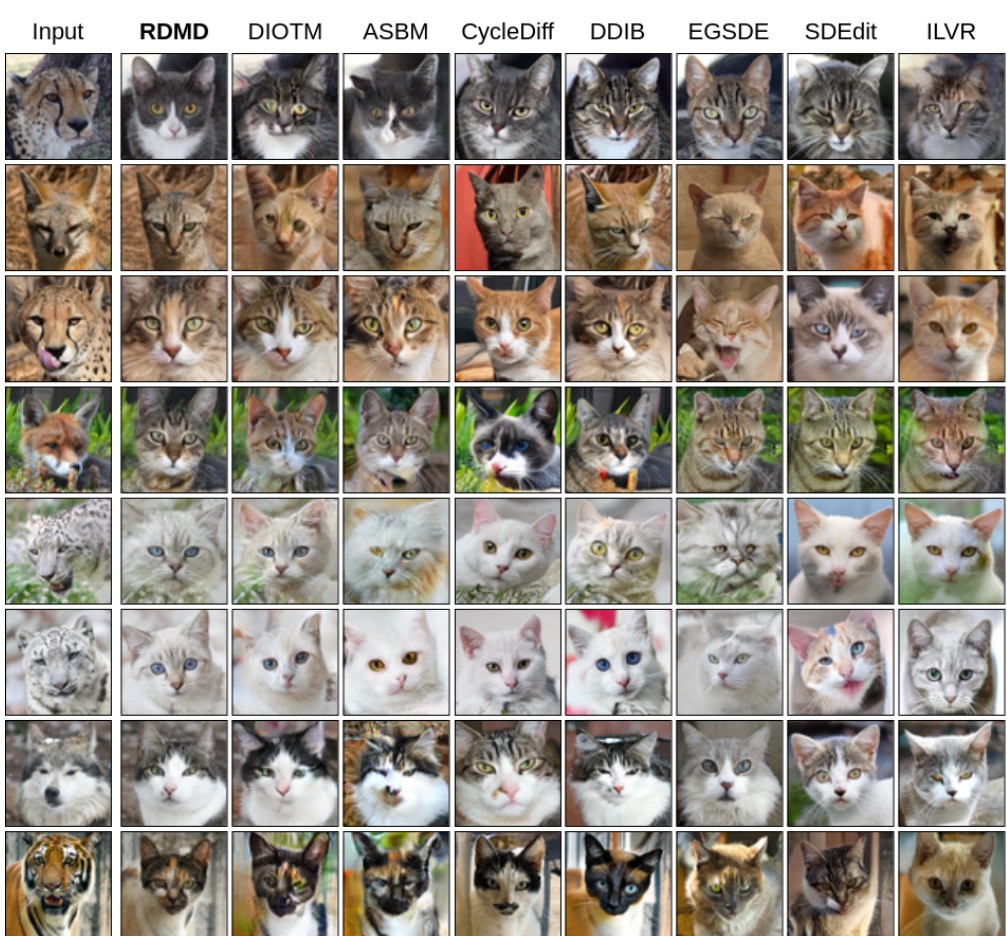

Figure 12: Visual comparison of RDMD with the baselines on $64 \times 64$ AFHQv2 *Wild $\rightarrow$ Cat* translation problem. Source images are chosen randomly from the test data set.

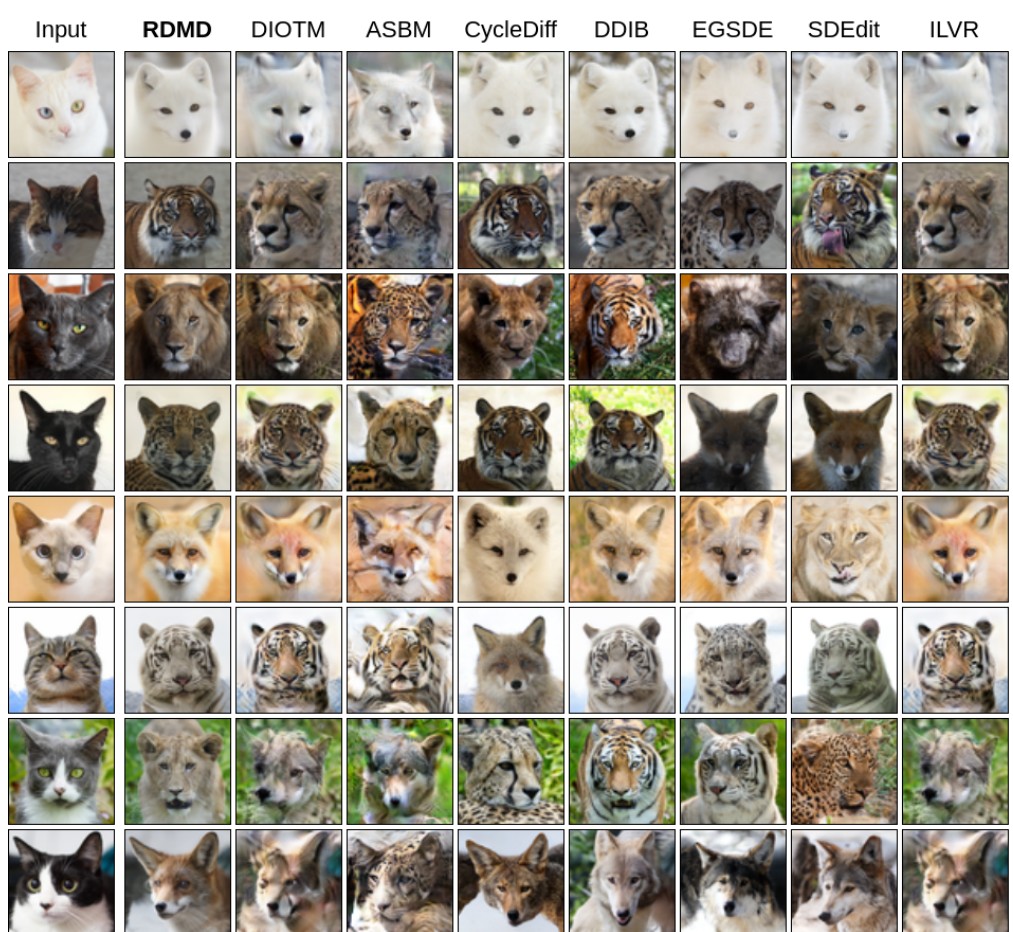

Figure 13: Visual comparison of RDMD with the baselines on $64 \times 64$ AFHQv2 *Cat* $\rightarrow$ *Wild* translation problem. Source images are chosen randomly from the test data set.

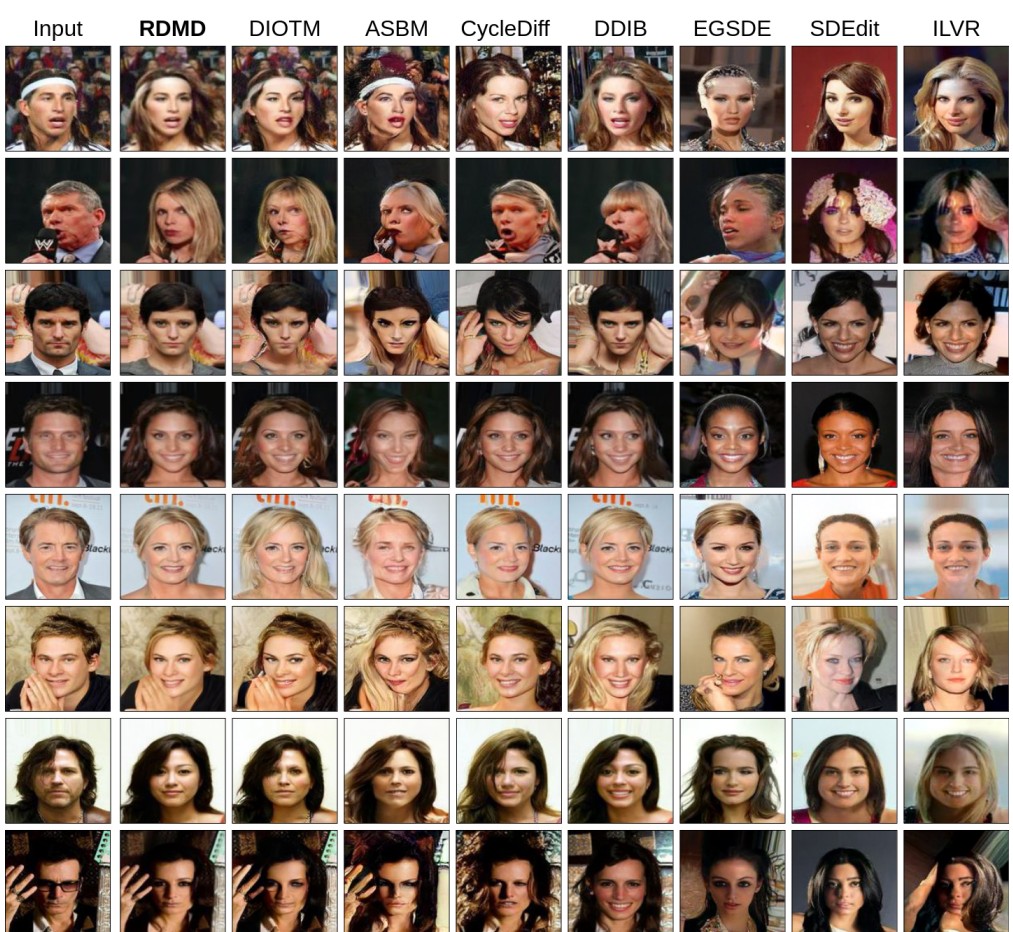

Figure 14: Visual comparison of RDMD with the baselines on $128 \times 128$ CelebA *Male → Female* translation problem. Source images are chosen randomly from the test data set.

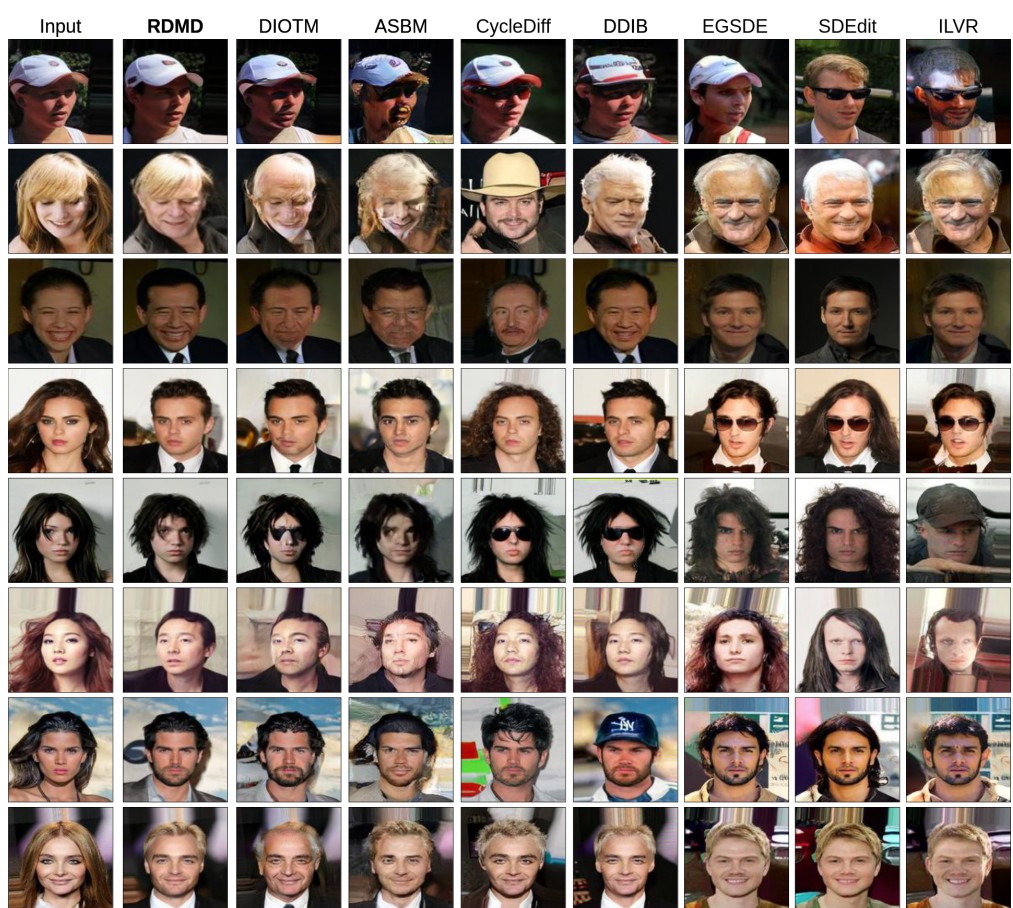

Figure 15: Visual comparison of RDMD with the baselines on $128 \times 128$ CelebA *Female → Male* translation problem. Source images are chosen randomly from the test data set.

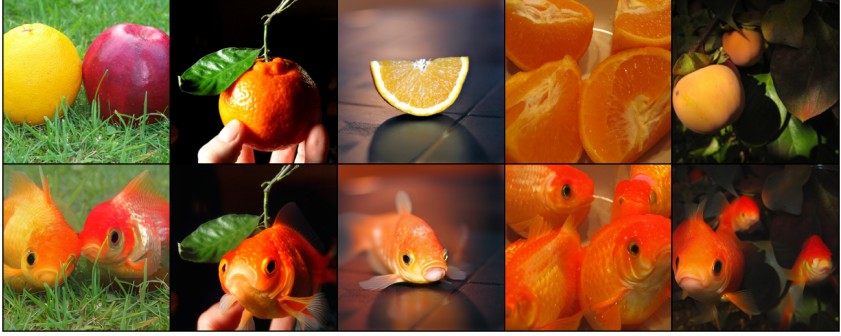

Figure 16: Example of RDMD ImageNet Orange → Goldfish Translation

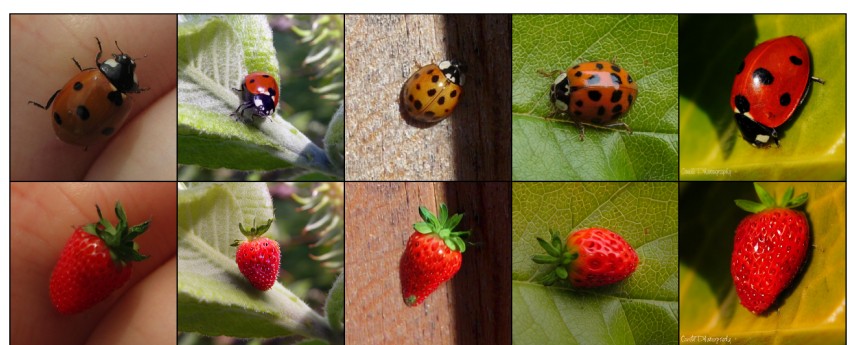

Figure 17: Example of RDMD ImageNet Ladybug → Strawberry Translation

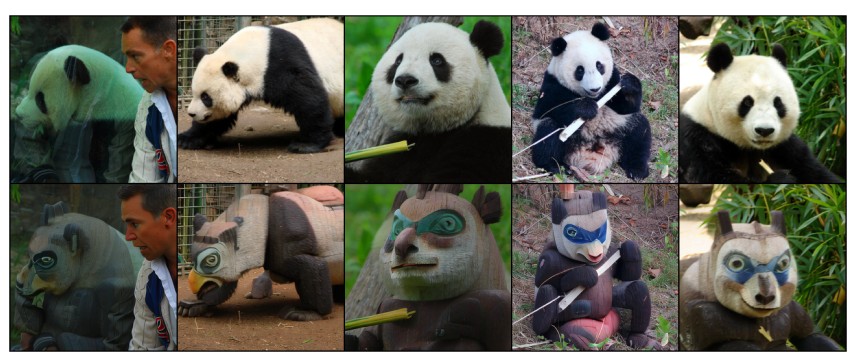

Figure 18: Example of RDMD ImageNet Giant Panda → Totem Pole Translation

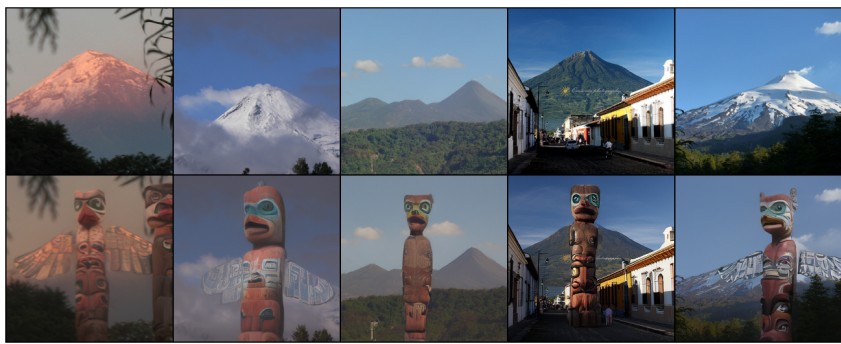

Figure 19: Example of RDMD ImageNet Volcano → Totem Pole Translation

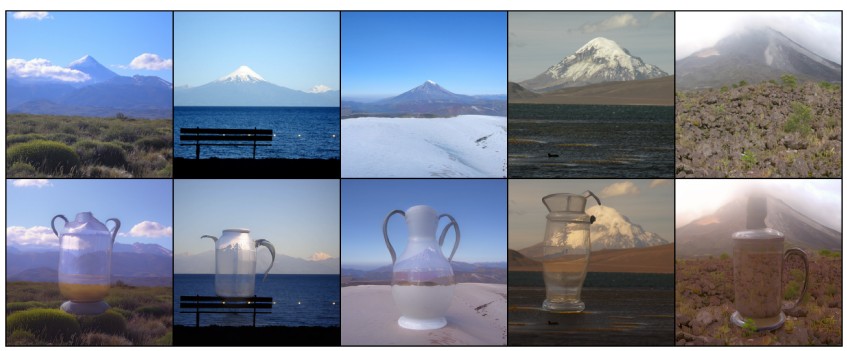

Figure 20: Example of RDMD ImageNet Volcano → Water Jug Translation

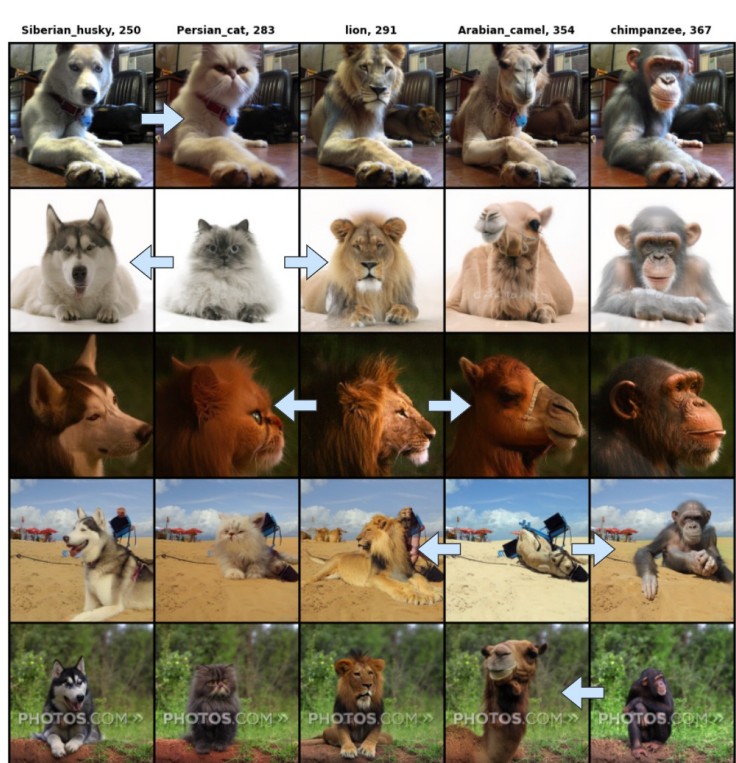

Figure 21: RDMD translation between all pairs of *Animal* classes.

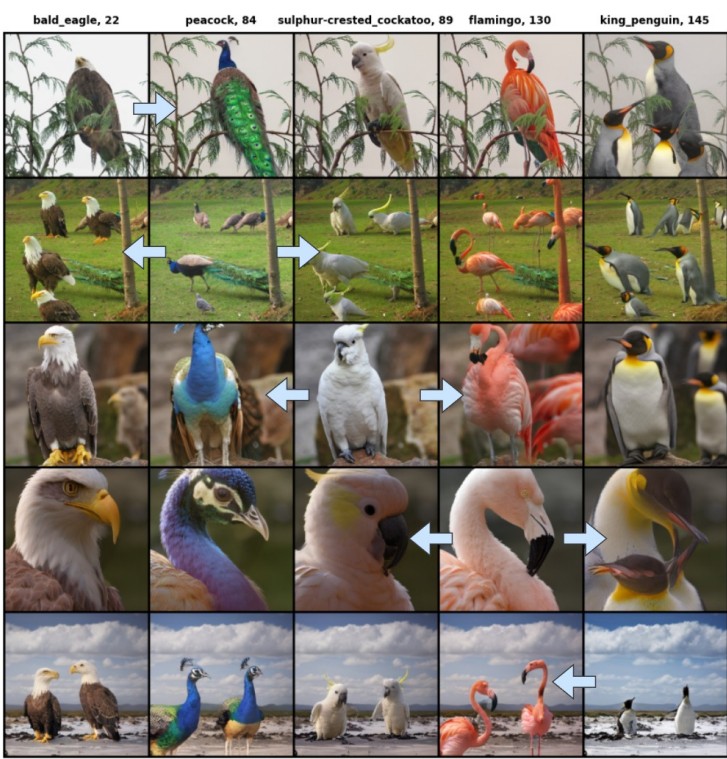

Figure 22: RDMD translation between all pairs of *Birds* classes.

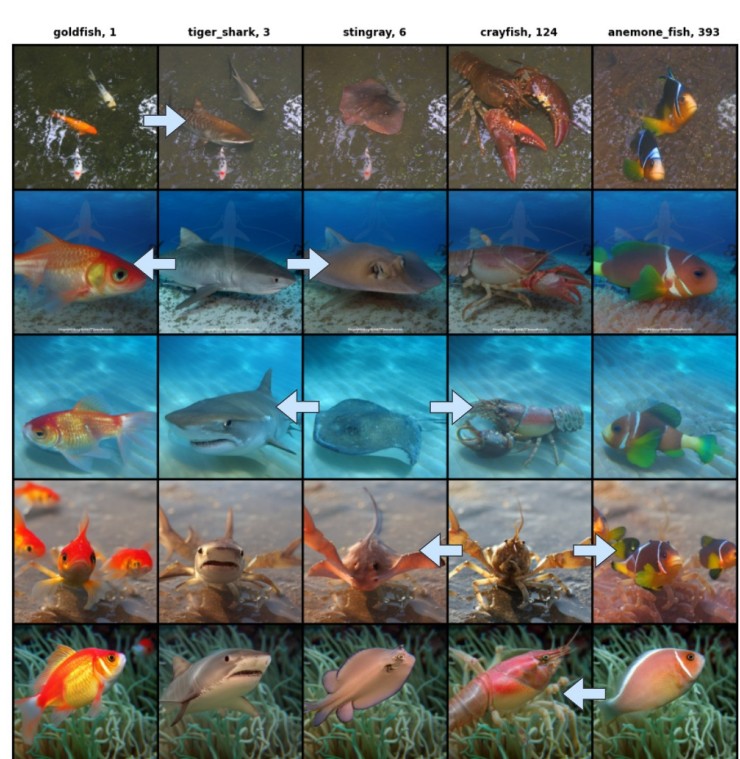

Figure 23: RDMD translation between all pairs of *Fish* classes.

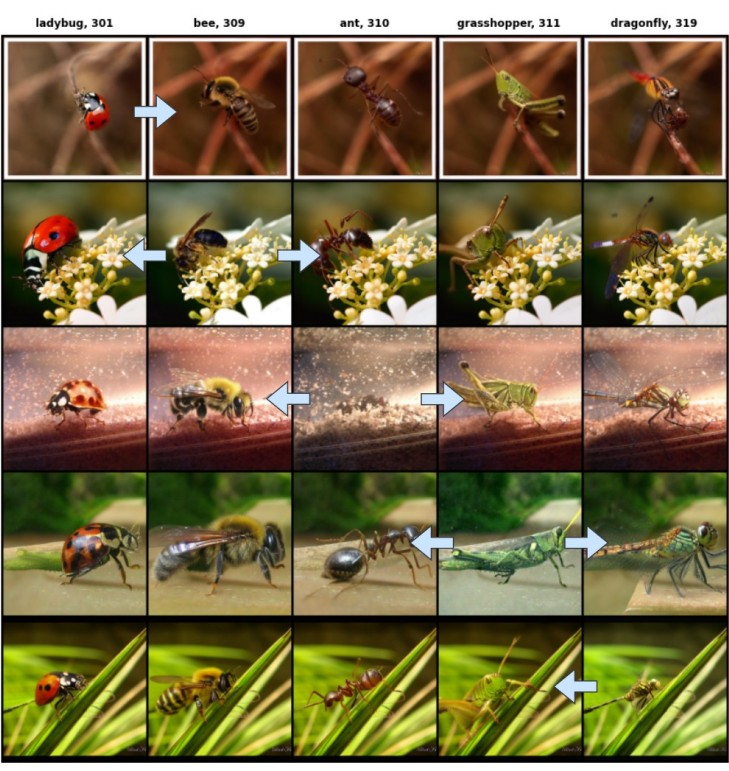

Figure 24: RDMD translation between all pairs of *Insects* classes.

Table 7: Quantitative comparison of RDMD with GAN-based and diffusion-based baselines. The results are taken from EGSDE (Zhao et al., 2022) and CycleDiffusion (Wu & De la Torre, 2023) papers and are obtained via taking them directly from the corresponding papers or running the publicly available code.

| Model | FID ↓ | L2 ↓ | PSNR ↑ | SSIM ↑ |
|---|---|---|---|---|
| AFHQv2 $256 \times 256$ Cat $\rightarrow$ Dog | | | | |
| CycleGAN (Zhu et al., 2017) | 85.9 | - | - | - |
| StarGAN v2 (Choi et al., 2020) | 54.88 | 133.65 | 10.63 | - |
| CUT (Park et al., 2020) | 76.21 | 59.78 | 17.48 | 0.601 |
| ITTR (CUT) (Zheng et al., 2022) | 68.6 | - | - | - |
| ILVR (Choi et al., 2021) | 74.37 | 56.95 | 17.77 | 0.363 |
| SDEdit (Meng et al., 2021) | 74.17 | 47.88 | 19.19 | 0.423 |
| EGSDE (Zhao et al., 2022) | 65.82 | 47.22 | 19.31 | 0.415 |
| EGSDE$^{\dagger}$ (Zhao et al., 2022) | 51.04 | 62.06 | 17.17 | 0.361 |
| CycleDiffusion (Wu & De la Torre, 2023) | 58.87 | - | 18.50 | 0.557 |
| RDMD ($\lambda = 0.2$) | **44.32** | 64.44 | 16.91 | 0.439 |

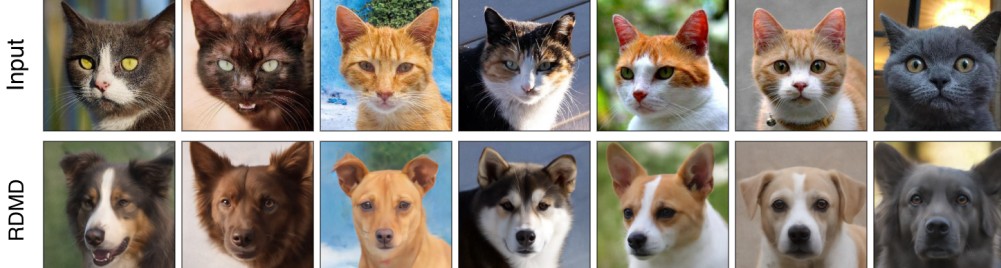

Figure 25: Visual performance of RDMD on AFHQv2 $256 \times 256$ *Cat $\leftrightarrow$ Dog* translation task in pixel space.

## G    ADDITIONAL EXPERIMENTS

### G.1    SCALING TO $256 \times 256$

We perform an additional comparison of RDMD with GAN-based and diffusion-based baselines on AFHQv2 Cat→Dog in $256 \times 256$ pixel space. We use the same ADM (Dhariwal & Nichol, 2021) diffusion model as in ILVR, SDEdit, EGSDE and CycleDiffusion as the backbone for RDMD. We train two RDMD models with $\lambda = 0.2$ and $\lambda = 0.3$. We present quantitative results in Table 7. Notably, RDMD trained with $\lambda = 0.2$ outperforms all the baselines in terms of FID. In particular, compared with EGSDE† and CUT at a similar faithfulness level (L2 in $[59.0, 65.0]$), RDMD has a significantly lower FID. We demonstrate RDMD's visual performance in Figure 25. TODO: add experimental details.

### G.2    CHANGING COST FUNCTION IN PIXEL SPACE

We perform additional experiments using the perceptual distance (LPIPS) as the cost function instead of L2. We test its performance on $64 \times 64$ AFHQv2 Wild→Cat translation problem, where we train 3 models with $\lambda \in \{0.1, 0.2, 0.3\}$. As in Section 4.1, we report faithfulness-realism trade-off curves with respect to L2, PSNR, SSIM and LPIPS in Figure 26. We observe that the LPIPS models achieve considerably worse trade-off in terms of pixel-based L2 and PSNR, worse trade-off in terms of SSIM and comparable trade-off in terms of LPIPS (with the exception of the largest $\lambda = 0.3$, where LPIPS is lower, which is expected), which indicates that in this translation problem L2 loss performs well at capturing semantic alignment. We report the corresponding metrics in Table 8,

Table 8: Quantitative comparison of RDMD with L2 and LPIPS transport costs with the baselines on $64 \times 64$ AFHQv2 Wild→Cat translation problem.

| Cost | FID ↓ | L2 ↓ | LPIPS ↓ | PSNR ↑ | SSIM ↑ |
|---|---|---|---|---|---|
| L2 ($\lambda = 0.1$) | **17.31** | **14.71** | 0.352 | **17.75** | **0.566** |
| LPIPS ($\lambda = 0.1$) | **17.17** | 19.49 | **0.361** | 15.47 | 0.473 |

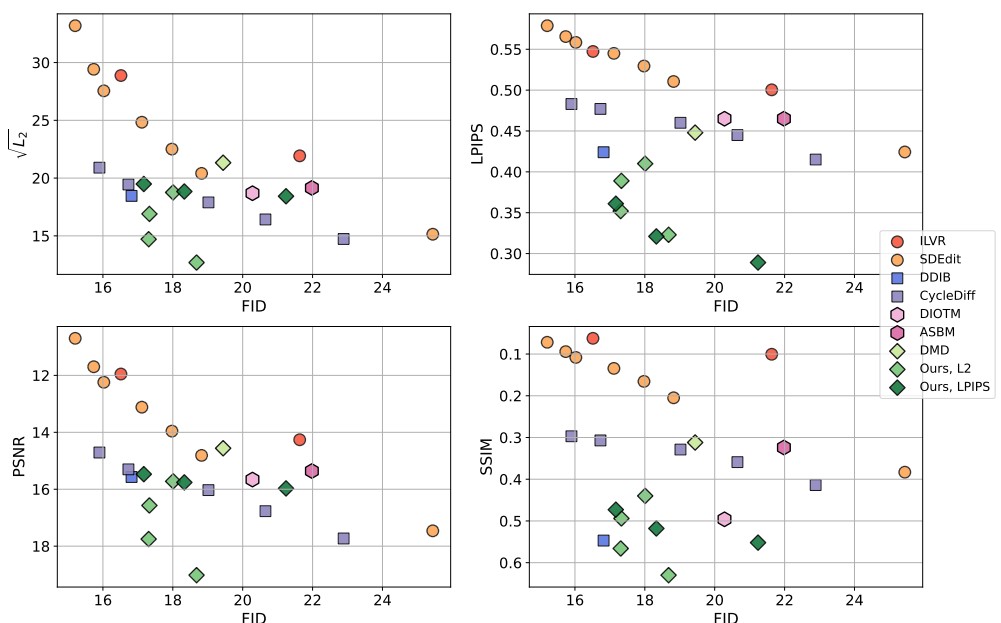

Figure 26: Comparison of RDMD with L2 and LPIPS transport costs with the baselines on $64 \times 64$ AFHQv2 Wild→Cat translation problem. The figure demonstrates the tradeoff between generation quality (FID↓) and the difference between the input and output (L2↓, LPIPS↓, PSNR↑, SSIM↑). In the cases of PSNR and SSIM, the y-axis is swapped for the sake of identical readability with the first plot (left is better, low is better).

where we choose $\lambda = 0.1$ for L2 and choose the LPIPS baseline with the closest FID, also achieved at $\lambda = 0.1$. In addition, we perform visual comparison between transport costs in Figure 27, where we observe only slight difference in the models' performance.

In addition to Wild→Cat, we compare L2 and LPIPS in the case, where L2 does not appropriately capture the desired alignment. To this end, we consider the unpaired translation problem from the color-inverted MNIST to MNIST and aim to learn the inversion function $f(x) = 1 - x$ that swaps black and white pixels. We visualize the performance of both models in Figure 28, where we observe that L2-RDMD fails at performing the desired translation due to the confrontation of the corresponding objectives. At the same time, LPIPS' ability of edge and structure detection allows for the proper translation between the domain and its inverse without any knowledge about the problem except the unpaired data sets of samples.

### G.3 LATENT-SPACE VS PIXEL-SPACE L2

In this experiment, we compare the performance of the latent-space RDMD model equipped with pixel-space and latent-space L2 transport cost. We choose the corresponding regularization coefficients in such a way to ensure similar performance in terms of realism. We report the quantitative comparison in Table 9 and visual comparison in Figure 29.

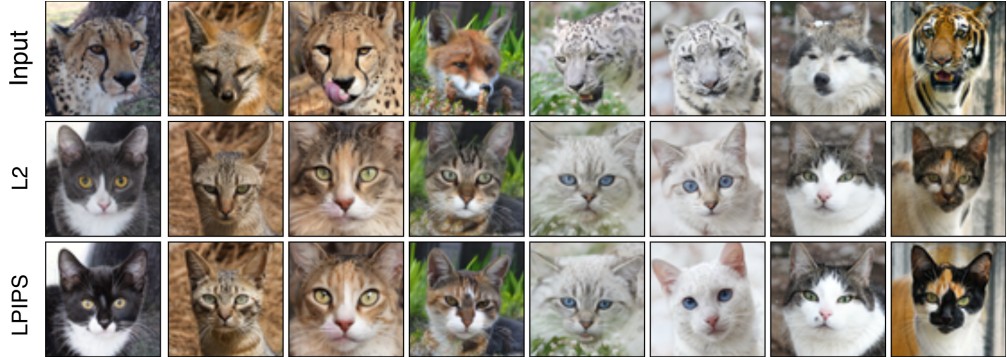

Figure 27: Visual performance of RDMD with L2 and LPIPS transport cost on AFHQv2 $64 \times 64$ *Wild ↔ Cat* translation task in pixel space.

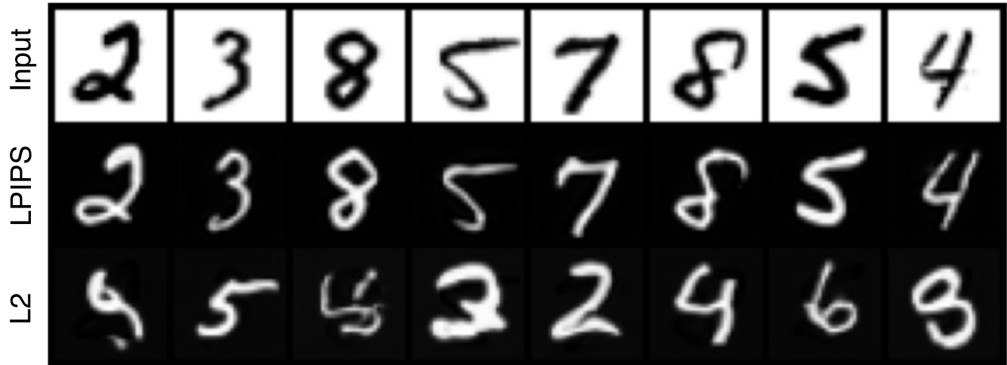

Figure 28: Visual performance of RDMD with L2 and LPIPS transport cost on Inverted MNIST→MNIST translation problem.

We observe that at similar FID levels, the model with pixel-space L2 performs slightly better in terms of per-pixel similarity, but performs worse in terms of the other metrics. Qualitatively, the generated faces are similar, which can be explained by the similar structure of the pixel space and the latent space of diffusion autoencoders, which work akin to the super-resolution models. Both choices offer competitive translation quality, but pixel-space L2 requires backpropagating through both encoder and decoder, which works approximately $1.5\times$ slower. We thus recommend to calculate L2 in the latent space to obtain faster convergence and slightly better semantic alignment.

G.4 COMPARISON WITH ZERO-SHOT FOUNDATION MODELS

In this subsection, we compare RDMD with the foundation multi-billion NanoBanana model on class-to-class ImageNet translation. To use NanoBanana, we formulate each translation problem as an image editing problem using the following prompt (with placeholders replaced by corresponding classes): *"Transform SOURCE CLASS into TARGET CLASS. Keep the scene, background, lighting,*

Table 9: Quantitative comparison of RDMD with pixel-space and latent-space L2 on $256 \times 256$ CelebA-HQ Male→Female translation problem.

| | FID ↓ | L2↓ | LPIPS ↓ | PSNR ↑ | SSIM ↑ |
|---|---|---|---|---|---|
| Latent L2 | 32.11 | 54.75 | 0.339 | 21.96 | 0.606 |
| Pixel L2 | 33.68 | 52.92 | 0.413 | 18.66 | 0.548 |

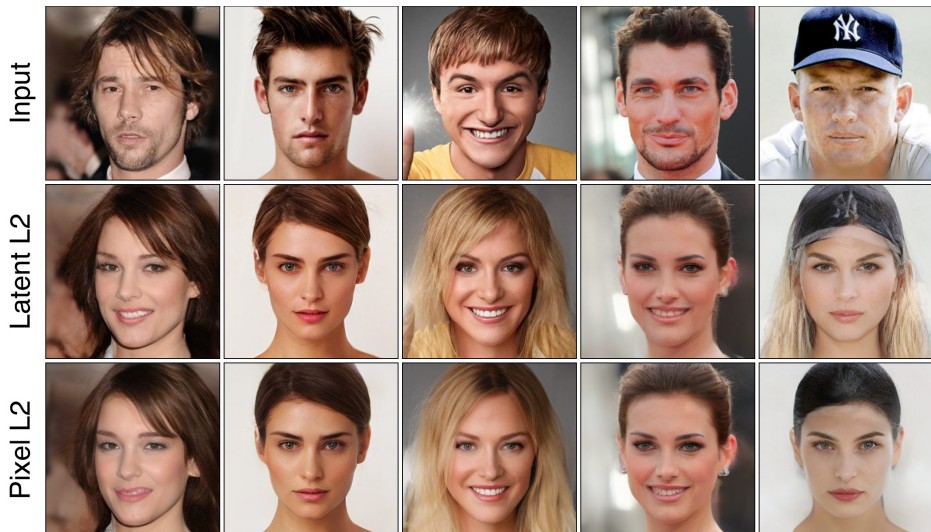

Figure 29: Visual performance of latent RDMD with L2 applied in pixel and latent space on 256 × 256 CelebA-HQ Male→Female experiment.

Table 10: Quantitative comparison of RDMD with zero-shot foundation NanoBanana model on ImageNet multiclass translation benchmarks.

| Model | Animals | | Birds | | Fish | | Insects | |
|---|---|---|---|---|---|---|---|---|
| | FID | LPIPS | FID | LPIPS | FID | LPIPS | FID | LPIPS |
| RDMD | **39.85** | **0.369** | **24.87** | 0.415 | **34.00** | 0.329 | **29.57** | 0.296 |
| NanoBanana | 42.29 | 0.386 | 37.96 | **0.331** | 52.38 | **0.280** | 39.93 | **0.231** |

*colors, camera viewpoint, and all non-SOURCE CLASS objects unchanged. For every transformation, preserve the original position, scale, pose, and interactions."*

We report the quantitative comparison in Table 10 and compare visual performance in Figure 30. Here, we observe that RDMD beats NanoBanana in terms of FID, but typically has worse input-output alignment measured in LPIPS. In this problem, a better FID does not directly imply better realism, since the desired translation may, e.g., place realistic animals in non-realistic environments, which leads to a bias compared to the original distribution. These scenarios are present for both models but are more pronounced for NanoBanana due to its stricter alignment with the input object shapes.

In Figure 30 we observe that NanoBanana typically excels at alignment and has high realism. In the case of the Animals benchmark, however, RDMD outperforms it in both metrics, which could be partially justified by the corresponding visual comparison, where RDMD has comparable realism and better preserves the overall pose of the animal. Overall, it visually seems that NanoBanana outperforms RDMD in terms of editing performance (the alignment is better, and the generated object is realistic). However, RDMD still offers competitive performance in terms of both metrics and visual quality. At the same time, NanoBanana is multi-billion commercial foundation model, while ImageNet RDMD has 400M parameters, takes less than 5 GPU-days of training given the pre-trained LDM, and offers efficient one-step inference with 3 total neural network evaluations (encode, translate, decode).

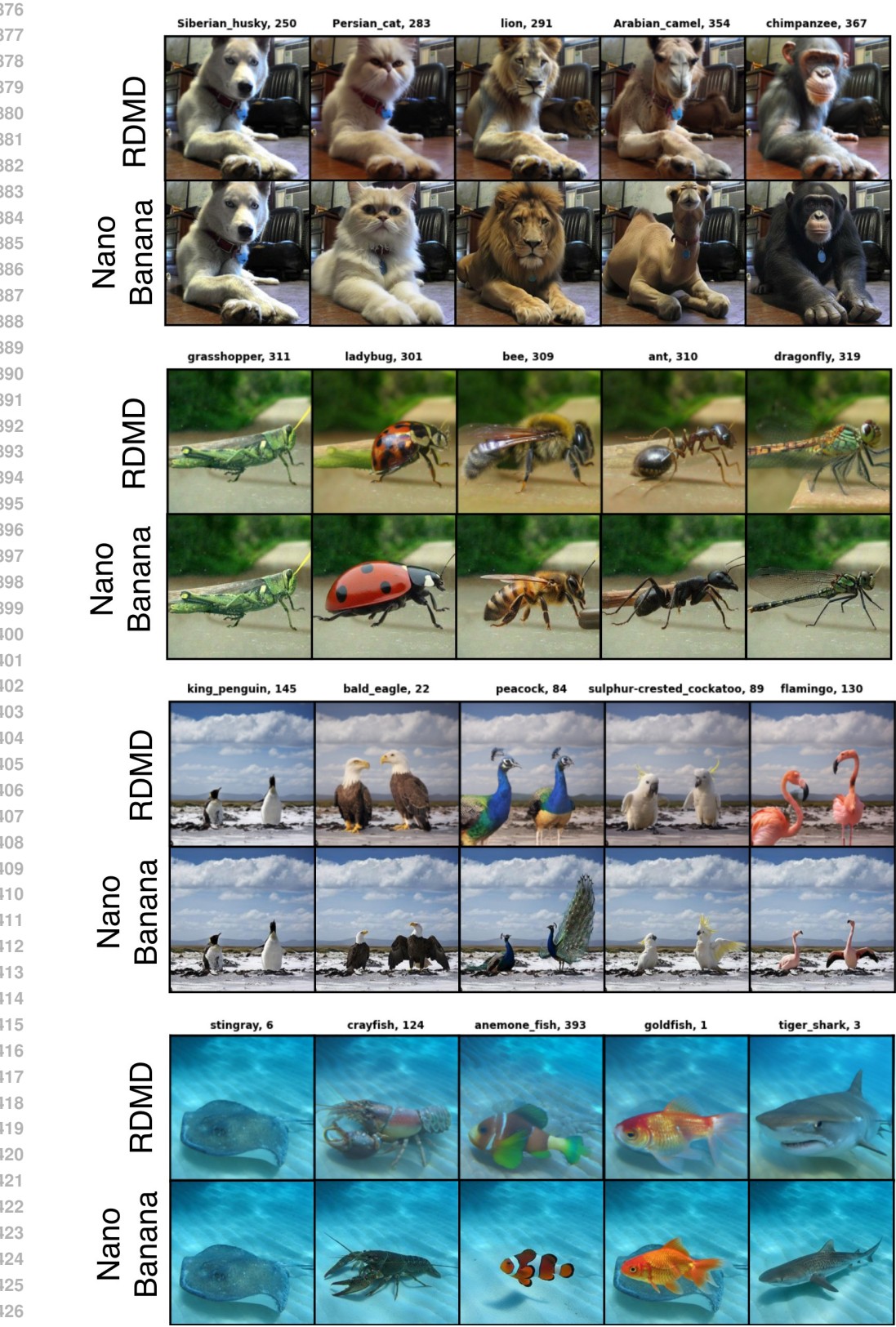

Figure 30: Visual performance of RDMD compared to zero-shot foundation NanoBanana model. Leftmost column denotes input image.

## G.5 Latent-space ImageNet-512 Experiment

**Experimental setup.** We train a single latent-space RDMD model to perform a class-conditional translation between all possible pairs of the following ImageNet classes: Siberian Husky, Persian Cat, Lion, Arabian Camel, Chimpanzee. Within this experiment, we label this dataset as *Animals*.

**Architecture.** We use the pre-trained class-conditional EDM2 (Karras et al., 2024) model with 498M parameters corresponding to `edm2-img512-m-guid-fid` configuration, which operates in the latent space of the Stable Diffusion (Rombach et al., 2022) VAE (namely, `sd-vae-ft-mse`) with dimension $4 \times 64 \times 64$. It achieves FID=2.01 with a classifier-free guidance scale equal to 1.2.

We use this model as the pre-trained target score and use it as initialization for both the generator and the fake score. We use the classifier-free guidance coefficient of 1.2 for the target score during the training.

**Training RDMD.** We initialize the generator from the pre-trained latent model with the fixed $\sigma = 0.4$. The generator is conditioned only on the target class. We set the regularization coefficient $\lambda = 0.02$ and train the generator with the Adam optimizer with a learning rate of $5 \cdot 10^{-5}$. The same learning rate is used for training the fake score. We perform 5 fake score updates per one generator update. We train the model for 17K iterations with a batch size of 256. Training takes 3 days on $1\times$ NVidia Tesla H100. We train the fake score using EDM2 (Karras et al., 2024) parametrization, which includes uncertainty estimation. We unset the uncertainty estimation layers for the generator since they do not participate in training.

**Visual Results.** We provide a curated qualitative evaluation of our model in Figure 31.

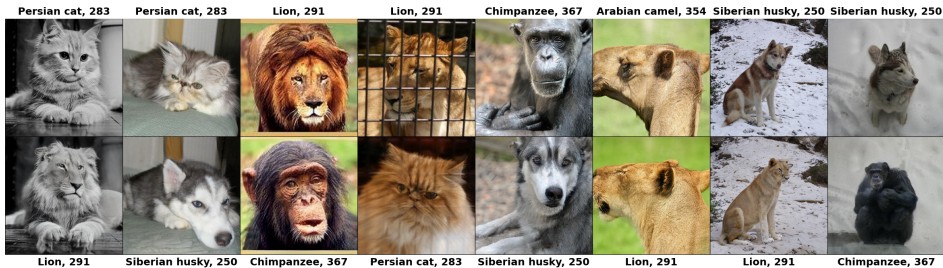

Figure 31: One-step translation between different ImageNet *Animals* classes with RDMD.

