# OpenReview forum: "One-step Optimal Transport via Regularized Distribution Matching Distillation"
_ICLR.cc/2026/Conference — Submitted to ICLR 2026_

### Official Review · Reviewer_cLN3 · 2025-10-28

**Soundness:** 3
**Presentation:** 3
**Contribution:** 2
**Rating:** 2
**Confidence:** 3

**Summary:**

This paper proposes Regularized Distribution Matching Distillation (RDMD) for unpaired image-to-image translation, adding a quadratic transport-cost regularizer to DMD to preserve input–output structural correspondence. RDMD enables one-step generation with a faithfulness–realism trade-off, showing competitive performance on multiple benchmarks.

**Strengths:**

-  The writing of the paper is clear and fluent.

-  Applying DMD to efficiently tackle unpaired I2I translation is an interesting approach.

**Weaknesses:**

- Limited resolution benchmarks. Experiments are restricted to low resolutions (64×64/128×128). It is unclear whether the approach scales to ≥ 256×256, where input–output alignment typically become more challenging.

- Overly simple regularization. The transport-cost regularizer is instantiated as plain L2, which is often insufficient to enforce semantic correspondence between input and output. This choice may limit the method’s expressiveness under complex cross-domain shifts.

- Have the authors considered using feature-level perceptual regularization[1] instead of pixel-space L2, for example by leveraging pretrained critics or reward models to better enforce semantic or structural details between the source and the one-step outputs?

[1] Li M, Yang T, Kuang H, et al. Controlnet++: Improving conditional controls with efficient consistency feedback

**Questions:**

see above

---

> ### Author Response · Authors · 2025-11-21
> **Weakness 1**
>
> Thank you for your comments and feedback! Below, we address your raised weaknesses:
>
> **Weakness 1.**
> > Limited resolution benchmarks. Experiments are restricted to low resolutions (64×64/128×128). It is unclear whether the approach scales to ≥ 256×256, where input–output alignment typically become more challenging.
>
> We note that besides $64\times 64$ and $128 \times 128$ pixel-space experiments, our paper **includes** $\mathbf{256 \times 256}$ latent-space experiment on ImageNet class-to-class translation. Moreover, we conduct the unpaired text detoxification experiment, which demonstrates applicability of the model on different domains. The results on the ImageNet benchmark show that the model is capable of preserving both input-output alignment and output realism sufficiently well.
>
> To further demonstrate the scalability of RDMD, we conduct an additional experiment: $256 \times 256$ pixel-space Cat$\to$Dog translation problem. We obtain the following results:
>
> | **Model**              | **FID ↓** | **L2 ↓** | **PSNR ↑** | **SSIM ↑** |
> | ---------------------- | --------- | -------- | ---------- | ---------- |
> | CycleGAN              | 85.9      | -        | -          | -          |
> | StarGANv2              | 54.88     | 133.65   | 10.63      | -          |
> | CUT                   | 76.21     | 59.78    | 17.48      | 0.601      |
> | ITTR (CUT)            | 68.6      | -        | -          | -          |
> | ILVR                   | 74.37     | 56.95    | 17.77      | 0.363      |
> | SDEdit                 | 74.17     | 47.88    | 19.19      | 0.423      |
> | EGSDE                  | 65.82     | 47.22    | 19.31      | 0.415      |
> | EGSDE$^{\dagger}$      | 51.04     | 62.06    | 17.17      | 0.361      |
> | CycleDiffusion         | 58.87     | -        | 18.50      | 0.557      |
> | RDMD ($\lambda = 0.2$) | **44.32** | 64.44    | 16.91      | 0.439      |
>
> RDMD significantly outperforms the baselines in terms of FID. Compared to the diffusion-based baselines, it achieves the comparable level of alignment as e.g. EGSDE$\dagger$, but has better realism. Due to
> the limitations in resources, we were able to train this RDMD checkpoint for only 20 hours on 4× H100 GPU. It
> did not fully converge but still shows the best FID among the baselines. We report additional details and visual
> performance of RDMD in Appendix G.1 and Figure 25 of the revised version of the paper (page 41).

---

> > ### Author Response · Authors · 2025-11-21
> >
> > **Weaknesses 2 and 3.**
> >
> > > Overly simple regularization. The transport-cost regularizer is instantiated as plain L2, which is often insufficient to enforce semantic correspondence between input and output. This choice may limit the method’s expressiveness under complex cross-domain shifts.
> >
> > > Have the authors considered using feature-level perceptual regularization[1] instead of pixel-space L2...?
> >
> > **Wild$\to$Cat w/ LPIPS**
> >
> > We have added an additional experiment on Wild$\to$Cat $64 \times 64$ benchmark to show the difference between using L2 and perceptual distances as cost functions. We choose LPIPS as the perceptual distance and report the results in the table below and in the revised version of the paper.
> >
> > | **Cost**                | **FID ↓** | **L2 ↓**  | **LPIPS ↓** | **PSNR ↑** | **SSIM ↑** |
> > | ----------------------- | --------- | --------- | ----------- | ---------- | ---------- |
> > | L2 ($\lambda = 0.1$)    | 17.31 | **14.71** | **0.352**   | **17.75**  | **0.566**  |
> > | LPIPS ($\lambda = 0.1$) | **17.17** | 19.49     | 0.361       | 15.47      | 0.473      |
> >
> > Overall, using LPIPS in this experiment leads to a worse alignment in terms of all similarity metrics (except LPIPS, where the results are comparable) under comparable realism. The overall trade-off achieved by LPIPS is worse in almost all similarity metrics as indicated by Figure 26 in Appendix G.2. Visually, we see little difference between the two costs (please see Appendix G.2, Figure 27 for the visual comparison). This prompts to the fact that in this and the similar benchmarks L2 works sufficiently well at preserving the semantic similarity.
> >
> > Additionally, we would like to mention that training models with L2 cost is the well-established choice in the OT literature. Such classic methods as OTM[1], UOTM[2] and NOT[3] use L2 cost explicitly. Moreover, all the Schrödinger Bridge based methods (e.g. DSBM [4], ASBM [5]) implicitly optimize the L2 cost due to its connection with L2-based entropic optimal transport.
> >
> > **Latent-space L2**
> >
> > Besides the additional experiment, please note that the paper contains latent-space experiments (ImageNet class-to-class translation and text detoxification). Here, the L2 cost is measured in the latent space, which corresponds to a different cost function when considering the translation problem in the initial space. To demonstrate the difference between pixel and latent-space L2, we perform the additional experiment, suggested by the reviewer **wb1N**. Specifically, we consider $256 \times 256$ CelebA-HQ Male$\to$Female translation experiment in the $4 \times 32 \times 32$ latent space, corresponding to the LDM autoencoder. We train two versions of the RDMD model: the first measures L2 in the latent space, the second measures L2 in the pixel space after decoding.
> >
> > |               | **FID ↓** | **L2 ↓** | **LPIPS ↓** | **PSNR ↑** | **SSIM ↑** |
> > | ------------- | --------- | -------- | ----------- | ---------- | ---------- |
> > | **Latent L2** | 32.11     | 54.75    | 0.339       | 21.96      | 0.606      |
> > | **Pixel L2**  | 33.68     | 52.92    | 0.413       | 18.66      | 0.548      |
> >
> > Though models perform similar, the per-pixel similarity for the latent-space L2 model is slightly worse, but other similarity metrics (including LPIPS) are better. Visually (as indicated by Figure 29, Appendix G.3. in the revised paper), the results are similar, but the facial expression alignment seems to be slightly better in the case of the latent model. This experiment shows that performing L2 in the latent space of the diffusion autoencoder may be considered as a more structure-preserving transport cost than the pixel-space L2.
> >
> > **Inverted MNIST**
> >
> > Despite the Wild$\to$Cat experiment, we agree that there may be scenarios where L2 cannot provide a meaningful signal for a good input-output alignment, whereas LPIPS can. To show this, we consider the unpaired translation problem from the color-inverted MNIST to MNIST and aim to learn the inversion function $f(x) = 1 - x$ that swaps black and white pixels. We visualize the performance of both models in Figure 28 (Appendix G.2.), where we observe that L2-RDMD fails at performing the desired translation due to the confrontation of the corresponding objectives. At the same time, LPIPS’ ability of edge and structure detection allows for the proper translation between the domain and its inverse without any knowledge about the problem except the unpaired data sets of samples.
> >
> > **References**
> >
> > [1] Scalable computation of monge maps with general costs.
> >
> > [2] Generative modeling through the semi-dual formulation of unbalanced optimal transport.
> >
> > [3] Neural Optimal Transport
> >
> > [4] Diffusion Schrödinger Bridge Matching
> >
> > [5] Adversarial Schrödinger Bridge Matching

---

> ### Comment · Reviewer_cLN3 · 2025-11-26
>
> thanks for your responses. some of my concerns have been addressed and i will raise my ratings accordingly.

---

> > ### Author Response · Authors · 2025-11-26
> > **Clarification of concerns**
> >
> > Thank you for your reply. In our response, we addressed the weaknesses that you identified:
> > 1. We demonstrated that the method is scalable to more challenging higher-dimensional settings on the Cat$\to$Dog $256 \times 256$ benchmark.
> > 2. We showed that an "overly simple" $L_{2}$ regularisation results in a sufficient semantic correspondence by comparing it with LPIPS on the Wild$\to$Cat $64 \times 64$ benchmark. Moreover, we acknowledged the situation where $L_{2}$ fails while a perceptual loss like LPIPS allows us to achieve a proper translation.
> > 3. We considered feature-level regularisation via LPIPS and via latent losses on ImageNet class-to-class and text detoxification problems, which are akin to feature-level regularisation. In an additional Celeba experiment, we demonstrated that $L_{2}$ in latent space could be regarded as more semantically and structurally preserving than $L_{2}$ in pixel space.
> >
> > We believe our answer addresses your concerns. Could you please clarify which of your concerns remain unaddressed and which particular concerns prevent a higher rating?

---

### Official Review · Reviewer_dJn6 · 2025-10-30

**Soundness:** 3
**Presentation:** 3
**Contribution:** 3
**Rating:** 6
**Confidence:** 3

**Summary:**

This manuscript proposes a novel unpaired translation method called RDMD. The method aims to address the limitations of existing approaches: diffusion models, while offering strong realism, are slow and have low faithfulness; Optimal Transport (OT) methods, while fast and faithful, rely heavily on unstable adversarial training. The core innovation of RDMD is replacing the adversarial loss with a stable, DMD, while explicitly adding a transport cost as a regularization term to ensure faithfulness. This design allows RDMD to leverage the strong prior of pre-trained diffusion models, achieve efficient one-step inference, and strike a better balance between realism and faithfulness.

**Strengths:**

This manuscript proposes an unpaired translation method named RDMD, which is validated on both image and text tasks. The writing is relatively clear, the theoretical part is complete, and it provides some inspiration for the community in solving unpaired translation tasks.

**Weaknesses:**

1. Figure 1 is shown, but it is neither mentioned nor described in the manuscript.

2. The proposed method (Equation 9) introduces a "fake" diffusion model, $D_t^{\phi}$, which reframes the training as a coordinate descent process involving both the generator $G_{\theta}$ and this "fake" model $D_t^{\phi}$. My concern is that this approach seemingly exchanges one form of complexity (the instability of adversarial training) for another (the complexity of jointly training two networks).

3. There is a key contradiction between the paper's theoretical claims and its practical results. Theorem 3.1 states that the RDMD solution $G^{\lambda}$ converges to the true Optimal Transport (OT) map $G^*$ only as $\lambda \to 0$. However, the empirical results (e.g., Figure 6) clearly show that the model performs poorly at $\lambda = 0.0$ and achieves its best results at a non-zero $\lambda$ (e.g., 0.2). This implies that the empirically optimal model is *not* the true OT map the theory focuses on. Could the authors please address this gap and clarify whether the asymptotic convergence (Theorem 3.1) is the true justification, or if the method is better understood as an empirically-tuned "regularized DMD" where the transport cost is simply a helpful, non-asymptotic, constraint?

4. The authors selected the quadratic cost function $||x-y||^2$. They claim that in practice, any cost function of interest can be chosen. However, in the image translation experiments, only this single cost function was used. Are there any experiments with other cost functions? To my knowledge, pixel-level losses often lead to image blurriness. How would a perceptual cost (e.g., LPIPS) perform?

5. Given the clear trade-off between FID (realism) and LPIPS (faithfulness) in the experiments (where one improves, the other often degrades), it is difficult to assert which method is "better" based solely on these two automatic metrics. Have the authors considered conducting a human evaluation? For example, presenting human evaluators with paired results from RDMD and the strongest baseline (e.g., DDIB) and asking them to score the outputs along the two dimensions of "image realism" and "similarity to the original image."

**Questions:**

See details in the Weaknesses.

---

> ### Author Response · Authors · 2025-11-21
> **Weaknesses 1 and 2**
>
> Thank you for the valuable discussion and comments! We would like to address your raised weaknesses below:
>
> **Weakness 1.**
> > Figure 1 is shown, but it is neither mentioned nor described in the manuscript.
>
> Thank you for noting it, this Figure is not tied to any experimental discussion and is meant to showcase the model's performance. We will add the appropriate reference in the text.
>
> **Weakness 2.**
> > The proposed method (Equation 9) introduces a "fake" diffusion model, which reframes the training as a coordinate descent process involving both the generator and this "fake" model. My concern is that this approach seemingly exchanges one form of complexity (the instability of adversarial training) for another (the complexity of jointly training two networks).
>
> First, we would like to note that **both** adversarial training and DMD have an additional complexity of jointly training the two networks, not only DMD. However, we believe that there is a significant methodological difference between adversarial training and DMD. Namely, the joint coordinate descent of the DMD procedure seems much more stable than the joint coordinate descent of the adversarial training.
>
> The adversarial nature of the GAN objective leads to the poor performance of the generator in the situations, where one of the two networks succeeds at tricking the opponent. This necessitates careful investigation of the training schedule to ensure balance between networks at each point of the optimization. Moreover, even from the theoretical standpoint very similar approaches as e.g. vanilla GANs [1] and Neural Optimal Transport [2] (if one sets the transport cost $c(x, y) = 0$, NOT will learn a generator of the target distribution, similar to GANs) make different theoretical conclusions. In the vanilla GAN, the objective is **min max**, in NOT the objective is **max min**, which results in more training steps for discriminator suggested in GANs and more training steps for generator suggested in NOT. At the same time, the fake score optimization problem is inherently **inner**, which suggests performing more fake score optimization steps to stabilize the procedure. This observation indeed leads to the stable training as indicated in the DMD2 [3] paper.
>
> **References**
>
> [1] Generative Adversarial Networks
>
> [2] Neural Optimal Transport
>
> [3] Improved Distribution Matching Distillation for Fast Image Synthesis

---

> > ### Author Response · Authors · 2025-11-21
> >
> > **Weakness 3.**
> > > There is a key contradiction between the paper's theoretical claims and its practical results. Theorem 3.1 states that the RDMD solution converges to the true Optimal Transport (OT) map only as $\lambda \to 0$. However, the empirical results (e.g., Figure 6) clearly show that the model performs poorly at and achieves its best results at a non-zero $\lambda$ (e.g., 0.2). This implies that the empirically optimal model is not the true OT map the theory focuses on. Could the authors please address this gap and clarify whether the asymptotic convergence (Theorem 3.1) is the true justification, or if the method is better understood as an empirically-tuned "regularized DMD" where the transport cost is simply a helpful, non-asymptotic, constraint?
> >
> > We respectfully disagree with the reviewer at this point. Note that $\lim\limits_{\lambda \to 0} \arg \min\limits_{G} \mathcal{L}^{\lambda}(G)$ does not always equal to $\arg \min\limits_{G} \lim\limits_{\lambda \to 0} \mathcal{L}^{\lambda}(G)$. In other words, considering a sequence of objectives and taking the limit of the minimizers is not equivalent to considering the limit of the objective and taking its minimizer. Our theorem states that if we consider a sequence of regularized objectives $\mathcal{L}^{\lambda}$, all of which have non-zero regularization coefficients, the corresponding solutions $G^{\lambda}$ converge to the OT map $G^\*$. However, $\lim\limits_{\lambda \rightarrow 0} G^{\lambda}$ does not have to be equal to the solution $G^{\lambda = 0}$ of the unregularized problem, similar to the existence of the discontinuous functions. Moreover, here there are many different solutions to the unregularized problem: any generator that fits the target distribution solves the unregularized problem, regardless of the input-output alignment. Among these solutions is the OT map $G^\*$ we wish to recover, which is not guaranteed without regularization.
> >
> > This main theorem statement may be simpler to analyze if we replace $\mathcal{L}^{\lambda} = \text{KL} + \lambda \mathbb{E} c(x, G(x))$ with the equivalent objective $\frac{1}{\lambda} \text{KL} + \mathbb{E} c(x, G(x))$ and take the corresponding limit. This suggests that under sufficiently small $\lambda$ (but **non-zero**, since we perform division by $\lambda$), the KL weight becomes infinite times larger than the unit weight corresponding to the transport cost. This ensures that even slight difference between distributions, that results in non-zero KL, is strictly penalized with the (almost) $+\infty$ value of the objective. Thus, the optimal generator's output distribution will be (almost) identical to the target distribution. At the same time, among all generators that satisfy the zero-KL property, we will obtain one with the least transport cost due to its explicit minimization.
> >
> > This reasoning is almost identical to the over-parameterized linear regression $\|X w - Y\|^2 $with $L_2$ regularization $\lambda \|w\|^2$. If the number of features is larger than the number of data entries, the standard linear regression has infinitely many solutions. At the same time, $L_2$ regularized regression always has only one solution. If we consider the sequence of the $L_2$ regularized solutions $w_\lambda$ and take the limit as $\lambda \to 0$, we will obtain the set of weights that solves the original linear regression problem and minimizes the $L_2$ norm among the solutions. Here, one should choose **small**, but **non-zero** value of the regularization coefficient, to achieve a point close to the solution with the minimal norm. Identically, in RDMD one should choose **small** but **non-zero** values of $\lambda$ to achieve a generator that is close to the map with the minimal transport cost.

---

> > > ### Author Response · Authors · 2025-11-21
> > >
> > > **Weakness 4.**
> > > > The authors selected the quadratic cost function. They claim that in practice, any cost function of interest can be chosen. However, in the image translation experiments, only this single cost function was used. Are there any experiments with other cost functions? To my knowledge, pixel-level losses often lead to image blurriness. How would a perceptual cost (e.g., LPIPS) perform?
> > >
> > > **Latent-space L2**
> > >
> > > We would like to note that the paper contains latent-space experiments (ImageNet class-to-class translation and text detoxification). Here, the L2 cost is measured in the latent space, which corresponds to a different cost function when considering the translation problem in the initial space. To demonstrate the difference between pixel and latent-space L2, we perform the additional experiment, suggested by the reviewer **wb1N**. In this experiment, we compare training latent RDMD with L2 cost measured in latent space and in pixel space (after decoding). The results are reported in the table below and in Appendix G.3:
> > >
> > > |               | **FID ↓** | **L2 ↓** | **LPIPS ↓** | **PSNR ↑** | **SSIM ↑** |
> > > | ------------- | --------- | -------- | ----------- | ---------- | ---------- |
> > > | **Latent L2** | 32.11     | 54.75    | 0.339       | 21.96      | 0.606      |
> > > | **Pixel L2**  | 33.68     | 52.92    | 0.413       | 18.66      | 0.548      |
> > >
> > > Visually (as indicated by Figure 29, Appendix G.3. in the revised paper), the results are similar, but the facial expression alignment seems to be slightly better in the case of the latent model. This experiment shows that performing L2 in the latent space of the diffusion autoencoder may be considered as a more structure-preserving transport cost than the pixel-space L2.
> > >
> > > **Wild->Cat w/ LPIPS**
> > >
> > > We have added an additional experiment on Wild$\to$Cat $64 \times 64$ benchmark to show the difference between using L2 and perceptual distances as cost functions. We choose LPIPS as the perceptual distance and report the results in the table below and in the revised version of the paper.
> > >
> > > | **Cost**                | **FID ↓** | **L2 ↓**  | **LPIPS ↓** | **PSNR ↑** | **SSIM ↑** |
> > > | ----------------------- | --------- | --------- | ----------- | ---------- | ---------- |
> > > | L2 ($\lambda = 0.1$)    | 17.31 | **14.71** | **0.352**   | **17.75**  | **0.566**  |
> > > | LPIPS ($\lambda = 0.1$) | **17.17** | 19.49     | 0.361       | 15.47      | 0.473      |
> > >
> > > Overall, using LPIPS in this experiment leads to a worse alignment in terms of all similarity metrics (except LPIPS, where the results are comparable) under comparable realism. The overall trade-off achieved by LPIPS is worse in almost all similarity metrics as indicated by Figure 26 in Appendix G.2. Visually, we see little difference between the two costs (please see Appendix G.2, Figure 27 for the visual comparison). This prompts to the fact that in this and the similar benchmarks L2 works sufficiently well at preserving the semantic similarity.
> > >
> > > **Inverted MNIST**
> > >
> > > We also construct an artificial example of color-inverted MNIST to MNIST conversion where L2 would fail to provide meaningful alignment signal, and LPIPS could be used as a cost function. The goal is to learn the inversion function $f(x) = 1 - x$, which swaps between the black and white pixels. We visualize the performance of both models in Figure 28 (Appendix G.2.), where we observe that L2-RDMD fails at performing the desired translation due to the confrontation of the corresponding objectives. At the same time, LPIPS’ ability of edge and structure detection allows for the proper translation between the domain and its inverse without any knowledge about the problem except the unpaired data sets of samples.
> > >
> > > **Weakness 5.**
> > > > Given the clear trade-off between FID (realism) and LPIPS (faithfulness) in the experiments (where one improves, the other often degrades), it is difficult to assert which method is "better" based solely on these two automatic metrics. Have the authors considered conducting a human evaluation? For example, presenting human evaluators with paired results from RDMD and the strongest baseline (e.g., DDIB) and asking them to score the outputs along the two dimensions of "image realism" and "similarity to the original image."
> > >
> > > Overall, comparing FID and LPIPS/other automated similarity metrics seems like a well-established protocol in the OT/I2I literature, where there is a wide range of moderate-scale translation problems instead of e.g. one very large-scale image editing problem. We agree, however, that this addition could further enhance validity of the methods' comparison. Unfortunately, we did not manage to perform human evaluation by the "soft deadline" of the rebuttal and will try our best to perform human evaluation or VLM-based comparison until the end of the discussion period.

---

> > > > ### Comment · Reviewer_dJn6 · 2025-11-27
> > > >
> > > > Thank you to the authors for your reply, which has helped me better understand your work. The current score is appropriate, and I will maintain my score.

---

> > > > > ### Author Response · Authors · 2025-12-03
> > > > > **Human Evaluation**
> > > > >
> > > > > Thank you for the suggestion to run the human evaluation of our method and DDIB (two-sided multi-step diffusion-based method with $\gg 1$ inference steps per translation). We conducted a side-by-side comparison on two benchmarks: Celeba Male$\to$Female $128 \times 128$ and AFHQv2 Wild$\to$Cat $64 \times 64$. We scored the models' performance using three metrics:
> > > > >
> > > > > - **Similarity** (which method produces outputs more similar to the input);
> > > > > - **Realism** (which method produces more realistic outputs with respect to the target distribution);
> > > > > - **Overall Preference** (which method succeeds better in translating the source class to the target class).
> > > > >
> > > > > We report the proportion of participants (in percent), who preferred our method to DDIB and label it as *Win Rate*. A Win rate higher than $50\%$ indicates that the majority of participants preferred our method. The results of the comparison are presented in the tables below:
> > > > >
> > > > > **Celeba Male$\to$Female** $128 \times 128$
> > > > >
> > > > > | **Metric**             | **Win Rate** |
> > > > > | ---------------------- | ------------ |
> > > > > | **Similarity**         | 85.6%        |
> > > > > | **Realism**            | 67.2%        |
> > > > > | **Overall Preference** | 90.9%        |
> > > > >
> > > > > **AFHQv2 Wild$\to$Cat** $64 \times 64$
> > > > >
> > > > > | **Metric**             | **Win Rate** |
> > > > > | ---------------------- | ------------ |
> > > > > | **Similarity**         | 81.0%        |
> > > > > | **Realism**            | 85.0%        |
> > > > > | **Overall Preference** | 87.5%        |
> > > > >
> > > > > Overall, we observe that RDMD outperforms DDIB in terms of human preference, which demonstrates the consistently strong performance of our method, in accordance with automated metrics reported in Table 2 of the paper.

---

### Official Review · Reviewer_wb1N · 2025-10-30

**Soundness:** 4
**Presentation:** 4
**Contribution:** 3
**Rating:** 6
**Confidence:** 4

**Summary:**

This paper proposes Regularized Distribution Matching Distillation (RDMD) for one-step unpaired image-to-image translation. The method builds on Distribution Matching Distillation (DMD) by introducing an optimal transport (OT)-motivated regularization that enforces source faithfulness during translation. Specifically, the generator is trained so that its outputs match the target distribution via DMD, while adding a source–target transport cost regularizer (implemented as a simple pixel-level L2 loss). The paper further provides a theoretical argument showing that the trained generator approximates a Monge OT map. Extensive experiments across a wide range of domains—multiple datasets, resolutions, pixel vs latent spaces, text conditions, and synthetic toy setups—demonstrate strong one-step translation performance and consistent generalization.

**Strengths:**

1. The manuscript is well written, with clear explanations and a well-designed experiment section. The range of datasets and conditions evaluated is notably broad, and the ablations are thoughtful. The appendix contains detailed implementation information, including training setups and dataset details, which makes the work meaningfully reproducible.
2. Unpaired image translation is still a meaningful and relevant task, and the paper’s focus on improving one-step approaches within this setting is justified, and the comparisons against other one-step baselines are appropriate.
3. At first glance, the modification relative to DMD—changing the regularization term from teacher–student to source–generator—could appear incremental. However, the authors provide a clear OT-based justification and a formal argument that the model approximates a Monge OT map. This theoretical framing substantially elevates the contribution beyond a simple loss engineering tweak.
4. Achieving competitive translation quality in a single step, across multiple datasets and resolutions, and demonstrating Pareto-optimal trade-offs between fidelity and style, is compelling.

**Weaknesses:**

1. Even with the introduction of latent space, the experiments are limited to 256 resolution, despite 512 being a commonly expected baseline in modern latent diffusion pipelines. I would like clarification on whether the method was unable to scale to higher resolutions in practice, and whether pixel-space training at 256 resolution was feasible or intentionally not pursued.
2. Similar to DMD, three diffusion models (target, fake, and generator) must be loaded simultaneously during training, which imposes a heavy memory requirement.
3. Training time appears nontrivial. For example, on AFHQ-64 the generator training alone takes approximately three additional days, which suggests that although inference is one-step, the overall training burden remains high.

**Questions:**

1. Is the L2 loss applied in latent space (on the latent experiment)? If so, does this impact source fidelity due to mismatch between perceptual structure and latent geometry? Have you evaluated applying the regularization after decoding back into pixel space, and if so, did it improve or worsen perceptual source consistency?
2. In Table 3, were the diffusion models also trained only on the same limited datasets, or were any pretrained on larger datasets? Clarification is necessary for fair comparison.

---

> ### Author Response · Authors · 2025-11-21
> **Answer to the Weakness 1**
>
> Thank you for your comments and feedback! Below, we address your raised weaknesses and questions:
>
> **Weakness 1.**
> > Even with the introduction of latent space, the experiments are limited to 256 resolution, despite 512 being a commonly expected baseline in modern latent diffusion pipelines. I would like clarification on whether the method was unable to scale to higher resolutions in practice, and whether pixel-space training at 256 resolution was feasible or intentionally not pursued.
>
> The method is scalable in practice, and in principle, there are no obstacles to training in 512x512 or even larger resolutions, given enough compute (since it practically inherits its practical aspects from highly scalable DMD/DMD2 [1] models). From the methodological perspective, our experiments are mostly limited to smaller resolutions (e.g., 64x64 or 128x128 in pixel space) due to our limitation in resources and the focus of the experimental seciton. We aimed at showing diverse applications of the model and comparing it with OT methods, which are painstakingly difficult to scale in practice and are limited to such resolutions. Moreover, this choice allowed us to evaluate a wide range of diffusion-based methods under a single framework, resulting in a comprehensive comparison.
>
> To show that our method is capable of working in a larger resolution, we trained RDMD on Cat$\to$Dog $256\times256$ benchmark in pixel space. We get the following results:
>
> | **Model**              | **FID ↓** | **L2 ↓** | **PSNR ↑** | **SSIM ↑** |
> | ---------------------- | --------- | -------- | ---------- | ---------- |
> | CycleGAN              | 85.9      | -        | -          | -          |
> | StarGANv2              | 54.88     | 133.65   | 10.63      | -          |
> | CUT                   | 76.21     | 59.78    | 17.48      | 0.601      |
> | ITTR (CUT)            | 68.6      | -        | -          | -          |
> | ILVR                   | 74.37     | 56.95    | 17.77      | 0.363      |
> | SDEdit                 | 74.17     | 47.88    | 19.19      | 0.423      |
> | EGSDE                  | 65.82     | 47.22    | 19.31      | 0.415      |
> | EGSDE$^{\dagger}$      | 51.04     | 62.06    | 17.17      | 0.361      |
> | CycleDiffusion         | 58.87     | -        | 18.50      | 0.557      |
> | RDMD ($\lambda = 0.2$) | **44.32** | 64.44    | 16.91      | 0.439      |
>
> Here, RDMD outperforms all the baselines in terms of FID. Compared to the diffusion baselines, it is close in faithfulness to the EGSDE$^{\dagger}$ model while having a significantly better FID score. We were able to train this RDMD checkpoint for only 20 hours on $4 \times$ H100 GPU. It did not fully converge but still shows the best FID among the baselines. We report additional details and visual performance of RDMD in Appendix G.1 and Figure 25 of the revised version of the paper (page 41).
>
> **References**
>
> [1] Improved Distribution Matching Distillation for Fast Image Synthesis

---

> ### Author Response · Authors · 2025-11-21
> **Weakness 2 and 3**
>
> **Weakness 2.**
> > Similar to DMD, three diffusion models (target, fake, and generator) must be loaded simultaneously during training, which imposes a heavy memory requirement.
>
> Indeed, we need to keep three models in memory during the training procedure. However, given a strong initialization and fast convergence obtained by using the pre-trained target diffusion model, we hypothesize that the procedure could be significantly optimized by training small-size adaptations (e.g. LoRA) or freezing parts of the weights.
>
> To verify this, we train RDMD on Cat$\to$Wild (C2W) and Wild$\to$Cat (W2C) $64 \times 64$ benchmarks with limited parameter sets. For the fake score, we freeze all the parameters except the time-embedding layers and the decoder block operating on $64 \times 64$ resolution, resulting in 11M trainable parameters of the network (instead of the original 55M). We also test the same freezing strategy on the generator network. In one case, we freeze only the encoder part, which yields 36M trainable parameters, and in the other, we keep only the last decoder block trainable, obtaining the same 11M count as in the partially frozen fake score. The generator's time-embedding layers remain trainable in both cases. We report the results of this experiment in the table below:
>
> | Model                                        | Fake score trainable parameters | Generator trainable params | Total trainable params | $\text{FID}_{w2c}$ | $\text{LPIPS}_{w2c}$ | $\text{FID}_{c2w}$ | $\text{LPIPS}_{c2w}$ |
> | -------------------------------------------- | ------------------------------- | -------------------------- | ---------------------- | ------------------ | -------------------- | ------------------ | -------------------- |
> | Baseline                                     | 55M                             | 55M                        | 110M                   | 17.31              | 0.352                | 14.43              | 0.374                |
> | partially frozen fake score & full generator | 11M                             | 55M                        | 66M                    | 16.55              | 0.321                | 14.58              | 0.357                |
> | partially frozen generator & fake score      | 11M                             | 36M                        | 47M                    | 16.67              | 0.329                | 14.77              | 0.363                |
> | partially frozen generator & fake score      | 11M                             | 11M                        | 22M                    | 18.01              | 0.315                | 25.73              | 0.342                |
>
> Indeed, freezing 80 percent of the parameters of the fake score does not worsen and even improves metrics on the selected benchmarks. Freezing the generator's encoder part does not harm the performance either. At the same time, reducing the generator's capacity to one-fifth of the original model is detrimental to the overall quality. We believe that reducing the number of trainable parameters in the fake score (and in the generator, to some extent), looks promising and represents a potential way to further optimize the model.
>
> **Weakness 3.**
> > Training time appears nontrivial. For example, on AFHQ-64 the generator training alone takes approximately three additional days, which suggests that although inference is one-step, the overall training burden remains high.
>
> Indeed, training our models until convergence takes a few additional GPU-days. At the same time, Figure 2 from the paper suggests that obtaining high (but slightly suboptimal) performance requires much less time, suggesting overall rapid convergence and room for improvement. Moreover, the model still has the potential for further optimization and acceleration. The important hyperparameter that could boost the performance is the distribution of the noise levels $\sigma$ sampled during training of the generator. In its original form (taken from DMD), $\sigma$s are sampled uniformly from the original EDM inference schedule with $\sigma_{\text{min}} = 0.002$ and $\sigma_{\text{max}} = 80.0$. Our primary experiments suggest that in some benchmarks, the convergence speed could be enhanced by sampling lower sigmas. In the table below, we compare the original uniform schedule with the schedule that uniformly samples from the lower $70\%$ of its values. The benchmark is AFHQv2 Wild$\to$Cat.
>
> |                     | **FID ↓** | **LPIPS ↓** | **Training (4x V100)**    |
> | ------------------- | ------- | --------- | ----------- |
> | **Original**        | 17.82   | 0.365     | ~3 days     |
> | **Lower $\sigma$s** | 17.31   | 0.352     | <1.5 days |
>
> This suggests that choosing the appropriate noise schedule for each becnhmark could result in both superior metrics and accelerating the training time, which we see as an important future work.

---

> > ### Author Response · Authors · 2025-11-21
> > **Questions**
> >
> > **Question 1.**
> > > Is the L2 loss applied in latent space (on the latent experiment)? If so, does this impact source fidelity due to mismatch between perceptual structure and latent geometry? Have you evaluated applying the regularization after decoding back into pixel space, and if so, did it improve or worsen perceptual source consistency?
> >
> > In our latent experiments, we use latent-space L2 regularization. Due to the similar structure of the diffusion autoencoders' latent space to the pixel space, we expect pixel-wise L2 to perform similarly. To verify this, we compare the two variants on CelebA-HQ Male$\to$Female $256 \times 256$ benchmark (and $4 \times 32 \times 32$ latent space of the LDM autoencoder) and obtain the following results:
> >
> > |               | **FID ↓** | **L2 ↓** | **LPIPS ↓** | **PSNR ↑** | **SSIM ↑** |
> > | ------------- | --------- | -------- | ----------- | ---------- | ---------- |
> > | **Latent L2** | 32.11     | 54.75    | 0.339       | 21.96      | 0.606      |
> > | **Pixel L2**  | 33.68     | 52.92    | 0.413       | 18.66      | 0.548      |
> >
> > As can be seen, both models perform similar in terms of realism. Per-pixel similarity for the latent-space L2 model is expectedly a bit worse, but other similarity metrics are better. Visually (as indicated by Figure 29, Appendix G.3. in the revised paper), the results are similar, but the facial expression alignment seems to be slightly better in case of the latent model. Additionally, computing pixel-wise L2 requires additional backpropagation through the auto-encoder, which makes each gradient step $\approx 1.5 \times$ slower. Therefore, we believe using the latent-space L2 is a more reasonable choice due to the faster convergence and (overall comparable, but) better semantic alignment.
> >
> > **Question 2.**
> > > In Table 3, were the diffusion models also trained only on the same limited datasets, or were any pretrained on larger datasets? Clarification is necessary for fair comparison.
> >
> > The diffusion models used in RDMD were also pre-trained on the datasets of the corresponding size, i.e. restricted 5k image dataset for the "5k" experiment and on the full dataset for the "Full data" experiment.

---

> > > ### Comment · Reviewer_wb1N · 2025-11-27
> > >
> > > The authors’ rebuttal has clarified my questions and helped improve my understanding of the paper, and I appreciate their efforts. However, I remain concerned that the maximum resolution is limited to 256, despite operating in a latent domain—a restriction that would have been more understandable had the method worked directly in pixel space. For this reason, I am maintaining my score, which I believe is appropriate and not overly low.

---

> > > > ### Author Response · Authors · 2025-12-03
> > > >
> > > > To help resolve any remaining concerns regarding the scalability of the method, we conducted an additional **latent-space experiment on a subset of the ImageNet dataset with $512 \times 512$ pixel resolution** using the pre-trained EDM2 [1] model as the backbone. This subset comprises five animal classes: Siberian Husky, Chimpanzee, Arabian Camel, Lion, and Persian Cat. The model is trained to perform class-conditional translation between any pairs of these classes. A detailed description of the experimental setup is provided in Appendix G.5 of the revised paper.
> > > >
> > > > This experiment demonstrates the scalability of our method to higher-dimensional data and its ability to perform unpaired image-to-image translation with satisfactory alignment. We present curated samples in Figure 31.
> > > >
> > > > [1] Analyzing and Improving the Training Dynamics of Diffusion Models.

---

> ### Author Response · Authors · 2025-11-27
>
> Thank you for your response and for appreciating our rebuttal! We would like to kindly ask for a brief clarification regarding the following part of your evaluation:
>
> > “However, I remain concerned that the maximum resolution is limited to 256, despite operating in a latent domain—a restriction that would have been more understandable had the method worked directly in pixel space.”
>
> In our response to weakness 1 regarding dimensionality, we reported an additional experiment on AFHQv2 Cat→Dog, conducted in **256×256 pixel space** rather than in the latent domain.
>
> We would appreciate it if you could let us know whether this experiment addresses your concern regarding the resolution limitations.

---

### Official Review · Reviewer_93vz · 2025-10-31

**Soundness:** 3
**Presentation:** 2
**Contribution:** 2
**Rating:** 4
**Confidence:** 4

**Summary:**

The paper proposes Regularized Distribution matching distillation for one-step image translation.

**Strengths:**

It proposes simple and intuitive approach to include OT-based regularization into DMD in order to make one-step diffusion model for I2I.

**Weaknesses:**

1. Although the paper proposes that GAN-based models are mostly superior to EGSDE, I am a bit suspicious on these statements. There are so many GAN-based Image translation methods such as StarGAN, StarGANv2, CUT, CycleGAN, etc. These methods show great FID score when it comes to AFHQ and CelebA-HQ. To clearly show the advantage of proposed one-step model, please include the GAN-based methods. Also, please show comparison output between recent zero-shot editing based methods (prompt-to-prompt, Nano Banana.. etc) for thorough evaluation.

2. Since the methods rely on DMD, the performance of I2I is limited on the teacher diffusion backbone. Also the DMD-based method requires additional network (fake teacher), the computation complexity and training time increases. Please show proper comaprison on this.

**Questions:**

No

---

> ### Author Response · Authors · 2025-11-21
>
> Thank you for your feedback and comments! Below we address your concerns:
>
> > *Although the paper proposes that GAN-based models are mostly superior to EGSDE, I am a bit suspicious on these statements. ... To clearly show the advantage of proposed one-step model, please include the GAN-based methods.*
>
> Our paper asserts that the GAN-based I2I methods are inferior to EGSDE. This claim is supported by the results reported in Table 1 in the original EGSDE paper. Specifically, the Cat$\to$Dog $256 \times 256$ benchmark in this paper contains a comprehensive comparison across many GAN-based methods (including CycleGAN, StarGANv2, and CUT) and diffusion-based methods. To address the comparison with GAN-based methods directly, we train RDMD on this benchmark and report the results in the table below (the results for other methods are taken directly from Table 1 of the EGSDE paper):
>
> | **Model**              | **FID ↓** | **L2 ↓** | **PSNR ↑** | **SSIM ↑** |
> | ---------------------- | --------- | -------- | ---------- | ---------- |
> | CycleGAN              | 85.9      | -        | -          | -          |
> | StarGANv2              | 54.88     | 133.65   | 10.63      | -          |
> | CUT                   | 76.21     | 59.78    | 17.48      | 0.601      |
> | ITTR (CUT)            | 68.6      | -        | -          | -          |
> | ILVR                   | 74.37     | 56.95    | 17.77      | 0.363      |
> | SDEdit                 | 74.17     | 47.88    | 19.19      | 0.423      |
> | EGSDE                  | 65.82     | 47.22    | 19.31      | 0.415      |
> | EGSDE$^{\dagger}$      | 51.04     | 62.06    | 17.17      | 0.361      |
> | CycleDiffusion         | 58.87     | -        | 18.50      | 0.557      |
> | RDMD ($\lambda = 0.2$) | **44.32** | 64.44    | 16.91      | 0.439      |
>
> RDMD significantly outperforms GAN-based methods in FID, which further supports our initial claim. Due to the limitations in resources, we were able to train this RDMD checkpoint for only 20 hours on $4 \times$ H100 GPU. It did not fully converge but still shows the best FID among the baselines. We report additional details and visual performance of RDMD in Appendix G.1 and Figure 25 of the revised version of the paper (page 41).
>
> > *Please show comparison output between recent zero-shot editing based methods (prompt-to-prompt, Nano Banana.. etc) for thorough evaluation*
>
> We compare RDMD with Gemini 2.5 Flash Image (aka Nano Banana) on the _Animals_, _Birds_, _Insects_ and _Fish_ ImageNet benchmarks, which consist of pairwise translations between the 5 selected classes in a given benchmark, excluding identity translations (for more details, see Section 4.2 of the paper). To use Nano Banana, we formulate each translation problem as an image editing problem using the following prompts and replacing placeholders with the corresponding classes:
> 1. ```Transform {source_cls} into {target_cls}. Keep the scene, background, lighting, colours, camera viewpoint, and all non-{source_cls} objects unchanged. For every transformation, preserve the original position, scale, pose, and interactions.```
> 2. ```Replace the {source_cls} with a {target_cls}. Do NOT alter any humans or people in the image. Maintain the exact same composition, lighting, and background.```
>
> The second prompt was used only when the first prompt failed and didn't pass the model's internal safety checks when humans were present in the input image. We didn't notice any significant difference between the outputs using these prompts. We report the following results:
>
> **Animals**
>
> | **Model**       | **FID**   | **LPIPS** |
> | --------------- | --------- | --------- |
> | Nano Banana | 42.29     | 0.386     |
> | RDMD        | **39.85**     | **0.369** |
>
> **Birds**
>
> | **Model**       | **FID**   | **LPIPS** |
> | --------------- | --------- | --------- |
> | Nano Banana | 37.97     | **0.331** |
> | RDMD        | **24.87**     | 0.415     |
>
> **Insects**
>
> | **Model**       | **FID**   | **LPIPS** |
> | --------------- | --------- | --------- |
> | Nano Banana | 39.94     | **0.231** |
> | RDMD        | **29.57**     | 0.296     |
>
> **Fish**
>
> | **Model**       | **FID**   | **LPIPS** |
> | --------------- | --------- | --------- |
> | Nano Banana | 52.38     | **0.280** |
> | RDMD        | **34.00**     | 0.329     |
>
> Here, we observe that RDMD beats NanoBanana in terms of FID, but typically has worse input-output alignment measured in LPIPS. In this problem, a better FID does not directly imply better realism, since the desired translation may, e.g., place realistic animals in non-realistic environments, which leads to a bias compared to the original distribution. These scenarios are present for both models but are more pronounced for NanoBanana due to its stricter alignment with the input object shapes.

---

> ### Author Response · Authors · 2025-11-21
>
> We also report visual comparison between RDMD and NanoBanana in Figure 30 (Appendix G.4, page 45 of the revised paper). NanoBanana typically excels at alignment and has high realism. In the case of the Animals benchmark, however, RDMD outperforms it in both metrics. Despite NanoBanana's remarkable translation quality, RDMD still offers competitive performance in terms of both metrics and visual performance. At the same time, NanoBanana is a closed-source model, its inference procedure, number of parameters and architecture are unknown, but are based on (assumably) multi-billion commercial foundation model. ImageNet RDMD has 400M parameters, takes less than 10 GPU-days of training given the pre-trained LDM, and offers efficient one-step inference with 3 total neural network evaluations (encode, translate, decode). Given this, we believe that RDMD achieves strong results and clearly demonstrates its potential for image editing tasks and general optimal transport problems.
>
> > _Since the methods rely on DMD, the performance of I2I is limited on the teacher diffusion backbone. Also the DMD-based method requires additional network (fake teacher), the computation complexity and training time increases. Please show proper comaprison on this._
>
> The quality of DMD and our method is indeed upper-bounded by the teacher diffusion backbone. However, in principle, it is possible to use an additional adversarial loss with a discriminator initialised from the encoder of the teacher model (akin to DMD2 [1] ) to outperform the teacher model.
>
> **Computational Complexity and Training Time**
>
> We provide complete training details (including training time on various benchmarks) for RDMD in Apppendix E.
>
> Among the baselines we compare with, only OT methods (and EGSDE, but it does not train generative models, only a classifier) namely ASBM and DIOTM, require additional training.  We report the training time comparison between OT method and RDMD on different benchmarks in the tables below:
>
> **AFHQv2 Wild$\to$Cat 64**
>
> | **Method** | **Training Time**  |
> | ---------- | ------------------ |
> | RDMD  | ~3 days, 4 V100 32 Gb    |
> | ASBM   | ~8 days, 1 V100 32 Gb   |
> | DIOTM  | ~1 day 18h, 1 V100 32 Gb |
>
> **CelebA Male$\to$Female 64**
>
> | **Method** | **Training Time**                            |
> | ---------- | -------------------------------------------- |
> | RDMD  | ~3 days, 4 V100 32Gb                              |
> | ASBM   | ~2 days 7h, 4 A100 40Gb + ~1 day 8 hours, 8 A100 40 Gb |
> | DIOTM  | ~1 day 18h, 1 V100 32 Gb                           |
>
> **CelebA Male$\to$Female 128**
>
> | **Method** | **Training Time** |
> | ---------- | ----------------- |
> | RDMD   | ~3.5 days, 8 A100 80 Gb |
> | ASBM   | ~7-10 days*, 1 A100 80 Gb  |
> | DIOTM  | 1 day 15h, 1 A100 80 Gb|
>
> (*) In case of the ASBM on CelebA Male$\to$Female 128, 7 days of training is reported in the original paper. In our runs, we have obtained approximately 10 days.
>
> Though RDMD typically requires larger training time, it almost always achieves better results in terms of both faithfulness and realism. At the same time, the experiment in Table 3 and Figure 4 showcases that RDMD is much more robust to the dataset size and does not generate artifacts in contrast to adversarial methods. Thus, the computational overhead is traded to the better quality of the method.
>
> **Potential improvement**
>
> Moreover, Figure 2 suggests that obtaining high (but slightly suboptimal) performance requires much less time, suggesting overall rapid convergence and room for improvement. Thus, the model still has the potential for further optimization and acceleration. The important hyperparameter that could boost the performance is the distribution of the noise levels $\sigma$ sampled during training of the generator. In its original form (taken from DMD), $\sigma$s are sampled uniformly from the original EDM inference schedule with $\sigma_{\text{min}} = 0.002$ and $\sigma_{\text{max}} = 80.0$. Our primary experiments suggest that in some benchmarks, the convergence speed could be enhanced by sampling lower sigmas. In the table below, we compare the original uniform schedule with the schedule that uniformly samples from the lower $70\%$ of its values. The benchmark is AFHQv2 $64 \times 64$ Wild$\to$Cat.
>
> |                     | **FID ↓** | **LPIPS ↓** | **Training (4x V100)**    |
> | ------------------- | ------- | --------- | ----------- |
> | **Original**        | 17.82   | 0.365     | ~3 days     |
> | **Lower $\sigma$s** | 17.31   | 0.352     | <1.5 days |
>
> This suggests that choosing the appropriate noise schedule for each benchmark could result in both superior metrics and accelerating the training time, which we see as an important future work.
>
> **References**
>
> [1] Improved Distribution Matching Distillation for Fast Image Synthesis

---

### Author Response · Authors · 2025-12-03
**Summary of Contributions, Revisions, and Response to Concerns**

Dear Area Chair and Reviewers,

Given the special circumstances of this year's review process, we have prepared a summary of our work for your convenience. It summarizes our contribution and demonstrates how our revisions address the reviewers' concerns. We genuinely thank the reviewers for their valuable feedback and discussion.

## Our contribution

In this work, we introduce a novel method for solving the optimal transport (OT) problem (with applications to unpaired **image to image** translation and **text detoxification**), called Regularized Distribution Matching Distillation (RDMD). This method solves the optimal transport problem by explicitly pushing faithfulness and realism of the performed tranlsation via combining transport cost minimization with the DMD loss. We obtain the method that achieves a superior/comparable faithfulness-realism trade-off compared to the baselines (Figure 3, Table 2, Table 4) and combines the best qualities of diffusion- and OT-based baselines:
* **High faithfulness**, which is typically hard to achieve with diffusion-based methods;
* **Competitive realism**, which is often complicated by the OT methods' tendency towards artifacts and their data-hungry nature (Table 3, Figure 4);
* **Theoretical ground** (RDMD approximates the OT map, Theorem 3.1);
* **Fast convergence**, shown in Figure 2;
* **Applicability in different domains**: pixel space (Figure 3, Table 2), VAE latent space (Figure 5, Table 4), and latent space of the continuous text diffusion models;
* **Applicability for conditional translation**: we train one model for class-to-class ImageNet (Section 4.2) tranlsation, which is akin to the large-scale image editing problems.

## Feedback of The Reviewers

The reviewers have highlighted several aspects of the work:

* **Clear writing of the manuscript** (wb1N, dJn6, cLN3);
* The approach is **simple and intuitive** (93vz), application of DMD in I2I problems **is interesting** (cLN3) and **provides inspiration** for solving the unpaired translation tasks (dJn6);
* The approach is **theoretically grounded** (wb1N, dJn6);
* **Well-designed experimental section**, broad range of datasets and conditions, implementation information (wb1N).

Among the reviewers' concerns and questions were: (we highlight the **addressed** conserns with **+** and the concerns, where we respectfully disagree with the reviewers with * ):
1) **(+)** Absence of experiments in $256 \times 256$ resolution in pixel space (wb1N, cLN3) or $512 \times 512$ resolution in latent space (wb1N);
2) **(+)** "L2 is often insufficient to enforce semantic correspondence" (cLN3); "Pixel-level losses often lead to image blurriness. How would a perceptual cost (e.g., LPIPS) perform?" (dJn6); suggestion to compare L2 in pixel and latent space (wb1N);
3) **(+)** Suggestion to compare with the GAN-based methods and suspicion about their worse performance compared to EGSDE (93vz); suggestion to compare with zero-shor image editing models as e.g. NanoBanana (93vz);
4) **(+)** Joint training of the two networks can be costly (93vz, wb1N);
5) **(+)** Suggestion to add human evaluation (dJn6);
6)  * Contradiction between the paper's theoretical claims and its practical results (dJn6); DMD training may be equally complex as adversarial training (dJn6);

---

> ### Author Response · Authors · 2025-12-03
> **Summary of Contributions, Revisions, and Response to Concerns**
>
> ## Our updates in the revision
>
> To address the raised concerns, we have
>
> 1) Conducted **both** pixel-space $256 \times 256$ (AFHQv2 Cat$\to$Dog) and latent-space $512 \times 512$ (ImageNet Animals$\to$Animals multiclass translation) experiments;
> * **(256 pixel space)** In our answers and Table 7, Figure 25 we demonstrate RDMD's performance in Cat$\to$Dog, where it outperforms **all** the baselines in terms of FID, while being trained for only 20 hours on 4x H100 GPU;
> * **(512 latent space)** In our answer to reviewer wb1N and Appendix G.5 we demonstrate the RDMD's capability of scaling to **$512 \times 512$ resolution** in **multiclass** translation and its applicability for different architectures (here, **EDM2** is used);
>
> 2) Conducted the latent CelebA-HQ 256 Male$\to$Female experiment (answer to wb1N, cLN3) that demonstrates that the **latent-space L2 loss allows for the better semantic alignment** than the pixel-space L2 loss, thus already covering the more perceptual-oriented distances;
> * Verified that in our benchmarks (AFHQv2 Wild$\to$Cat) **L2 may itself achieve sufficient semantic alignment** and perform better than the perceptual-based LPIPS (novel experiment, answers to dJn6, cLN3; Table 8, Figure 26);
> * Conduct the experiment, where L2 fails due to the nature of the problem, but RDMD with the different cost still works: MNIST$\to$ Inverted MNIST experiment (targeting to swap black and white pixels). RDMD with LPIPS here achieves good translation (Appendix G.2, Figure 28), while RDMD-L2 does not;
>
> 3) Provided the comparison with the GAN-based methods (answer to 93vz, Table 7), where RDMD **significantly outperforms all the GAN-based methods** in the novel AFHQv2 256 experiment;
> * Provided the comparison with the **multi-billion, commercial** NanoBanana model (answer to 93vz, Table 10, Figure 30), where **RDMD beats NanoBanana** in the Animals ImageNet benchmark. In other benchmarks, RDMD has inferior alignment, but still competitive results;
>
> 4) Showed in the new experiment that RDMD can maintain its performance while training **only 20% of the fake score weights** and **65% of the generator weights** (answer to wb1N);
> * Showed in the new experiment that RDMD's convergence can be further amplified via tuning the noise level sampling strategy: $3\to 1.5$ training days on AFHQv2 Cat$\to$Wild with better quality (answer to wb1N, 93vz);
>
> 5) Performed the human evaluation in $64 \times 64$ Wild$\to$Cat and $128 \times 128$ Male$\to$Female experiments, where RDMD is preferable in $65 - 90$ percent of cases in terms of both faithfulness and realism (answer to dJn6);
>
> 6) Described in detail the intuition behind our main theoretical results and why DMD loss is a more stable procedure than the adversarial loss (answer to dJn6).
>
> ## Conclusion
>
> The reviews and the rebuttal phase have produced a very valuable discussion and a plethora of new experiments and insights about our work. We believe our detailed responses, new experiments, and manuscript revisions fully address the key concerns raised during the review. The new results, particularly **scalability to 256 pixel and 512 latent space**, **significantly better performance compared to GANs**, sometimes even **competitive performance compared to the NanoBanana model** and  **improvements in training efficiency**, strongly reinforce the paper's core contribution and practical value. We are grateful for the reviewers' feedback, which has significantly strengthened the paper. Thank you for your time and consideration.

---

### Meta-Review · Area_Chair_ddaf · 2026-01-05

**Summary:**

The paper received mixed scores: two borderline accepts (6, wb1N & dJn6), one borderline reject (4, 93vz), and one reject (2, cLN3). The reviewers appreciated the simplicity, theoretical foundation, and strong results of he proposed method as well as the clarity of the paper. However, they also raised multiple concerns with increased training time and computational complexity (93vz, wb1N), experiments limited to small resolutions (wb1N, cLN3), lack of comparisons with relevant work (93vz), lack of ablation study on the cost-regularizer (dJn6, cLN3), contradiction between the theoretical claim and the practical results (dJn6), lack of human evaluation (dJn6). The authors responded to these comments through the rebuttal and revision, but failed to assuage all of them, in particular, those with the complexity in training and small image resolutions. Even among the OT techniques, the proposed method demands more training time and GPU resources. The authors claimed that the computational overhead is traded to better generation quality, but to support this assertion, they should provide additional evaluation and analysis (e.g., comparisons with the competitors trained under the same conditions). Also, the last response to the reviewer 93vz, which was provided to demonstrate the potential improvement of training efficiency though, shows that the proposed method is sensitive to the hyperparameter setting. Regarding the resolution issue, the authors failed to present experimental results on higher resolution images and thus even a positive reviewer (wb1N) was not satisfied by the response and did not upgrade his/her initial rating. They finally provided experimental results, but on a tiny subset of ImageNet-512 with only a few qualitative results without quantitative analysis or comparisons with prior work. Putting these together, the AC considers the remaining concerns outweigh the positive comments and the rebuttal, and thus regrets to recommend rejection. The authors are encouraged to reflect the reviewers' valuable comments and submit to next venues.

**Reviewer Concerns:**

[Concerns that have not been fully resolved]
- Increased training time and computational complexity (93vz, wb1N)
	- *The authors compared their method only with other OT techniques (that require additional training as the proposed method does) in terms of training time and computational complexity. Even among the OT techniques, the proposed one demanded more training time and more GPU resources. The authors argued that the computational overhead is traded to the better generation quality. The AC believes, however, that additional evaluation and analysis such as comparisons with the competitors trained under the same conditions must be presented to show if the tradeoff is beneficial. Also, the AC guesses that the reviewer 93vz may want to quantify the computational overhead of the method over non-OT methods to better evaluate the complexity-quality tradeoff of the method, which however was not reported in the rebuttal. Moreover, the last response to the reviewer 93vz, which was provided to demonstrate the potential improvement of training efficiency though, shows that the proposed method is sensitive to the hyperparameter setting.*
- Experiments limited to small resolutions (wb1N, cLN3)
	- *The authors did not present any experimental results on higher-resolution images, yet simply mentioned that the low-resolution setting was inevitable for experiments given limited computing resources, for comprehensive comparisons with previous work, and for demonstrating diverse applications. (The AC suspects this issue is connected to the high training complexity discussed above and thus is hard to address.) The reviewer wb1N was not satisfied by the response and kept his/her initial rating.*
	- *The authors finally provided experimental results on a tiny subset of ImageNet-512, but only a few qualitative results were provided without either quantitative analysis or comparison with prior work.*

[Concerns that have been successfully addressed]
- Lack of comparisons with relevant work (93vz)
	- *The reviewer 93vz asked comparisons with GAN-based image translation methods and recent zero-shot editing techniques for thorough evaluation. As suggested the authors compared their method with these both qualitatively and quantitatively; the proposed method achieved favorable performance especially when regarding that the model has not been sufficiently trained due to the limited time and resources.*
- Contradiction between the theoretical claim and the practical results (dJn6)
	- *This has been well addressed by a thorough response and theoretical analysis in the rebuttal. The reviewer dJn6 looks satisfied by the response.*
- Lack of ablation study on the cost-regularizer (dJn6, cLN3)
	- *The reviewer dJn6 pointed out that there is no empirical verification for the argument that the proposed method can employ any cost function of interest. Also, the reviewer asked the justification for the use of a pixel-level cost and the results of the method when using a perceptual cost. The reviewer cLN3 also suggested using a perceptual cost instead of the pixel-level L2 one. The authors addressed this issue by comparing results of applying the L2 loss to the pixel space and those of applying the same loss to a latent space (something like a perceptual cost). The AC considers this concern was successfully assuaged, since, although only two cases are compared, the results demonstrate that the method can employ different cost functions and works well with a perceptual cost.*
- Unclear practical advantage of the proposed method in quantitative analysis (dJn6)
	- *For this reason, the reviewer dJn6 recommended to add human evaluation results. The authors conducted human evaluation as suggested and reported the results in the last response to the reviewer. However, details of the evaluation (e.g., the number of people involved in this evaluation) are missing and its scope is limited (only two benchmarks with tiny resolutions). Nevertheless, since the reviewer looks satisfied, the AC considers this concern has been assuaged.*
- Minor presentation issues (dJn6)
	- *They are not sole ground for rejection.*

**Reviewer Scores:**

The two positive reviewers (wb1N, dJn6) left their final comments and kept their initial ratings (6, borderline accepts). The most negative reviewer (cLN3) mentioned that he or she upgraded the rating, but the AC guesses their final score was still below the borderline according to the last comment of the authors. The AC expects that the other negative reviewer kept his / her initial negative rating even if the discussion has continued, since the concern with the complexity in training has not been successfully addressed.

---

### Decision · Program_Chairs · 2026-01-26

Reject